# Get rich quick: exact solutions reveal how unbalanced initializations promote rapid feature learning

**Daniel Kunin**[*1]   **Allan Raventós**[*1]   **Clémentine Dominé**[2]   **Feng Chen**[1]
**David Klindt**[3]   **Andrew Saxe**[2]   **Surya Ganguli**[1]

[1]Stanford University   [2]University College London   [3]Cold Spring Harbor Laboratory

## Abstract

While the impressive performance of modern neural networks is often attributed to their capacity to efficiently extract task-relevant features from data, the mechanisms underlying this *rich feature learning regime* remain elusive, with much of our theoretical understanding stemming from the opposing *lazy regime*. In this work, we derive exact solutions to a minimal model that transitions between lazy and rich learning, precisely elucidating how unbalanced *layer-specific* initialization variances and learning rates determine the degree of feature learning. Our analysis reveals that they conspire to influence the learning regime through a set of conserved quantities that constrain and modify the geometry of learning trajectories in parameter and function space. We extend our analysis to more complex linear models with multiple neurons, outputs, and layers and to shallow nonlinear networks with piecewise linear activation functions. In linear networks, rapid feature learning only occurs from balanced initializations, where all layers learn at similar speeds. While in nonlinear networks, unbalanced initializations that promote faster learning in earlier layers can accelerate rich learning. Through a series of experiments, we provide evidence that this unbalanced rich regime drives feature learning in deep finite-width networks, promotes interpretability of early layers in CNNs, reduces the sample complexity of learning hierarchical data, and decreases the time to grokking in modular arithmetic. Our theory motivates further exploration of unbalanced initializations to enhance efficient feature learning.

## 1   Introduction

Deep learning has transformed machine learning, demonstrating remarkable capabilities in a myriad of tasks ranging from image recognition to natural language processing. It's widely believed that the impressive performance of these models lies in their capacity to efficiently extract task-relevant features from data. However, understanding this feature acquisition requires unraveling a complex interplay between datasets, network architectures, and optimization algorithms. Within this framework, two distinct regimes, determined at initialization, have emerged: the lazy and the rich.

**Lazy regime.** Various investigations have revealed a notable phenomenon in overparameterized neural networks, where throughout training the networks remain close to their linearization [1, 2, 3, 4, 5]. Seminal work by Jacot et al. [6], demonstrated that in the infinite-width limit, the Neural Tangent Kernel (NTK), which describes the evolution of the neural network through training, converges to a deterministic limit. Consequently, the network learns a solution akin to kernel regression with the NTK matrix. Termed the *lazy* or *kernel* regime, this domain has been characterized by a deterministic NTK [6, 7], minimal movement in parameter space [8], static hidden representations, exponential learning curves, and implicit biases aligned with a reproducing kernel Hilbert space (RKHS) norm [9]. However, Chizat et al. [8] challenged this understanding, asserting that the lazy regime isn't a

---

*Equal contribution. Correspondence to `kunin@stanford.edu` and `aravento@stanford.edu`.

38th Conference on Neural Information Processing Systems (NeurIPS 2024).

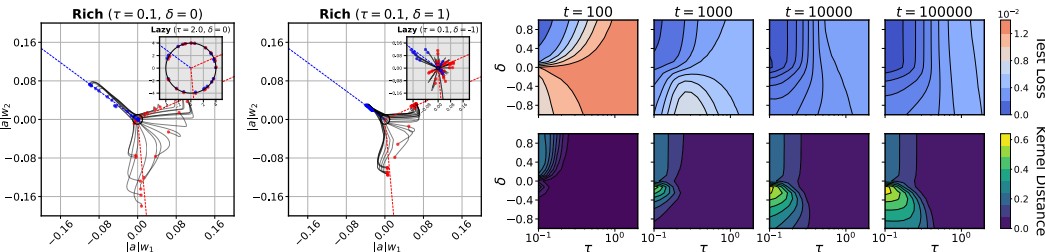

(a) Overall and relative scale impact feature learning    (b) A complex phase portrait of feature learning

Figure 1: **Unbalanced initializations lead to rapid rich learning and generalization.** We follow the experimental setup used in Fig. 1 of Chizat et al. [8] – a wide two-layer student ReLU network $f(x;\theta) = \sum_{i=1}^{h} a_i \max(0, w_i^\intercal x)$ trained on a dataset generated from a narrow two-layer teacher ReLU network. The student parameters are initialized as $w_i \sim \text{Unif}(\mathbb{S}^{d-1}(\frac{\tau}{\alpha}))$ and $a_i = \pm\alpha\tau$, such that $\tau > 0$ controls the *overall scale* of the function, while $\alpha > 0$ controls the *relative scale* of the first and second layers through the conserved quantity $\delta = \tau^2(\alpha^2 - \alpha^{-2})$. (a) Shows the training trajectories of $|a_i|w_i$ (color denotes $\text{sgn}(a_i)$) when $d = 2$ for four different settings of $\tau, \delta$. The left plot confirms that small overall scale leads to rich and large overall scale to lazy. The right plot shows that even at small overall scale, the relative scale can move the network between rich and lazy as well. Here an upstream initialization $\delta > 0$ shows striking alignment to the teacher (dotted lines), while a downstream initialization $\delta < 0$ shows no alignment. (b) Shows the test loss and kernel distance from initialization computed through training over a sweep of $\tau$ and $\delta$ when $d = 100$. Lazy learning happens when $\tau$ is large, rich learning happens when $\tau$ is small, and rapid rich learning happens when *both* $\tau$ is small and $\delta$ is large – an upstream initialization. This initialization also leads to the smallest test loss. See Fig. 10 in Appendix D.1 for supporting figures.

product of the infinite-width architecture, but is contingent on the *overall scale* of the network at initialization. They demonstrated that given any finite-width model $f(x;\theta)$ whose output is zero at initialization, a scaled version of the model $\tau f(x;\theta)$ will enter the lazy regime as the scale $\tau$ diverges. However, they also noted that these scaled models often perform worse in test error. While the lazy regime offers insights into the network's convergence to a global minimum, it does not fully capture the generalization capabilities of neural networks trained with standard initializations. It is thus widely believed that a different regime, driven by small or vanishing initializations, underlies the many successes of neural networks.

**Rich regime.** In contrast to the lazy regime, the *rich* or *feature-learning* or *active* regime is distinguished by a learned NTK that evolves through training, non-convex dynamics traversing between saddle points [10, 11, 12], sigmoidal learning curves, and simplicity biases such as low-rankness [13] or sparsity [14]. Yet, the exact characterization of rich learning and the features it learns frequently depends on the specific problem at hand, with its definition commonly simplified as what it is not: lazy. Recent analyses have shown that beyond overall scale, other aspects of the initialization can substantially impact the extent of feature learning, such as the effective rank [15], layer-specific initialization variances [16, 17, 18], and large learning rates [19, 20, 21, 22]. Azulay et al. [9] demonstrated that in two-layer linear networks, the relative difference in weight magnitudes between the first and second layer, termed the *relative scale* in our work, can impact feature learning, with balanced initializations yielding rich learning dynamics, while unbalanced ones tend to induce lazy dynamics. However, as shown in Fig. 1, for nonlinear networks unbalanced initializations can induce both rich and lazy dynamics, creating a complex phase portrait of learning regimes influenced by both overall and relative scale. Building on these observations, our study aims to precisely understand how layer-specific initialization variances and learning rates determine the transition between lazy and rich learning in finite-width networks. Moreover, we endeavor to gain insights into the inductive biases of both regimes, and the transition between them, during training and at interpolation, with the ultimate goal of elucidating how the rich regime acquires features that facilitate generalization.

**Our contributions.** Our work begins with an exploration of the two-layer single-neuron linear network proposed by Azulay et al. [9] as a minimal model displaying both lazy and rich learning. In Section 3, we derive exact solutions for the gradient flow dynamics with layer-specific learning rates of this model by employing a combination of hyperbolic and spherical coordinate transformations.

Alongside recent work by Xu and Ziyin [23][1], our analysis stands out as one of the few analytically tractable models for the transition between lazy and rich learning in a finite-width network, marking a notable contribution to the field. Our analysis reveals that the layer-specific initialization variances and learning rates conspire to influence the learning regime through a simple set of conserved quantities that constrain the geometry of learning trajectories. Additionally, it reveals that a crucial aspect of the relative scale overlooked in prior analysis is its directionality. While a *balanced initialization* results in all layers learning at similar rates, an *unbalanced initialization* can cause faster learning in either earlier layers, referred to as an *upstream initialization*, or later layers, referred to as a *downstream initialization*. Due to the depth-dependent expressivity of layers in a network, upstream and downstream initializations often exhibit fundamentally distinct learning trajectories. In Section 4 we extend our analysis of the relative scale developed in the single-neuron model to more complex linear models with multiple neurons, outputs, and layers and in Section 5 to two-layer nonlinear networks with piecewise linear activation functions. We find that in linear networks, rapid rich learning can only occur from balanced initializations, while in nonlinear networks, upstream initializations can actually accelerate rich learning. Finally, through a series of experiments, we provide evidence that upstream initializations drive feature learning in deep finite-width networks, promote interpretability of early layers in CNNs, reduce the sample complexity of learning hierarchical data, and decrease the time to grokking in modular arithmetic.

**Notation.** In this work, we consider a feedforward network $f(x; \theta) : \mathbb{R}^d \to \mathbb{R}^c$ parameterized by $\theta \in \mathbb{R}^m$. Unless otherwise specified, $c = 1$. The network is trained by gradient flow $\dot{\theta} = -\eta_\theta \cdot \nabla_\theta \mathcal{L}(\theta)$, with an initialization $\theta_0$ and layer-specific learning rate $\eta_\theta \in \mathbb{R}^m_+$, to minimize the mean squared error $\mathcal{L}(\theta) = \frac{1}{2} \sum_{i=1}^n (f(x_i; \theta) - y_i)^2$ computed over a dataset $\{(x_1, y_1), \dots, (x_n, y_n)\}$ of size $n$. We denote the input matrix as $X \in \mathbb{R}^{n \times d}$ with rows $x_i \in \mathbb{R}^d$ and the label vector as $y \in \mathbb{R}^n$. The network's output $f(x; \theta)$ evolves according to the differential equation, $\partial_t f(x; \theta) = \sum_{i=1}^n \Theta(x, x_i; \theta)(y_i - f(x_i; \theta))$, where $\Theta(x, x'; \theta) : \mathbb{R}^d \times \mathbb{R}^d \to \mathbb{R}$ is the *Neural Tangent Kernel (NTK)*, defined as $\Theta(x, x'; \theta) = \sum_{p=1}^m \eta_{\theta_p} \partial_{\theta_p} f(x; \theta) \partial_{\theta_p} f(x'; \theta)$. The NTK quantifies how one gradient step with data point $x'$ affects the evolution of the networks's output evaluated at another data point $x$. When $\eta_{\theta_p}$ is shared by all parameters, the NTK is the kernel associated with the feature map $\nabla_\theta f(x; \theta) \in \mathbb{R}^m$. We also define the *NTK matrix* $K \in \mathbb{R}^{n \times n}$, which is computed across the training data such that $K_{ij} = \Theta(x_i, x_j; \theta)$. The NTK matrix evolves from its initialization $K_0$ to convergence $K_\infty$ through training. Lazy and rich learning exist on a spectrum, with the extent of this evolution serving as the distinguishing factor. Various studies have proposed different metrics to track the evolution of the NTK matrix [24, 25, 26]. We use *kernel distance* [27], defined as $S(t_1, t_2) = 1 - \langle K_{t_1}, K_{t_2} \rangle / (\|K_{t_1}\|_F \|K_{t_2}\|_F)$, which is a scale invariant measure of similarity between the NTK at two times. In the lazy regime $S(0, t) \approx 0$, while in the rich regime $0 \ll S(0, t) \leq 1$.

## 2 Related Work

**Linear networks.** Significant progress in studying the rich regime has been achieved in the context of linear networks. In this setting, $f(x; \theta) = \beta(\theta)^\intercal x$ is linear in its input $x$, but can exhibit highly nonlinear dynamics in parameter $\theta$ and function $\beta(\theta)$ space. Foundational work by Saxe et al. [10] provided exact solutions to gradient flow dynamics in linear networks with task-aligned initializations. They achieved this by solving a system of Bernoulli differential equations that prioritize learning the most salient features first, which can be beneficial for generalization [28]. This analysis has been extended to wide [29, 30] and deep [31, 32, 33] linear networks with more flexible initialization schemes [34, 35, 36]. It has also been applied to study the evolution of the NTK [37] and the influence of the scale on the transition between lazy and rich learning [12, 23]. In this work, we present novel exact solutions for a minimal model utilizing a mix of Bernoulli and Riccati equations to showcase a complex phase portrait of lazy and rich learning with separate alignment and fitting phases.

**Implicit bias.** An effective analysis approach to understanding the rich regime studies how the initialization influences the inductive bias at interpolation. The aim is to identify a function $Q(\theta)$ such that the network converges to a first-order KKT point minimizing $Q(\theta)$ among all possible interpolating solutions. Foundational work by Soudry et al. [38] pioneered this approach for a linear classifier trained with gradient descent, revealing a max margin bias. These findings have been extended to deep linear networks [39, 40, 41], homogeneous networks [42, 43, 44], and quasi-homogeneous networks [45]. A similar line of research expresses the learning dynamics of networks

---

[1]Xu and Ziyin [23] presented exact NTK dynamics for a linear model trained with one-dimensional data.

trained with mean squared error as a *mirror flow* for some potential $\Phi(\beta)$, such that the inductive bias can be expressed as a *Bregman divergence* [46]. This approach has been applied to diagonal linear networks, revealing an inductive bias that interpolates between $\ell^1$ and $\ell^2$ norms in the rich and lazy regimes respectively [14]. However, finding the potential $\Phi(\beta)$ is problem-specific and requires solving a second-order differential equation, which may not be solvable even in simple settings [47, 48]. Azulay et al. [9] extended this analysis to a time-warped mirror flow, enabling the study of a broader class of architectures. In this work we derive exact expressions for the inductive bias of our minimal model and extend the results in Azulay et al. [9] to wide and deep linear networks.

**Two-layer networks.** Two-layer, or single-hidden layer, piecewise linear networks have emerged as a key setting for advancing our understanding of the rich regime. Maennel et al. [49] observed that in training two-layer ReLU networks from small initializations, the first-layer weights concentrate along fixed directions determined by the training data, irrespective of network width. This phenomenon, termed *quantization*, has been proposed as a *simplicity bias* inherent to the rich regime, driving the network towards low-rank solutions when feasible. Subsequent studies have aimed to precisely elucidate this effect by introducing structural constraints on the training data [50, 51, 52, 53, 54, 55]. Across these analyses, a consistent observation is that the learning dynamics involve distinct phases: an initial alignment phase characterized by quantization, followed by fitting phases where the task is learned. All of these studies assumed a balanced (or nearly balanced) initialization between the first and second layer. In this study, we explore how unbalanced initializations influence the phases of learning, demonstrating that it can eliminate or augment the quantization effect.

**Infinite-width networks.** Many recent advancements in understanding the rich regime have come from studying how the initialization variance and layer-wise learning rates should scale in the infinite-width limit to ensure constant movement in the activations, gradients, and outputs. In this limit, analyzing dynamics becomes simpler in several respects: random variables concentrate and quantities will either vanish to zero, remain constant, or diverge to infinity [17]. A set of works used tools from statistical mechanics to provide analytic solutions for the rich population dynamics of two-layer nonlinear neural networks initialized according to the *mean field* parameterization [56, 57, 58, 59]. These ideas were extended to deeper networks through a *tensor program* framework, leading to the derivation of *maximal update parametrization* ($\mu$P) [16, 18]. The $\mu$P parameterization has also been derived through a self-consistent dynamical mean field theory [60] and a spectral scaling analysis [61]. In this study, we focus on finite-width neural networks, but discuss the connection between our work and these width-dependent parameterizations in Section 5.

## 3 A Minimal Model of Lazy and Rich Learning with Exact Solutions

Here we explore an illustrative setting simple enough to admit exact gradient flow dynamics, yet complex enough to showcase lazy and rich learning regimes. We study a two-layer linear network with a single hidden neuron defined by the map $f(x; \theta) = a w^\intercal x$ where $a \in \mathbb{R}$, $w \in \mathbb{R}^d$ are the parameters. We examine how the parameter initializations $a_0, w_0$ and the layer-wise learning rates $\eta_a, \eta_w$ influence the training trajectory in parameter space, function space (defined by the product $\beta = aw$), and the evolution of the the NTK matrix,

$$K = X \left( \eta_w a^2 \mathbf{I}_d + \eta_a w w^\intercal \right) X^\intercal. \quad (1)$$

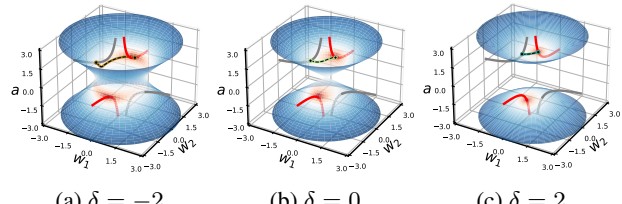

(a) $\delta = -2$     (b) $\delta = 0$     (c) $\delta = 2$

Figure 2: **Balance determines geometry of trajectory.** The quantity $\delta = \eta_w a^2 - \eta_a \|w\|^2$ is conserved through gradient flow, which constrains the trajectory to: (a) a one-sheeted hyperboloid for downstream initializations, (b) a double cone for balanced initializations, and (c) a two-sheeted hyperboloid for upstream initializations. Gradient flow dynamics for three different initializations $a_0, w_0$ with the same product $\beta_0 = a_0 w_0$ are shown. The minima manifold is shown in red and the manifold of equivalent $\beta_0$ initializations in gray. The surface is colored according to training loss, with blue representing higher loss and red representing lower loss.

Except for a measure zero set of initializations which converge to saddle points[2], all gradient flow trajectories will converge to a global minimum, determined by the normal equations $X^\intercal X a w = X^\intercal y$. However, even when $X^\intercal X$ is invertible such that the global minimum $\beta_*$ is unique, the rescaling symmetry between $a$ and $w$ results in a manifold

---

[2]The set of saddle points $\{(a, w)\}$ is the $d - 1$ dimensional subspace satisfying $a = 0$ and $w^\intercal X^\intercal y = 0$.

of minima in parameter space. The minima manifold is a one-dimensional hyperbola where $w \propto \beta_*$ and has two distinct branches for positive and negative $a$. The symmetry also imposes a constraint on the network's trajectory, maintaining the difference $\delta = \eta_w a^2 - \eta_a \|w\|^2 \in \mathbb{R}$ throughout training (see Appendix A.1 for details). This confines the parameter dynamics to the surface of a hyperboloid where the magnitude and sign of the conserved quantity determines the geometry, as shown in Fig. 2. An upstream initialization occurs when $\delta > 0$, a balanced initialization when $\delta = 0$, and a downstream initialization when $\delta < 0$.

**Deriving exact solutions in parameter space.** We initially assume[3] whitened input $X^\intercal X = \mathbf{I}_d$ such that the ordinary least squares solution is $\beta_* = X^\intercal y$, and the gradient flow dynamics simplify to $\dot{a} = \eta_a \left(w^\intercal \beta_* - a\|w\|^2\right), \dot{w} = \eta_w \left(a\beta_* - a^2 w\right)$. Notice that $w(t) \in \mathrm{span}(\{w_0, \beta_*\})$, and through training, $w$ aligns in direction to $\pm\beta_*$ depending on the basin of attraction[4] the parameters are initialized in. Therefore, we can monitor the dynamics by tracking the hyperbolic geometry between $a$ and $\|w(t)\|$ and the spherical angle between $w(t)$ and $\beta_*$. We study the variables $\mu = a\|w\|$, an invariant under the rescale symmetry, and $\phi = \frac{w^\intercal \beta_*}{\|w\|\|\beta_*\|}$, the cosine of the spherical angle. From these two scalar quantities $\mu(t), \phi(t)$ and the initialization $a_0, w_0$, we can determine the trajectory $a(t)$ and $w(t)$ in parameter space. The dynamics for $\mu, \phi$ are given by the coupled nonlinear ODEs,

$$\dot{\mu} = \sqrt{\delta^2 + 4\eta_a\eta_w\mu^2}\,(\phi\|\beta_*\| - \mu), \qquad \dot{\phi} = \frac{\eta_a\eta_w 2\mu\|\beta_*\|}{\sqrt{\delta^2 + 4\eta_a\eta_w\mu^2} - \delta}\left(1 - \phi^2\right). \qquad (2)$$

Amazingly, this system can be solved exactly, as discussed in Appendix A.2, and shown in Fig. 3. Without delving into the specifics, we can develop an intuitive understanding of the solutions by examining the influence of the relative scale $\delta$.

*Upstream.* When $\delta \gg 0$, the updates for both $\mu$ and $\phi$ diverge, but $\phi$ updates much more rapidly. We can decouple the dynamics of $\mu$ and $\phi$ by separation of their time scales and assume $\phi$ has reached its steady-state of $\pm 1$ before $\mu$ has updated. Then, the dynamics of $\mu$ is linear and proceeds exponentially to $\pm\|\beta_*\|$. This regime exhibits minimal kernel movement (see Fig. 3 (c)) because the kernel is dominated by the $\eta_w a^2 \mathbf{I}_d$ term, whereas it is mainly $w$ that updates.

*Balanced.* When $\delta = 0$, $\mu$ follows a Bernoulli differential equation driven by a time-dependent signal $\phi\|\beta_*\|$, and $\phi$ follows a Riccati equation evolving from an initial value to $\pm 1$ depending on the basin of attraction. For vanishing initialization $\|\beta_0\| \to 0$, the temporal dynamics of $\mu$ and $\phi$ decouple such that there are two phases of

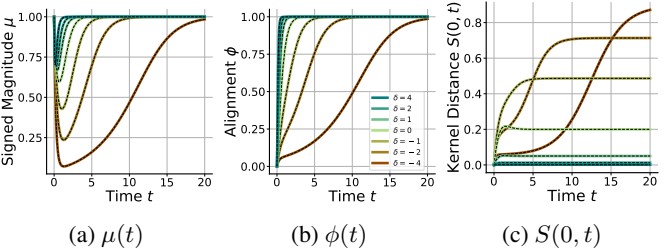

(a) $\mu(t)$      (b) $\phi(t)$      (c) $S(0,t)$

Figure 3: **Exact solutions for the single hidden neuron model.** Our theoretical predictions (black dashed lines) agree with gradient flow simulations (solid lines, color-coded based on $\delta$ values), shown here for three key metrics: $\mu$ (left), $\phi$ (middle), and $S(0,t)$ (right). Each metric starts at the same value for all $\delta$, but varying $\delta$ has a pronounced effect on the metric's dynamics. For upstream initializations ($\delta \gg 0$), $\mu$ changes only slightly, $\phi$ exponentially aligns, and $S$ remains near zero, indicative of the lazy regime. For balanced initializations ($\delta = 0$), both $\mu$ and $\phi$ change significantly and $S$ quickly moves away from zero, indicative of the rich regime. For downstream initializations ($\delta \ll 0$), $\mu$ quickly drops to zero, then $\mu$ and $\phi$ slowly climb back to one. Similarly, $S$ remains small before a sudden transition towards one, indicative of a delayed rich regime. See Appendix A.2 for further details.

learning: an initial alignment phase where $\phi \to \pm 1$, followed by a fitting phase where $\mu \to \pm\|\beta_*\|$. In the first phase, $w$ aligns to $\beta_*$ resulting in a rank-one update to the NTK, identical to the silent alignment effect described in Atanasov et al. [37]. In the second phase, the dynamics of $\mu$ simplify to the Bernoulli equation studied in Saxe et al. [10] and the kernel evolves solely in overall scale.

*Downstream.* When $\delta \ll 0$, the updates for $\mu$ diverge, while the updates for $\phi$ vanishes. In this regime the dynamics proceed by an initial fast phase where $\mu$ converges exponentially to its steady state of $\phi\|\beta_*\|$. Plugging this steady state into the dynamics of $\phi$ gives a Bernoulli differential equation

---

[3]We relax this assumption when considering the dynamics of $\beta$ in function space and their implicit bias.

[4]The basin is given by $\mathrm{sgn}(a_0)$ for $\delta \geq 0$ or $\mathrm{sgn}(w_0^\intercal\beta_* + \frac{a_0}{2}(\delta + \sqrt{\delta^2 + 4\|\beta_*\|^2}))$ for $\delta < 0$. See A.2.5.

$\dot{\phi} = \eta_a \eta_w \|\beta_*\|^2 |\delta|^{-1} \phi(1 - \phi^2)$. Due to the coefficient $|\delta|^{-1}$, the second alignment phase proceeds very slowly as $\phi$ approaches $\pm 1$, assuming $\phi, \mu \neq 0$, which is a saddle point. In this regime, the dynamics proceed by an initial lazy fitting phase, followed by a rich alignment phase, where the delay is determined by the magnitude of $\delta$.

**Identifying regimes of learning in function space.** Here we take an alternative route towards understanding the influence of the relative scale by directly examining the dynamics in function space, an analysis strategy we will generalize to broader setups in Sections 4 and 5. The network's function is determined by the product $\beta = aw$ and governed by the ODE,

$$\dot{\beta} = - \underbrace{\left(\eta_w a^2 \mathbf{I}_d + \eta_a ww^\mathsf{T}\right)}_{M} X^\mathsf{T} \rho, \quad (3)$$

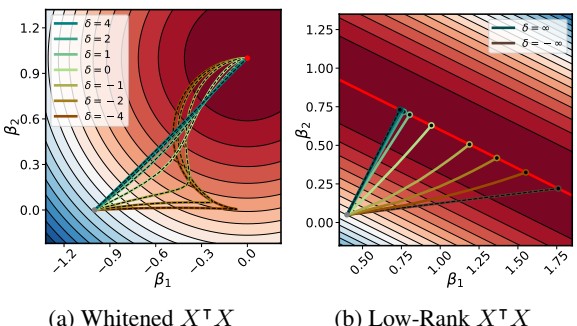

(a) Whitened $X^\mathsf{T}X$     (b) Low-Rank $X^\mathsf{T}X$

where $\rho = X\beta - y$ is the residual. These dynamics can be interpreted as preconditioned gradient flow on the loss in function space where the preconditioning matrix $M$ depends on time through its dependence on $a^2$ and $ww^\mathsf{T}$. Whenever $\|\beta\| \neq 0$, we can express $M$ directly in terms of $\beta$ and $\delta$ as

$$M = \frac{\kappa + \delta}{2} \mathbf{I}_d + \frac{\kappa - \delta}{2} \frac{\beta\beta^\mathsf{T}}{\|\beta\|^2}, \quad (4)$$

where $\kappa = \sqrt{\delta^2 + 4\eta_a\eta_w\|\beta\|^2}$ (see Appendix A.3 for a derivation). This establishes a *self-consistent* equation for the dynamics of $\beta$ regulated by $\delta$. Additionally,

Figure 4: **Balance modulates $\beta$ dynamics and implicit bias.** Here we show the dynamics of $\beta = aw$ with different values of $\delta$, but the same initial $\beta_0$. When $X^\mathsf{T}X$ is whitened (left), we can solve for the dynamics exactly using our expressions for $\mu, \phi$ (black dashed lines). Upstream initializations follow the trajectory of gradient flow on $\beta$, downstream initializations first move in the direction of $\beta_0$ before sweeping around towards $\beta_*$, and balanced initializations take an intermediate trajectory between these two. When $X^\mathsf{T}X$ is low-rank (right), then we can only predict the trajectories in the limit of $\delta = \pm\infty$. If the interpolating manifold is one-dimensional, then we can solve for the solution in terms of $\delta$ exactly (black dots). See Appendix A.4 for details.

notice that $M$ characterizes the NTK matrix Eq. (1). Thus, understanding the evolution of $M$ along the trajectory $\beta_0$ to $\beta_*$ offers a method to discern between lazy and rich learning. **Upstream.** When $\delta \gg 0$, $M \approx \delta\mathbf{I}_d$, and the dynamics of $\beta$ converge to the trajectory of linear regression trained by gradient flow. Along this trajectory the NTK matrix remains constant, confirming the dynamics are lazy. **Balanced.** When $\delta = 0$, $M = \sqrt{\eta_a\eta_w}\|\beta\|(\mathbf{I}_d + \frac{\beta\beta^\mathsf{T}}{\|\beta\|^2})$. Here the dynamics balance between following the lazy trajectory and attempting to fit the task by only changing in norm. As a result the NTK changes in both magnitude and direction through training, confirming the dynamics are rich. **Downstream.** When $\delta \ll 0$, $M \approx |\delta|\frac{\beta\beta^\mathsf{T}}{\|\beta\|^2}$, and $\beta$ follows a projected gradient descent trajectory, attempting to reach $\beta_*$ in the direction of $\beta_0$. Along this trajectory the NTK matrix doesn't evolve. However, if $\beta_0$ is not aligned to $\beta_*$, then at some point the dynamics of $\beta$ will slowly align. In this second alignment phase the NTK matrix will change, confirming the dynamics are initially lazy followed by a delayed rich phase. See Appendix A.3.1 for a derivation of the NTK dynamics $\dot{K}$.

**Determining the implicit bias via mirror flow.** So far we have considered whitened or full rank $X^\mathsf{T}X$, ensuring the existence of a unique least squares solution $\beta_*$. In this setting, $\delta$ influences the trajectory the model takes from $\beta_0$ to $\beta_*$, as shown in Fig. 4 (a). Now we consider low-rank $X^\mathsf{T}X$, such that there exist infinitely many interpolating solutions in function space. By studying the structure of $M$, we can characterize how $\delta$ determines the interpolating solution the dynamics converge to. Extending a time-warped mirror flow analysis strategy pioneered by Azulay et al. [9] to allow $\delta < 0$ (see Appendix A.4 for details), we prove the following theorem, which shows a tradeoff between reaching the minimum norm solution and preserving the direction of the initialization $\beta_0$.

**Theorem 3.1** (Extending Theorem 2 in Azulay et al. [9])**.** *For a single hidden neuron linear network, for any $\delta \in \mathbb{R}$, and initialization $\beta_0$ such that $\beta(t) \neq 0$ for all $t \geq 0$, if the gradient flow solution $\beta(\infty)$ satisfies $X\beta(\infty) = y$, then,*

$$\beta(\infty) = \arg\min_{\beta \in \mathbb{R}^d} \Psi_\delta(\beta) - \psi_\delta |\frac{\beta_0}{\|\beta_0\|}^\mathsf{T} \beta \quad \text{s.t.} \quad X\beta = y \quad (5)$$

*where $\Psi_\delta(\beta) = \frac{1}{3}\left(\sqrt{\delta^2 + 4\|\beta\|^2} - 2\delta\right)\sqrt{\sqrt{\delta^2 + 4\|\beta\|^2} + \delta}$ and $\psi_\delta = \sqrt{\sqrt{\delta^2 + 4\|\beta_0\|^2} - \delta}$.*

We observe that for vanishing initializations there is functionally no difference between the inductive bias of the upstream ($\delta \gg 0$) and balanced ($\delta = 0$) settings. However, in the downstream setting ($\delta \ll 0$), it is the second term preserving the direction of the initialization that dominates the inductive bias. This tradeoff in inductive bias as a function of $\delta$ is presented in Fig. 4 (b), where if the null space of $X^\mathsf{T} X$ is one-dimensional, we can solve for $\beta(\infty)$ in closed form (see Appendix A.4).

## 4 Wide and Deep Linear Networks

We now show how the analysis techniques used to study the influence of relative scale in the single-neuron setting can be applied to linear networks with multiple neurons, outputs, and layers.

**Wide linear networks.** We consider the dynamics of a two-layer linear network with $h$ hidden neurons and $c$ outputs, $f(x; \theta) = A^\mathsf{T} W x$, where $W \in \mathbb{R}^{h \times d}$ and $A \in \mathbb{R}^{h \times c}$. We assume $h \geq \min(d, c)$, such that this parameterization can represent all linear maps from $\mathbb{R}^d \to \mathbb{R}^c$. The rescaling symmetry between $A$ and $W$ implies the $h \times h$ matrix $\Delta = \eta_w A_0 A_0^\mathsf{T} - \eta_a W_0 W_0^\mathsf{T}$ is conserved throughout gradient flow [62]. Drawing insights from our analysis of the single-neuron scenario ($h = c = 1$), we consider the dynamics of $\beta = W^\mathsf{T} A \in \mathbb{R}^{d \times c}$,

$$\mathrm{vec}\left(\dot{\beta}\right) = -\underbrace{\left(\eta_w A^\mathsf{T} A \oplus \eta_a W^\mathsf{T} W\right)}_{M} \mathrm{vec}(X^\mathsf{T} X \beta - X^\mathsf{T} Y), \tag{6}$$

where $\mathrm{vec}(\cdot)$ denotes the vectorization operator and $\oplus$ denotes the Kronecker sum[5]. As in the single-neuron setting, we find that the dynamics of $\beta$ are preconditioned by a matrix $M$ that depends on quadratics of $A$ and $W$ and characterizes the NTK matrix $K = (\mathbf{I}_c \otimes X) M (\mathbf{I}_c \otimes X^\mathsf{T})$. We now show how $M$ can be expressed[6] in terms of the rank-1 matrices $\beta_k = w_k a_k^\mathsf{T} \in \mathbb{R}^{d \times c}$, which represent the contribution to $\beta$ of a single neuron with parameters $w_k, a_k$ and conserved quantity $\delta_k = \Delta_{kk}$.

**Theorem 4.1.** *Whenever $\|\beta_k\|_F \neq 0$ for all $k \in [h]$, the matrix $M$ can be expressed as the sum $M = \sum_{k=1}^h M_k$ over hidden neurons where $M_k$ is defined as,*

$$M_k = \left(\frac{\sqrt{\delta_k^2 + 4\eta_a \eta_w \|\beta_k\|_F^2} + \delta_k}{2}\right) \frac{\beta_k^\mathsf{T} \beta_k}{\|\beta_k\|_F^2} \oplus \left(\frac{\sqrt{\delta_k^2 + 4\eta_a \eta_w \|\beta_k\|_F^2} - \delta_k}{2}\right) \frac{\beta_k \beta_k^\mathsf{T}}{\|\beta_k\|_F^2}. \tag{7}$$

By studying the dependence of $M$ on the conserved quantities $\delta_k$ and the dimensions $d$, $h$ and $c$, we can determine the influence of the relative scale on the learning regime. When $\min(d, c) \leq h < \max(d, c)$, and assuming independent initializations for all $\beta_k$, then networks which narrow from input to output ($d > c$) enter the lazy regime when all $\delta_k \gg 0$, whereas networks which expand from input to output ($d < c$) do so when all $\delta_k \ll 0$. However, with opposite signs for $\delta_k$, and assuming all $\beta_k(0) \not\propto \beta_*$, these networks enter a *delayed rich regime*. As elaborated in Appendix B.1.5, this occurs because in these regimes a solution $\beta_*$ does not exist within the space spanned by $M$ at initialization. When $h \geq \max(d, c)$ all networks enter the lazy regime when all $\delta_k \gg 0$ or all $\delta_k \ll 0$. Conversely, as all $\delta_k \to 0$, all networks transition into the rich regime regardless of dimensions. While Theorem 4.1 offers valuable insight into the learning regimes in the limits of $\delta_k$, understanding the transition between regimes remains challenging. To achieve this, we aim to express $M$ in terms of $\beta$, rather than $\beta_k$, by introducing structure on the conserved quantities $\Delta$.

**Theorem 4.2.** *When $\Delta = \delta \mathbf{I}_h$ and $h = d$ if $\delta < 0$ or $h = c$ if $\delta > 0$, then the matrix $M$ can be expressed as $M = \sqrt{\eta_a \eta_w \beta^\mathsf{T} \beta + \frac{\delta^2}{4} \mathbf{I}_c} \otimes \mathbf{I}_d + \mathbf{I}_c \otimes \sqrt{\eta_a \eta_w \beta \beta^\mathsf{T} + \frac{\delta^2}{4} \mathbf{I}_d}$.*

From Theorem 4.2 the resulting dynamics of $\beta$ simplify to a self-consistent equation regulated by $\delta$,

$$\dot{\beta} = -X^\mathsf{T} P \sqrt{\eta_a \eta_w \beta^\mathsf{T} \beta + \frac{\delta^2}{4} \mathbf{I}_c} - \sqrt{\eta_a \eta_w \beta \beta^\mathsf{T} + \frac{\delta^2}{4} \mathbf{I}_d} X^\mathsf{T} P, \tag{8}$$

where $P = X\beta - Y$ is the residual. Under our isotropic assumption on the conserved quanitities $\Delta = \delta \mathbf{I}_h$, these dynamics are exact. Concurrent to our work, Tu et al. [63] finds that $\beta$ *approximately* follows these dynamics in the overparameterized setting $h \gg \max(d, c)$ under a Gaussian initialization $\mathcal{N}(0, \sigma^2)$ of the parameters where $\sigma^2 h$ is analogous to $\delta$.

---

[5]The Kronecker sum is defined for square matrices $C \in \mathbb{R}^{c \times c}$ and $D \in \mathbb{R}^{d \times d}$ as $C \oplus D = C \otimes \mathbf{I}_d + \mathbf{I}_c \otimes D$.
[6]When $h = c = 1$ we can recover Eq. (4) presented in the single-neuron setting directly from Eq. (7).

Equipped with a self-consistent equation for the dynamics of $\beta$ we now aim to interpret these dynamics as a mirror flow with a $\delta$-dependent potential. As presented in Theorem B.6, the dynamics of the singular values of $\beta$ can be described as a mirror flow with a *hyperbolic entropy* potential, which smoothly interpolates between an $\ell^1$ and $\ell^2$ penalty on the singular values for the rich ($\delta \to 0$) and lazy ($\delta \to \pm\infty$) regimes respectively. This potential was first identified as the inductive bias for diagonal linear networks by Woodworth et al. [14] and the same mirror flow on the singular values is derived from a different initialization choice in prior work by Varre et al. [64].

**Deep linear networks**. As presented in Theorem B.10, we generalize the inductive bias derived for rich two-layer linear networks by Azulay et al. [9] to deep linear networks. For a depth-$(L+1)$ linear network, $f(x;\theta) = A^\intercal \prod_{l=1}^{L} W_l x$, where $\beta = \prod_{l=1}^{L} W_l^\intercal A$, we find that the inductive bias of the rich regime is $Q(\beta) = (\frac{L+1}{L+2})\|\beta\|^{\frac{L+2}{L+1}} - \|\beta_0\|^{-\frac{L}{L+1}} \beta_0^\intercal \beta$. This inductive bias strikes a depth-dependent balance between attaining the minimum norm solution and preserving the initialization direction.

# 5 Piecewise Linear Networks

We now take a first step towards extending our analysis from linear networks to piecewise linear networks with activation functions of the form $\sigma(z) = \max(z, \gamma z)$. The input-output map of a piecewise linear network with $L$ hidden layers and $h$ hidden neurons per layer is comprised of potentially $O(h^{dL})$ convex *activation regions* [65]. Each region is defined by a unique *activation pattern* of the hidden neurons. The input-output map is linear within each region and continuous at the boundary between regions. Collectively, the activation regions form a 2-colorable[7] convex partition of input space, as shown in Fig. 5. We investigate how the relative scale influences the evolution of this partition and the linear maps within each region.

**Two-layer network.** We consider the dynamics of a two-layer piecewise linear network without biases, $f(x;\theta) = a^\intercal \sigma(Wx)$, where $W \in \mathbb{R}^{h \times d}$ and $a \in \mathbb{R}^h$. Following the approach in Section 4, we consider the contribution to the input-output map from a single hidden neuron $k \in [h]$ with parameters $w_k \in \mathbb{R}^d$, $a_k \in \mathbb{R}$ and conserved quantity $\delta_k = \eta_w a_k^2 - \eta_a \|w_k\|^2$ [62]. However, unlike the linear setting, the neuron's contribution to $f(x_i; \theta)$ is regulated by whether the input $x_i$ is in the neuron's *active halfspace*. Let $C \in \mathbb{R}^{h \times n}$ be the matrix with elements $c_{ki} = \sigma'(w_k^\intercal x_i)$, which determines the activation of the $k^{\text{th}}$ neuron for the $i^{\text{th}}$ data point. The dynamics of $\beta_k = a_k w_k$ are,

$$\dot{\beta}_k = -\underbrace{\left(\eta_w a_k^2 \mathbf{I}_d + \eta_a w_k w_k^\intercal\right)}_{M_k} \underbrace{\sum_{i=1}^{n} c_{ki} x_i (f(x_i; \theta) - y_i)}_{\xi_k}. \tag{9}$$

The matrix $M_k \in \mathbb{R}^{d \times d}$ is a preconditioning matrix on the dynamics, and when $\beta_k \neq 0$, it can be expressed in terms of $\beta_k$ and $\delta_k$. Unlike the linear setting, $\xi_k \in \mathbb{R}^d$ driving the dynamics is not shared for all neurons because of its dependence on $c_{ki}$. Additionally, the NTK matrix in this setting depends on $M_k$ and $C$, with elements $K_{ij} = \sum_{k=1}^{h} c_{ki} x_i^\intercal M_k x_j c_{kj}$. To examine the evolution of $K$, we consider a *signed spherical coordinate* transformation separating the dynamics of $\beta_k$ into its directional $\hat{\beta}_k = \text{sgn}(a_k) \frac{\beta_k}{\|\beta_k\|}$ and radial $\mu_k = \text{sgn}(a_k) \|\beta_k\|$ components, such that $\beta_k = \mu_k \hat{\beta}_k$. $\hat{\beta}_k$ determines the direction and orientation of the halfspace where the $k^{\text{th}}$ neuron is active, while $\mu_k$ determines the slope of the contribution in this halfspace. These coordinates evolve according to,

$$\dot{\mu}_k = -\sqrt{\delta_k^2 + 4\eta_a \eta_w \mu_k^2} \hat{\beta}_k^\intercal \xi_k, \qquad \dot{\hat{\beta}}_k = -\frac{\sqrt{\delta_k^2 + 4\eta_a \eta_w \mu_k^2} + \delta_k}{2\mu_k} \left(\mathbf{I}_d - \hat{\beta}_k \hat{\beta}_k^\intercal\right) \xi_k. \tag{10}$$

***Downstream.*** When $\delta_k \ll 0$, $M_k \approx |\delta_k| \hat{\beta}_k \hat{\beta}_k^\intercal$, and the dynamics are approximately $\partial_t \hat{\beta}_k = 0$ and $\partial_t \mu_k = -|\delta_k| \hat{\beta}_k^\intercal \xi_k$. Irrespective of $\xi_k$, $\hat{\beta}_k(t) = \hat{\beta}_k(0)$, which implies the overall partition map doesn't change (Fig. 5, bottom), nor the activation patterns $C$, nor $M_k$. Only $\mu_k$ changes to fit the data, while the NTK remains constant. If the number of hidden neurons is insufficient to fit the data, there is a delayed rich alignment phase where the kernel will change, with $|\delta_k|$ determining the delay.

***Balanced.*** When $\delta_k = 0$, $M_k = \sqrt{\eta_a \eta_w} |\mu_k| (\mathbf{I}_d + \hat{\beta}_k \hat{\beta}_k^\intercal)$, and the dynamics simplify to, $\partial_t \hat{\beta}_k = -\sqrt{\eta_a \eta_w} \text{sgn}(\mu_k)(\mathbf{I}_d - \hat{\beta}_k \hat{\beta}_k^\intercal)\xi_k$ and $\partial_t \mu_k = -2\sqrt{\eta_a \eta_w}|\mu_k|\hat{\beta}_k^\intercal \xi_k$. Here both the direction and magnitude of $\beta_k$ evolve, resulting in changes to the activation regions, patterns $C$, and NTK $K$. For vanishing initializations where $\|\beta_k(0)\| \to 0$ for all $k \in [h]$, we can decouple the dynamics into two

---

[7]To our knowledge, this property has not been recognized before. See Appendix C.1 for a formal statement.

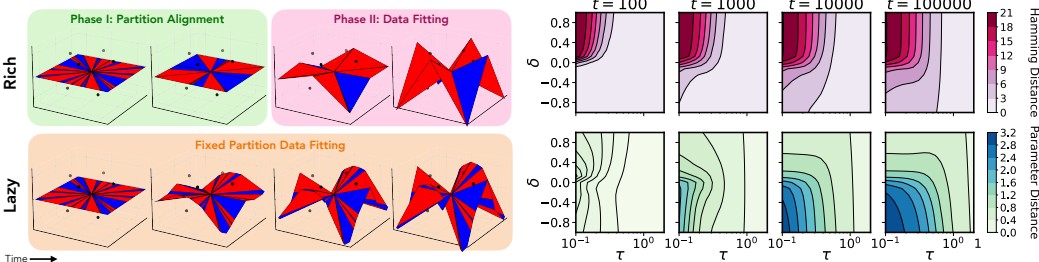

(a) Evolution of a ReLU network's input-output map    (b) Hamming and parameter distance over $\tau$-$\delta$ sweep

Figure 5: **Rapid feature learning is caused by large activation changes with minimal parameter movement.** (a) We show the surface of a two-layer ReLU network trained on an XOR-like task, starting with a near-zero input-output map, $f(x; \theta_0) \approx 0$. The surface consists of convex conic regions, each with a distinct activation pattern, colored by the parity of active neurons. A lazy initialization (*bottom*) maintains a fixed activation partition throughout training, reweighting the hidden neurons to fit the data. In contrast, a rich balanced or upstream initialization (*top*) features an initial alignment phase where the partition map changes rapidly while the input-output map remains close to zero, followed by a data-fitting phase. (b) We show the evolution of Hamming distance in activation patterns and parameter distance, relative to $t = 0$, as a function of *overall* and *relative* scales (same experiments as in Fig. 1(b)). *Rapid feature learning* occurs from a small-$\tau$ upstream initialization that promotes faster learning in early layers, driving a large change in Hamming distance, but a small change in parameter space. In contrast, small-$\tau$ downstream initializations require large parameter movement to fit the data in the delayed rich regime.

distinct phases of training (Fig. 5, top), analogous to the rich regime discussed in Section 3. *Phase I: Partition alignment.* At vanishing scale, the output $f(x; \theta_0) \approx 0$ for all input $x$, such that the vector driving the dynamics $\xi_k \approx - \sum_{i=1}^{n} c_{ki} x_i y_i$ is independent of the other hidden neurons. At the same time, the radial dynamics slow down relative to the directional dynamics, and the function's output will remain small as each neuron aligns to certain data-dependent fixed points, decoupled from the rest. Prior works have introduced structural constraints on the training data, such as orthogonally separable [50, 53, 54], pair-wise orthonormal [52], linearly separable and symmetric [51] or small angle [55], to analytically determine the fixed points of this alignment phase. *Phase II: Data fitting.* After enough time, the magnitudes of $\beta_k$ have grown such that we can no longer assume $f(x; \theta) \approx 0$ and thus the residual will depend on all $\beta_k$. In this phase, the radial dynamics dominate the learning driving the network to fit the data. However, it is possible for the directions to continue to change, and therefore some prior works have further decomposed this phase into multiple stages.

***Upstream.*** When $\delta_k \gg 0$, $M_k \approx \delta_k \mathbf{I}_d$, and the dynamics are approximately $\partial_t \hat{\beta}_k = -\delta_k \mu_k^{-1} (\mathbf{I}_d - \hat{\beta}_k \hat{\beta}_k^{\mathsf{T}}) \xi_k$ and $\partial_t \mu_k = -\delta_k \hat{\beta}_k^{\mathsf{T}} \xi_k$. Again, both the direction and magnitude of $\beta_k$ change. However, unlike the balanced setting, in this setting $M_k$ is independent of $\beta_k$ and stays constant through training. Yet, as $\beta_k$ change in direction, so can $C$, and thus the NTK. This setting is unique, because it is rich due to a changing activation pattern, but the dynamics do not move far in parameter space. Furthermore, unlike in the balanced scenario where scale adjusts the speed of radial dynamics, here it regulates the speed of directional dynamics, with vanishing initializations prompting an extremely fast alignment phase, as observed in Fig. 1 and in Fig. 5.

**Connections to infinite-width.** Our study of learning regimes in finite-width two-layer ReLU networks as a function of the overall and relative scale is consistent with existing infinite-width analysis of feature learning. For example, in Luo et al. [17] they consider a network $f(x) = \frac{1}{\alpha} \sum_{k=1}^{h} a_k \sigma(w_k^{\mathsf{T}} x)$ with weights initialized as $a_k \sim \mathcal{N}(0, \beta_a^2)$ and $w_k \sim \mathcal{N}(0, \beta_W^2 \mathbf{I}_d)$ as width $h \to \infty$. They obtain a phase diagram at infinite width capturing the dependence of learning regime on the overall function scale $\beta_a \beta_W / \alpha$ and the relative initialization scale $\beta_a / \beta_W$, each suitably normalized as a function of width. The resulting phase portrait is analogous to ours in Fig. 1 (b), where we use the conserved quantity $\delta$ rather than the relative scale $\beta_a / \beta_W$. Specifically, there is a lazy regime that includes the NTK parameterization, which is always achieved at large scale (as in the large-$\tau$ regions of Fig. 1 (b)), but is also achieved at small scale if the first layer variance is sufficiently larger than the second (as in the downstream initializations at small $\tau$ in Fig. 1 (b)). On the other side of the phase boundary is the infinite-width analog of rapid rich learning, where all neurons condense to a few directions. This is induced either at small function scale, or at larger scales if $\beta_a / \beta_W$ is sufficiently large, such that $W$ learns fast enough relative to $a$. The phase boundary, in

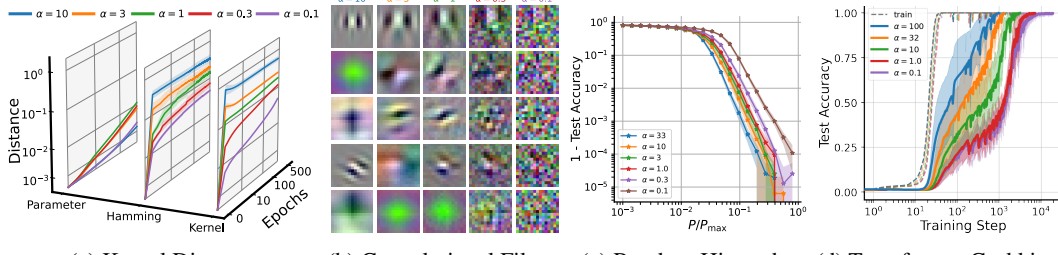

(a) Kernel Distance  (b) Convolutional Filters  (c) Random Hierarchy  (d) Transformer Grokking

Figure 6: **Impact of upstream initializations in practice.** Here we provide evidence that an upstream initialization (a) drives feature learning through changing activation patterns, (b) promotes interpretability of early layers in CNNs, (c) reduces the sample complexity of learning hierarchical data, and (d) decreases the time to grokking in modular arithmetic. In these experiments, we regulate the first layer's learning speed relative to the rest of the network by dividing its initialization by $\alpha$. For models without normalization layers, we also scale the last layer's initialization by $\alpha$ to preserve the input-output map. $\alpha = 1$ represents standard parameterization, while $\alpha \gg 1$ and $\alpha \ll 1$ correspond to upstream and downstream initializations, respectively. See Appendix D.3 for details.

turn, which exists only at infinite width, contains a range of parametrizations, including the mean-field parametrization. More broadly, across width-dependent parametrizations, the random initialization of weights induces a distribution over per-neuron conserved quantities. While the distinction between the NTK and the mean-field parametrizations has been extensively studied, both lead to the same distribution of per-neuron conserved quantities, which is zero in expectation with a non-vanishing variance. A more thorough study of what role the *distribution* of per-neuron conserved quantities plays in feature learning at finite-widths is left to future work.

**Unbalanced initializations in practice.** Our analysis shows that upstream initializations can drive rapid rich learning in nonlinear networks. Further experiments in Fig. 6 show that upstream initializations are relevant across various domains of deep learning: (a) Standard initializations see significant NTK evolution early in training [27]. We show the movement is linked to changes in activation patterns rather than large parameter shifts. Adjusting the initialization variance of the first and last layers can amplify or diminish this movement. (b) Filters in CNNs trained on image classification tasks often align with edge detectors [66]. We show that adjusting the learning speed of the first layer can enhance or degrade this alignment. (c) Deep learning models are believed to avoid the curse of dimensionality and learn with limited data by exploiting hierarchical structures in real-world tasks. Using the Random Hierarchy Model, introduced by Petrini et al. [67] as a framework for synthetic hierarchical tasks, we show that modifying the relative scale can decrease or increase the sample complexity of learning. (d) Networks trained on simple modular arithmetic tasks will suddenly generalize long after memorizing their training data [68]. This behavior, termed grokking, is thought to result from a transition from lazy to rich learning [69, 70, 71] and believed to be important towards understanding emergent phenomena [72]. We show that decreasing the variance of the embedding in a single-layer transformer ($< 6\%$ of all parameters) significantly reduces the time to grokking.

## 6 Conclusion

In this work, we derived exact solutions to a minimal model that can transition between lazy and rich learning to precisely elucidate how unbalanced layer-specific initialization variances and learning rates determine the degree of feature learning. We further extended our analysis to wide and deep linear networks and shallow piecewise linear networks. We find through theory and empirics that unbalanced initializations, which promote faster learning at earlier layers, can actually accelerate rich learning. **Limitations.** The primary limitation lies in the difficulty to extend our theory to deeper nonlinear networks. In contrast to linear networks, where additional symmetries simplify dynamics, nonlinear networks require consideration of the activation pattern's impact on subsequent layers. One potential solution involves leveraging the path framework used in Saxe et al. [73]. Another limitation is our omission of discretization and stochastic effects of SGD, which disrupt the conservation laws central to our study and introduce additional simplicity biases [74, 75, 76, 77]. **Future work.** Our theory encourages further investigation into unbalanced initializations to optimize efficient feature learning. Understanding how the *learning speed profile* across layers impacts feature learning, inductive biases, and generalization is an important direction for future work.

## Acknowledgments and Disclosure of Funding

We thank Francisco Acosta, Alex Atanasov, Yasaman Bahri, Abby Bertics, Blake Bordelon, Nan Cheng, Alex Infanger, Mason Kamb, Guillaume Lajoie, Nina Miolane, Cengiz Pehlevan, Ben Sorscher, Javan Tahir, Atsushi Yamamura for helpful discussions. D.K. thanks the Open Philanthropy AI Fellowship for support. S.G. thanks the James S. McDonnell and Simons Foundations, NTT Research, and an NSF CAREER Award for support. This research was supported in part by grant NSF PHY-1748958 to the Kavli Institute for Theoretical Physics (KITP).

## Author Contributions

This project originated from conversations between Daniel and Allan at the Kavli Institute for Theoretical Physics. Daniel, Allan, and Feng are primarily responsible for the single neuron analysis in Section 3. Daniel, Clem, Allan, and Feng are primarily responsible for the wide and deep linear analysis in Section 4. Daniel is primarily responsible for the nonlinear analysis in Section 5. Allan, Feng, and David are primarily responsible for the empirics in Fig. 1 and Fig. 6. Daniel is primarily responsible for writing the main sections. All authors contributed to the writing of the appendix and the polishing of the manuscript.

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

# A  Single-Neuron Linear Network

In this section, we provide a detailed analysis of the two-layer linear network with a single hidden neuron discussed in Section 3. The network is defined by the function $f(x; \theta) = a w^\mathsf{T} x$, where $a \in \mathbb{R}$ and $w \in \mathbb{R}^d$ are the parameters. We aim to understand the impact of the initializations $a_0, w_0$ and the layer-wise learning rates $\eta_a, \eta_w$ on the training trajectory in parameter space, function space (defined by the product $\beta = aw$), and the evolution of the Neural Tangent Kernel (NTK) matrix $K$:

$$K = X \left( \eta_w a^2 \mathbf{I}_d + \eta_a w w^\mathsf{T} \right) X^\mathsf{T}. \tag{11}$$

The gradient flow dynamics are governed by the following coupled ODEs:

$$\dot{a} = -\eta_a w^\mathsf{T} \left( X^\mathsf{T} X a w - X^\mathsf{T} y \right), \qquad\qquad a(0) = a_0, \tag{12}$$

$$\dot{w} = -\eta_w a \left( X^\mathsf{T} X a w - X^\mathsf{T} y \right), \qquad\qquad w(0) = w_0. \tag{13}$$

The global minima of this problem are determined by the normal equations $X^\mathsf{T} X a w = X^\mathsf{T} y$. Even when $X^\mathsf{T} X$ is invertible, yielding a unique global minimum in function space $\beta_* = (X^\mathsf{T} X)^{-1} X^\mathsf{T} y$, the symmetry between $a$ and $w$, permitting scaling transformations, $a \to a\alpha$ and $w \to w/\alpha$ for any $\alpha \neq 0$ without changing the product $aw$, results in a manifold of minima in parameter space. This minima manifold is a one-dimensional hyperbola where $aw = \beta_*$, with two distinct branches for positive and negative $a$. The set of saddle points $\{(a, w)\}$ forms a $(d-1)$-dimensional subspace satisfying $a = 0$ and $w^\mathsf{T} X^\mathsf{T} y = 0$. Except for a measure zero set of initializations that converge to the saddle points, all gradient flow trajectories will converge to a global minimum. In Appendix A.2.5, we detail the basin of attraction for each branch of the minima manifold and the $d$-dimensional surface of initializations that converge to saddle points, separating the two basins.

## A.1  Conserved quantity

The scaling symmetry between $a$ and $w$ results in a conserved quantity $\delta \in \mathbb{R}$ throughout training, as noted in many prior works [10, 62, 74], where

$$\delta = \eta_w a^2 - \eta_a \|w\|^2. \tag{14}$$

This can be easily verified by explicitly writing out the dynamics of $\delta$. Define $\rho = \left( X^\mathsf{T} X a w - X^\mathsf{T} y \right)$ for succinct notation, such that

$$\begin{aligned} \dot{\delta} &= 2\eta_w a \dot{a} - 2\eta_a w^\mathsf{T} \dot{w} \\ &= 2\eta_w a \left( -\eta_a w^\mathsf{T} \rho \right) - 2\eta_a w^\mathsf{T} \left( -\eta_w a \rho \right) \\ &= 0. \end{aligned}$$

The conserved quantity confines the parameter dynamics to the surface of a hyperboloid where the magnitude and sign of the conserved quantity determines the geometry, as shown in Fig. 2. A hyperboloid of the form $\sum_{i=1}^{k} x_i^2 - \sum_{i=k+1}^{n} x_i^2 = \alpha$, with $\alpha \geq 0$, exhibits varied topology and geometry based on $k$ and $\alpha$. It has two sheets when $k = 1$ and one sheet otherwise. Its geometry is primarily dictated by $\alpha$: as $\alpha$ tends to infinity, curvature decreases, while at $\alpha = 0$, a singularity occurs at the origin.

## A.2  Exact solutions

To derive exact dynamics we assume the input data is whitened such that $X^\mathsf{T} X = \mathbf{I}_d$ and $\beta_* = X^\mathsf{T} y$ such that $\beta_* \neq 0$. The dynamics of $a$ and $w$ can then be simplified as

$$\dot{a} = \eta_a \left( w^\mathsf{T} \beta_* - a \|w\|^2 \right), \qquad\qquad a(0) = a_0 \tag{15}$$

$$\dot{w} = \eta_w \left( a \beta_* - a^2 w \right), \qquad\qquad w(0) = w_0. \tag{16}$$

### A.2.1  Deriving the dynamics for $\mu$ and $\phi$

As discussion in Section 3 we study the variables $\mu = a\|w\|$, an invariant under the rescale symmetry, and $\phi = \frac{w^\mathsf{T} \beta_*}{\|w\| \|\beta_*\|}$, the cosine of the angle between $w$ and $\beta_*$. This change of variables can also be understood as a signed spherical decomposition of $\beta$: $\mu$ is the signed magnitude of $\beta$ and $\phi$ is the

cosine angle between $\beta$ and $\beta_*$. Through chain rule, we obtain the dynamics for $\mu$ and $\phi$, which can be expressed as

$$\dot{\mu} = \sqrt{\delta^2 + 4\eta_a\eta_w\mu^2}\left(\phi\|\beta_*\| - \mu\right), \qquad\qquad \mu(0) = a_0\|w_0\|, \qquad (17)$$

$$\dot{\phi} = \frac{\eta_a\eta_w 2\mu\|\beta_*\|}{\sqrt{\delta^2 + 4\eta_a\eta_w\mu^2} - \delta}\left(1 - \phi^2\right), \qquad\qquad \phi(0) = \frac{w_0^\mathsf{T}\beta_*}{\|w_0\|\|\beta_*\|}. \qquad (18)$$

We leave the derivation to the reader, but emphasize that a key simplification used is to express the sum $\eta_w a^2 + \eta_a\|w\|^2$ in terms of $\delta$,

$$\eta_w a^2 + \eta_a\|w\|^2 = \sqrt{\delta^2 + 4\eta_a\eta_w\mu^2}. \qquad (19)$$

Additionally, notice that $\eta_a$ and $\eta_w$ only appear in the dynamics for $\mu$ and $\phi$ as the product $\eta_a\eta_w$ or in the expression for $\delta$. If we were to define $\mu' = \sqrt{\eta_a\eta_w}\mu$ and $\beta'_* = \sqrt{\eta_a\eta_w}\beta_*$, then it is not hard to show that the product $\eta_a\eta_w$ is absorbed into the dynamics. Thus, without loss of generality we can assume the product $\eta_a\eta_w = 1$, resulting in the following coupled system of nonlinear ODEs,

$$\dot{\mu} = \sqrt{\delta^2 + 4\mu^2}\left(\phi\|\beta_*\| - \mu\right), \qquad\qquad \mu(0) = a_0\|w_0\| \qquad (20)$$

$$\dot{\phi} = \frac{2\mu\|\beta_*\|}{\sqrt{\delta^2 + 4\mu^2} - \delta}\left(1 - \phi^2\right), \qquad\qquad \phi(0) = \frac{w_0^\mathsf{T}\beta_*}{\|w_0\|\|\beta_*\|} \qquad (21)$$

We will now show how to solve this system of equations for $\mu$ and $\phi$. We will solve this system when $\delta = 0$, $\delta > 0$, and $\delta < 0$ separately. We will then in Appendix A.2.6 show a general treatment on how to obtain the individual coordinates of $a$ and $w$ from the solutions for $\mu$ and $\phi$.

### A.2.2 Balanced $\delta = 0$

When $\delta = 0$, the dynamics for $\mu, \phi$ are,

$$\dot{\mu} = \operatorname{sgn}(\mu)2\mu(\phi\|\beta_*\| - \mu), \qquad\qquad \mu(0) = a_0\|w_0\|, \qquad (22)$$

$$\dot{\phi} = \operatorname{sgn}(\mu)\|\beta_*\|(1 - \phi^2), \qquad\qquad \phi(0) = \frac{w_0^\mathsf{T}\beta_*}{\|w_0\|\|\beta_*\|}. \qquad (23)$$

First, we show that the sign of $\mu$ cannot change through training and $\operatorname{sgn}(\mu) = \operatorname{sgn}(a)$. Because $\delta = 0$, the dynamics of $a$ and $w$ are constrained to a double cone with a singularity at the origin $(a = 0, w = 0)$. This point is a saddle point of the dynamics, so the trajectory cannot pass through this point to move from one cone to the other. In other words, the cone where the dynamics are initialized on is the cone they remain on. Without loss of generality, we assume $a_0 > 0$, and solve the dynamics. The dynamics of $\mu$ is a Bernoulli differential equation driven by a time-dependent signal $\phi\|\beta_*\|$. The dynamics of $\phi$ is decoupled from $\mu$ and is in the form of a Riccati equation evolving from an initial value $\phi_0$ to 1, as we have assumed an initialization with positive $a_0$. This ODE is separable with the solution,

$$\phi(t) = \tanh\left(c_\phi + \|\beta_*\|t\right), \qquad (24)$$

where $c_\phi = \tanh^{-1}(\phi_0)$. Plugging this solution into the dynamics for $\mu$ gives a Bernoulli differential equation,

$$\dot{\mu} = 2\|\beta_*\|\tanh\left(c_\phi + \|\beta_*\|t\right)\mu - 2\mu^2, \qquad (25)$$

with the solution,

$$\mu(t) = \frac{2\cosh^2\left(c_\phi + \|\beta_*\|t\right)}{2\left(c_\phi + \|\beta_*\|t\right) + \sinh\left(2(c_\phi + \|\beta_*\|t)\right) + c_\mu}, \qquad (26)$$

where $c_\mu = 2\mu_0^{-1}\cosh^2(c_\phi) - (2c_\phi + \sinh(2c_\phi))$. Note, if $\phi_0 = -1$, then $\dot{\phi} = 0$, and the dynamics of $\mu$ will be driven to 0, which is a saddle point.

### A.2.3 Upstream $\delta > 0$

When $\delta > 0$, the dynamics are constrained to a hyperboloid composed of two identical sheets determined by the sign of $a_0$ (as shown in Fig. 2 (c)). Without loss of generality we assume $a_0 > 0$, which ensures $a(t) > 0$ for all $t \geq 0$. However, unlike in the balanced setting, the dynamics of $\mu$

and $\phi$ do not decouple, making it difficult to solve. Instead, we consider $\nu = \frac{w^\mathsf{T}\beta_*}{a}$, which evolves according to the Riccati equation,

$$\dot{\nu} = \|\beta_*\|^2 - \delta\nu - \nu^2, \qquad\qquad \nu(0) = \frac{w_0^\mathsf{T}\beta_*}{a_0}. \qquad (27)$$

The solution is given by,

$$\nu(t) = \frac{2R\nu_0 \cosh(Rt) + \left(2\|\beta_*\|^2 - \delta\nu_0\right)\sinh(Rt)}{2R\cosh(Rt) + (2\nu_0 + \delta)\sinh(Rt)}, \qquad (28)$$

where $R = \frac{1}{2}\sqrt{\delta^2 + 4\|\beta_*\|^2}$. The trajectory of $a(t)$ is given by the Bernoulli equation,

$$\dot{a} = a(\nu(t) + \delta - a^2), \qquad\qquad a(0) = a_0, \qquad (29)$$

which can be solved analytically using $\nu(t)$. For $a_0 > 0$, we have that

$$a(t) = 2e^{t\delta/2}\|\beta_*\|\sqrt{\delta}\left(\operatorname{sech}^2\left(Y(t)\right)\left[4e^{t\delta}\|\beta_*\|^2 - \frac{\left(\delta^2 + 4\|\beta_*\|^2\right)\left(\|\beta_*\|^2\left(\delta - a_0^2\right) + b_0^2\right)}{b_0^2 - a_0^2\|\beta_*\|^2 + a_0 b_0\delta}\right.\right.$$

$$\left.\left. - \delta e^{\delta t}\left(\delta\cosh\left(2Y(t)\right) - \sqrt{\delta^2 + 4\|\beta_*\|^2}\sinh\left(2Y(t)\right)\right)\right]\right)^{-1/2}$$

where $b_0 = w_0^\mathsf{T}\beta_*$, and $Y(t) = \frac{1}{2}\sqrt{\delta^2 + 4\|\beta_*\|^2}t + \operatorname{atanh}\left(\frac{\frac{2b_0}{a_0} + \delta}{\sqrt{\delta^2 + 4\|\beta_*\|^2}}\right)$. From the solutions for $\nu, a$, we can easily obtain dynamics for $\mu, \phi$.

### A.2.4  Downstream $\delta < 0$

When $\delta < 0$, the dynamics are constrained to a hyperboloid composed of a single sheet (as shown in Fig. 2 (a)). However, unlike in the upstream setting, $a$ may change sign. A zero-crossing in $a$ leads to a finite time blowup in $\nu$. Consequently, applying the approach used to solve for the dynamics in the upstream setting becomes more intricate. First we show the following lemma:

**Lemma A.1.** *If $a_0 \neq 0$ or $w_0^\mathsf{T}\beta_* \neq 0$, then $a(t)w(t)^\mathsf{T}\beta_* = 0$ has at most one solution for $t \geq 0$.*

*Proof.* Let $\omega(t) = w(t)^\mathsf{T}\beta_*$. The two-dimensional dynamics of $a(t)$ and $\omega(t)$ are given by,

$$\dot{a} = \omega - a(a^2 - \delta), \qquad (30)$$

$$\dot{\omega} = a\|\beta_*\|^2 - a^2\omega. \qquad (31)$$

Consider the orthant $O^+ = \{(a, \omega) | a > 0, \omega > 0\}$. The boundary $\partial O^+$ is formed by two orthogonal subspaces. On $\{(a, \omega) | a = 0, \omega \geq 0\}$, $\dot{a} \geq 0$. On $\{(a, \omega) | a \geq 0, \omega = 0\}$, $\dot{\omega} \geq 0$. Therefore, $O^+$ is a positively invariant set. Similarly, $O^- = \{(a, \omega) | a < 0, \omega < 0\}$ is a positively invariant set. On the boundary $\partial O^+ \cup \partial O^- = \{(a, \omega) | a\omega = 0\}$, the flow is contained only at the origin $a = 0, \omega = 0$, which represents all saddle points of the dynamics of $(a, w)$. By assumption, $(a, w)$ is not initialized at a saddle point, and thus the origin is not reachable for $t \geq 0$. As a result, the trajectory $(a(t), \omega(t))$ will at most intersect the boundary $\partial O_+ \cup \partial O_-$ once. $\square$

From Lemma A.1, we conclude that either $a$ crosses zero, $w^\mathsf{T}\beta_*$ crosses zero, or neither crosses zero. When $a$ doesn't cross zero, then $\nu$ is well-defined for $t \geq 0$, and our argument from Appendix A.2.3 still holds, leading to solutions for $\mu, \phi$. When $a$ does cross zero, instead of $\nu$, we consider $\upsilon = \frac{a}{w^\mathsf{T}\beta_*}$, the inverse of $\nu$. In this case, we know from Lemma A.1 that $w^\mathsf{T}\beta_*$ does not cross zero and thus $\upsilon$ is well-defined for $t \geq 0$ and evolves according to the Riccatti equation,

$$\dot{\upsilon} = 1 + \delta\upsilon - \|\beta_*\|^2\upsilon^2, \qquad\qquad \upsilon(0) = \frac{a_0}{w_0^\mathsf{T}\beta_*}. \qquad (32)$$

These dynamics have a solution similar to Eq. (28), which we leave to the reader. With $\upsilon(t)$, we can then solve for the dynamics of $w^\mathsf{T}\beta_*$. Let $\omega = w^\mathsf{T}\beta_*$, then $\omega$ evolves according to the Bernoulli equation,

$$\dot{\omega} = \omega\left(\upsilon\|\beta\|^2 - \upsilon^2\omega^2\right), \qquad\qquad \omega(0) = w(0)^\mathsf{T}\beta_*, \qquad (33)$$

which can be solved analytically using $\upsilon(t)$, analogous to the solution for $a(t)$ in Appendix A.2.3. From the solutions for $\upsilon, \omega$, we can easily obtain dynamics for $\mu, \phi$.

### A.2.5 Basins of attraction

From Lemma A.1 we know that $a$ can cross zero no more than once during its trajectory. Consequently, we can identify the basin of attraction by determining the conditions under which $a$ changes sign. This analysis is crucial because initial conditions leading to a sign change in $a$ correspond to scenarios where initial positive and negative values of $a_0$ are drawn towards the negative and positive branches of the minima manifold, respectively. From Eq. (28) we can immediately see that $a$ will change sign when the denominator vanishes. This can happen if $\sqrt{\delta^2 + 4\|\beta_*\|^2} < -2\nu_0 - \delta$. For $\delta < 0$, this is satisfied if $\nu_0 < \frac{1}{2}\left(-\delta - \sqrt{\delta^2 + 4\|\beta_*\|^2}\right)$, which gives the hyperplane $w_0^\mathsf{T}\beta_* + \frac{a_0}{2}\left(\delta + \sqrt{\delta^2 + 4\|\beta_*\|^2}\right) = 0$ that separates between initializations for which $a$ changes sign and initializations for which it does not (Fig. 7). Consequently, letting $S^+$ be the set of initializations attracted to the minimum manifold with $a > 0$, we have that:

$$
S^+ = \left\{ (w_0, a_0) \;\middle|\; \begin{array}{ll} a_0 > 0 & \text{if } \delta \geq 0 \\ w_0^\mathsf{T}\beta_* > -\frac{a_0}{2}\left(\delta + \sqrt{\delta^2 + 4\|\beta_*\|^2}\right) & \text{if } \delta < 0 \end{array} \right\}
\tag{34}
$$

where the bottom inequality means that $\beta_0$ is sufficiently aligned to $\beta_*$ in the case of $a_0 \geq 0$ or sufficiently misaligned in the case of $a_0 \leq 0$. We can similarly define the analogous $S^-$. An initialization on the separating hyperplane will converge to a saddle point where $w^\mathsf{T}\beta_* = a = 0$.

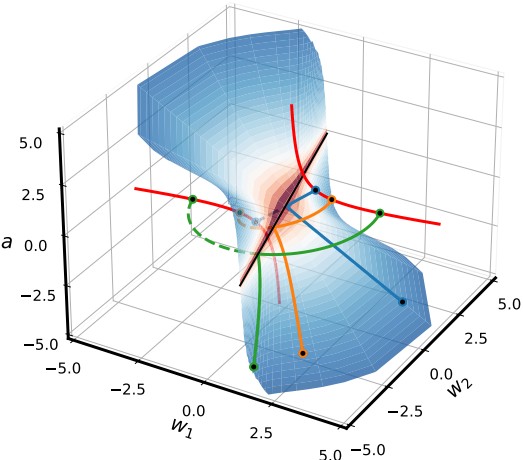

Figure 7: **Two basins of attraction.** For this model, parameter space is partitioned into two basins of attraction, one for the positive and negative branch of the minima manifold. The surface separating the basins of attraction is determined by the equation $w_0^\mathsf{T}\beta_* + \frac{a_0}{2}\left(\delta + \sqrt{\delta^2 + 4\|\beta_*\|^2}\right) = 0$. For a given $\delta$, this equation describes a hyperplane through the origin. However, a given $\delta$ can only be achieved on the surface of some hyperboloid. Thus, the separating surface is the union of the intersections of a hyperplane and a hyperboloid, both parameterized by $\delta$. This intersection is empty if $\delta > 0$. Initializations exactly on the separating surface will travel along the surface to a saddle point where $w^\mathsf{T}\beta_* = a = 0$.

### A.2.6 Recovering parameters $(a, w)$ from $(\mu, \phi)$

We now discuss how to recover the dynamics of the parameters $(a, w)$ from our solutions for $(\mu, \phi)$. We can recover $a$ and $\|w\|$ from $\mu$. Using Eq. (19) discussed previously, we can show

$$
a = \operatorname{sgn}(\mu)\sqrt{\frac{\sqrt{\delta^2 + 4\mu^2} + \delta}{2}}, \qquad \|w\| = \sqrt{\frac{\sqrt{\delta^2 + 4\mu^2} - \delta}{2}}.
\tag{35}
$$

We now discuss how to obtain the vector $w$ from $\phi$. The key observation, as discussed in Section 3, is that $w$ only moves in the span of $w_0$ and $\beta_*$. This means we can express $w(t)$ as

$$w(t) = c_1(t) \left( \frac{\beta_*}{\|\beta_*\|} \right) + c_2(t) \left( \frac{\left( \mathbf{I}_d - \frac{\beta_* \beta_*^\intercal}{\|\beta_*\|^2} \right) w_0}{\sqrt{\|w_0\|^2 - \left( \frac{\beta_*^\intercal w_0}{\|\beta_*\|} \right)^2}} \right) \tag{36}$$

where $c_1(t)$ is the coefficient in the direction of $\beta_*$ and $c_2(t)$ is the coefficient in the direction orthogonal to $\beta_*$ on the two-dimensional plane defined by $w_0$. From the definition of $\phi$ we can easily obtain the coefficients $c_1 = \|w\|\phi$ and $c_2 = \sqrt{\|w\|^2 - c_1^2}$. We always choose the positive square root for $c_2$, as $c_2(t) \geq 0$ for all $t$. See Appendix D.2 for experimental details of how we ran our simulations and a notebook generating these exact solutions.

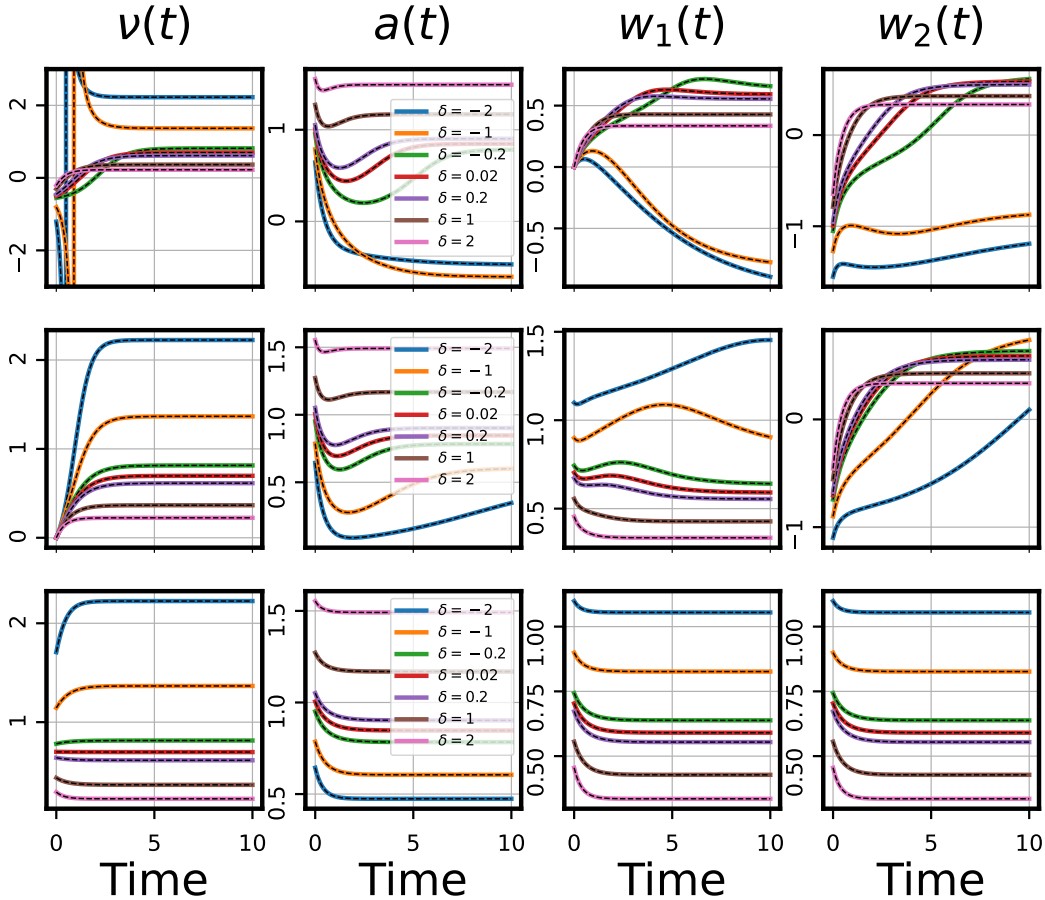

Figure 8: **Exact temporal dynamics of relevant variables in single-hidden neuron model.** Our theory recovers the time evolution under gradient flow of the quantities considered in this section, specifically $\nu$, $\varphi$, and $\zeta$, as well as the resulting dynamics of the model parameters $\{a, w_1, w_2\}$. The true $\beta_*$ is a unit vector pointing in $\pi/4$ direction; $\beta(0)$ is a unit vector pointing towards $3\pi/2$, $-\pi/4$, and $\pi/4$ directions, respectively, for each of the three rows. $\delta$ then defines how $a(0)$ and $\|w(0)\|$ are chosen for a particular $\beta(0)$ where by convention we choose $a(0) > 0$.

## A.3   Function space dynamics of $\beta$

The network's function is determined by the product $\beta = aw$ and governed by the ODE,

$$\dot{\beta} = a\dot{w} + \dot{a}w = - \underbrace{\left( \eta_w a^2 \mathbf{I}_d + \eta_a w w^\intercal \right)}_{M} \underbrace{\left( X^\intercal X \beta - X^\intercal y \right)}_{X^\intercal \rho}. \tag{37}$$

Notice, that the vector $X^\mathsf{T}\rho$ driving the dynamics of $\beta$ is the gradient of the loss with respect to $\beta$, $X^\mathsf{T}\rho = \nabla_\beta \mathcal{L}$. Thus, these dynamics can be interpreted as preconditioned gradient flow on the loss in $\beta$ space where the preconditioning matrix $M$ depends time through its dependence on $a^2$ and $ww^\mathsf{T}$. The matrix $M$ also characterizes the NTK matrix, $K = XMX^\mathsf{T}$. As discussed in Section 3, our goal is to understand the evolution of $M$ along a trajectory $\{\beta(t) \in \mathbb{R}^d \; : \; t \geq 0\}$ solving Eq. (37).

First, notice that by expanding $\|\beta\|^2 = a^2\|w\|^2$ in terms of the conservation law, we can show

$$a^2 = \frac{\sqrt{\delta^2 + 4\eta_a\eta_w\|\beta\|^2} + \delta}{2\eta_w}, \tag{38}$$

which is the unique positive solution of the quadratic expression $\eta_w a^4 - \delta a^2 - \eta_a\|\beta\|^2 = 0$. When $a^2 > 0$ we can use this solution and the outer product $\beta\beta^\mathsf{T} = a^2 ww^\mathsf{T}$ to solve for $ww^\mathsf{T}$ in terms of $\beta$,

$$ww^\mathsf{T} = \frac{\sqrt{\delta^2 + 4\eta_a\eta_w\|\beta\|^2} - \delta}{2\eta_a} \frac{\beta\beta^\mathsf{T}}{\|\beta\|^2}. \tag{39}$$

Plugging these expressions into $M$ gives

$$M = \frac{\sqrt{\delta^2 + 4\eta_a\eta_w\|\beta\|^2} + \delta}{2}\mathbf{I}_d + \frac{\sqrt{\delta^2 + 4\eta_a\eta_w\|\beta\|^2} - \delta}{2} \frac{\beta\beta^\mathsf{T}}{\|\beta\|^2}. \tag{40}$$

Thus, given any initialization $a_0, w_0$ such that $a(t)^2 \neq 0$ for all $t \geq 0$, we can express the dynamics of $\beta$ entirely in terms of $\beta$. This is true for all initializations with $\delta \geq 0$, except if initialized on the saddle point at the origin. It is also true for all initializations with $\delta < 0$ where the sign of $a$ does not switch signs. In the next section we will show how to interpret these trajectories as time-warped mirror flows for a potential that depends on $\delta$. As a means of keeping the analysis entirely in $\beta$ space, we will make the slightly more restrictive assumption to only study trajectories given any initialization $\beta_0$ such that $\|\beta(t)\| > 0$ for all $t \geq 0$.

Notice, that $\eta_a$ and $\eta_w$ only appear in the dynamics for $\beta$ as the product $\eta_a\eta_w$ or in the expression for $\delta$. By defining $\beta' = \sqrt{\eta_a\eta_w}\beta$ and $y' = \sqrt{\eta_a\eta_w}y$ and studying the dynamics of $\beta'$, we can absorb $\eta_a\eta_w$ into the $\beta$ terms in $M$ and the additional factor $\sqrt{\eta_a\eta_w}$ into the $\beta$ and $y$ terms in $\rho$. This transformation of $\beta$ and $y$ merely rescales $\beta$ space without changing the loss landscape or location of critical points. As a result, from here on we will, without loss of generality, study the dynamics of $\beta$ assuming $\eta_a\eta_w = 1$.

### A.3.1 Kernel dynamics

The dynamics of the NTK matrix $K = XMX^\mathsf{T}$ is determined by $\dot{M}$. From Eq. (4), which is derived in this section, we can write $\dot{M} = \frac{2\|\beta\|}{\kappa}(\mathbf{I}_d + \hat{\beta}\hat{\beta}^\mathsf{T})\partial_t\|\beta\| + \frac{\kappa - \delta}{2}\partial_t(\hat{\beta}\hat{\beta}^\mathsf{T})$ where $\hat{\beta} = \frac{\beta}{\|\beta\|}$. From this expression we see that the change in $M$ is driven by two terms, one that depends on the change in the magnitude of $\beta$ and another that depends on the change in the direction of $\beta$. As done in the main text, we consider $\delta \gg 0$, $\delta \ll 0$, and $\delta = 0$ to identify different regimes of learning. For $\delta \gg 0$, the coefficients in front of both terms vanish, and thus, irrespective of the trajectory taken from $\beta(0)$ to $\beta_*$, the change in the NTK is vanishing, indicative of a lazy regime. For $\delta \ll 0$, the coefficient for the first term vanishes, while the coefficient on the second term diverges. Here, the change in the NTK is driven solely by the change in the direction of $\beta$. This is why for large negative delta we observe a delayed rich regime, where the eventual alignment of $\beta$ to $\beta_*$ leads to a dramatic change in the kernel. When $\delta = 0$, the coefficients for both terms are roughly of the same order, and thus changes in both the magnitude and direction of $\beta$ contribute to a change in the kernel, indicative of a rich regime.

### A.4 Deriving the inductive bias

Until now, we have primarily considered that $X^\mathsf{T}X$ is either whitened or full rank, ensuring the existence of a unique least squares solution $\beta_*$. In this setting, $\delta$ influences the trajectory the model takes from initialization to convergence, but all models eventually converge to the same point, as shown in Fig. 4. Now we consider the over-parameterized setting where we have more features $d$ than observations $n$ such that $X^\mathsf{T}X$ is low-rank and there exists infinitely many interpolating solutions in function space. By studying the structure of $M$ we can characterize or even predict how $\delta$ determines which interpolating solution the dynamics converge to among all possible interpolating solutions. To do this we will extend a time-warped mirror flow analysis strategy pioneered by Azulay et al. [9].

### A.4.1 Overview of time-warped mirror flow analysis

Here we recap the standard analysis for determining the implicit bias of a linear network through mirror flow. As first introduced in Gunasekar et al. [46], if the learning dynamics of the predictor $\beta$ can be expressed as a *mirror flow* for some strictly convex potential $\Phi_\alpha(\beta)$,

$$\dot{\beta} = - \left( \nabla^2 \Phi_\alpha(\beta) \right)^{-1} X^\mathsf{T} \rho, \tag{41}$$

where $\rho = (X\beta - y)$ is the residual, then the limiting solution of the dynamics is determined by the constrained optimization problem,

$$\beta(\infty) = \arg\min_{\beta \in \mathbb{R}^d} D_{\Phi_\alpha}(\beta, \beta(0)) \quad \text{s.t.} \quad X\beta = y, \tag{42}$$

where $D_{\Phi_\alpha}(p, q) = \Phi_\alpha(p) - \Phi_\alpha(q) - \langle \nabla \Phi_\alpha(q), p - q \rangle$ is the Bregman divergence defined with $\Phi_\alpha$. To understand the relationship between mirror flow Eq. (41) and the optimization problem Eq. (42), we consider an equivalent constrained optimization problem

$$\beta(\infty) = \arg\min_{\beta \in \mathbb{R}^d} Q(\beta) \quad \text{s.t.} \quad X\beta = y, \tag{43}$$

where $Q(\beta) = \Phi_\alpha(\beta) - \nabla \Phi_\alpha(\beta(0))^\mathsf{T} \beta$, which is often referred to as the *implicit bias*. $Q(\beta)$ is strictly convex, and thus it is sufficient to show that $\beta(\infty)$ is a first order KKT point of the constrained optimization (43). This is true iff there exists $\nu \in \mathbb{R}^n$ such that $\nabla Q(\beta(\infty)) = X^\mathsf{T} \nu$. The goal is to derive $\nu$ from the mirror flow Eq. (41). Notice, we can rewrite Eq. (41) as, $(\nabla \Phi_\alpha(\beta))^{\cdot} = -X^\mathsf{T} \rho$, which integrated over time gives

$$\nabla \Phi_\alpha(\beta(\infty)) - \nabla \Phi_\alpha(\beta(0)) = -X^\mathsf{T} \int_0^\infty \rho(t)dt. \tag{44}$$

The LHS is $\nabla Q(\beta(\infty))$. Thus, by defining $\nu = \int_0^\infty \rho(t)dt$, which assumes the residual decays fast enough such that this is well defined, then we have shown the desired KKT condition. Crucial to this analysis is that there exists a solution to the second-order differential equation

$$\nabla^2 \Phi_\alpha(\beta) = \left( \nabla_\theta \beta \nabla_\theta \beta^\mathsf{T} \right)^{-1}, \tag{45}$$

which even for extremely simple Jacobian maps may not be true [47]. Azulay et al. [9] showed that if there exists a smooth positive function $g(\beta) : \mathbb{R}^d \to (0, \infty)$ such that the ODE,

$$\nabla^2 \Phi_\alpha(\beta) = g(\beta) \left( \nabla_\theta \beta \nabla_\theta \beta^\mathsf{T} \right)^{-1}, \tag{46}$$

has a solution, then the previous interpretation holds for $\Phi_\alpha(\beta)$ with $\nu = \int_0^\infty g(\beta(t'))\rho(t')dt'$. As before, it is crucial that this integral exists and is finite. Azulay et al. [9] further explained that this scalar function $g(\beta)$ can be considered as warping time $\tau(t) = \int_0^t g(\beta(t'))dt'$ on the trajectory taken in predictor space $\beta(\tau(t))$. So long as this warped time doesn't "stall out", that is we require that $\tau(\infty) = \infty$, then this will not change the interpolating solution.

### A.4.2 Applying time-warped mirror flow analysis

Here show how to apply the time-warped mirror flow analysis to the dynamics of $\beta$ derived in Appendix A.3 where $\nabla_\theta \beta \nabla_\theta \beta^\mathsf{T} = M$. We will only consider initializations $\beta_0$ such that $\|\beta(t)\| > 0$ for all $t \geq 0$, such that $M$ can be expressed as

$$M = \frac{\sqrt{\delta^2 + 4\|\beta\|^2} + \delta}{2} I_d + \frac{\sqrt{\delta^2 + 4\|\beta\|^2} - \delta}{2} \frac{\beta\beta^\mathsf{T}}{\|\beta\|^2}. \tag{47}$$

**Computing $M^{-1}$.** Whenever $\|\beta\| > 0$, then $M$ is a positive definite matrix with a unique inverse that can be derived using the Sherman–Morrison formula, $(A + uv^\mathsf{T})^{-1} = A^{-1} - \frac{A^{-1}uv^\mathsf{T}A^{-1}}{1 + u^\mathsf{T}A^{-1}v}$. Here we can define $A$, $u$, and $v$ as

$$A = \left( \frac{\sqrt{\delta^2 + 4\|\beta\|^2} + \delta}{2} \right) I_d, \quad u = \left( \frac{\sqrt{\delta^2 + 4\|\beta\|^2} - \delta}{2\|\beta\|^2} \right) \beta, \quad v = \beta \tag{48}$$

First notice the following simplification, $u^\mathsf{T} A^{-1} v = \frac{\sqrt{\delta^2 + 4\|\beta\|^2} - \delta}{\sqrt{\delta^2 + 4\|\beta\|^2} + \delta}$. After some algebra, $M^{-1}$ is

$$M^{-1} = \left( \frac{2}{\sqrt{\delta^2 + 4\|\beta\|^2} + \delta} \right) I_d - \left( \frac{\frac{\sqrt{\delta^2 + 4\|\beta\|^2} - \delta}{\sqrt{\delta^2 + 4\|\beta\|^2} + \delta}}{\|\beta\|^2 \sqrt{\delta^2 + 4\|\beta\|^2}} \right) \beta \beta^\mathsf{T} \tag{49}$$

To make notation simpler we will define the following two scalar functions,

$$f_\delta(x) = \frac{2}{\sqrt{\delta^2 + 4x} + \delta}, \qquad h_\delta(x) = \frac{\sqrt{\delta^2 + 4x} - \delta}{x \sqrt{\delta^2 + 4x} \left( \sqrt{\delta^2 + 4x} + \delta \right)}, \tag{50}$$

such that we can express $M^{-1} = f_\delta \left( \|\beta\|^2 \right) I_d - h_\delta \left( \|\beta\|^2 \right) \beta \beta^\mathsf{T}$.

**Proving $M^{-1}$ is not a Hessian map.** If $M^{-1}$ is the Hessian of some potential, then we can show that the dynamics of $\beta$ are a mirror flow. However, from our expression for $M^{-1}$ we can actually prove that it is *not* a Hessian map. As discussed in Gunasekar et al. [47], a symmetric matrix $H(\beta)$ is the Hessian of some potential $\Phi(\beta)$ if and only if it satisfies the condition,

$$\forall \beta \in \mathbb{R}^m, \quad \forall i, j, k \in [m] \quad \frac{\partial H_{ij}(\beta)}{\partial \beta_k} = \frac{\partial H_{ik}(\beta)}{\partial \beta_j}. \tag{51}$$

We will use this property to show $M^{-1}$ is not a Hessian map. First, notice this condition is trivially true when $i = j = k$. Second, notice that for all $i \neq j \neq k$,

$$\frac{\partial M_{ij}^{-1}}{\partial \beta_k} = \frac{\partial M_{ik}^{-1}}{\partial \beta_j} = -2 \nabla h_\delta \left( \|\beta\|^2 \right) \beta_i \beta_j \beta_k \tag{52}$$

Thus, $M^{-1}$ is a Hessian map if and only if for all $i \neq j$, $\frac{\partial M_{ii}^{-1}}{\partial \beta_j} = \frac{\partial M_{ij}^{-1}}{\partial \beta_i}$. Using our expression for $M^{-1}$, the LHS is

$$\frac{\partial M_{ii}^{-1}}{\partial \beta_j} = 2 \nabla f_\delta \left( \|\beta\|^2 \right) \beta_j - 2 \nabla h_\delta \left( \|\beta\|^2 \right) \beta_j \beta_i^2 \tag{53}$$

while the RHS is

$$\frac{\partial M_{ij}^{-1}}{\partial \beta_i} = -h_\delta \left( \|\beta\|^2 \right) \beta_j - 2 \nabla h_\delta \left( \|\beta\|^2 \right) \beta_j \beta_i^2 \tag{54}$$

Thus, $M^{-1}$ is a Hessian map if and only if $2 \nabla f_\delta(x) + h_\delta(x) = 0$. Plugging in our definitions of $f_\delta(x)$ and $h_\delta(x)$ we find

$$2 \nabla f_\delta(x) + h_\delta(x) = \frac{-4}{\sqrt{\delta^2 + 4x}(\sqrt{\delta^2 + 4x} + \delta)^2}, \tag{55}$$

which does not equal zero and thus $M^{-1}$ is not a Hessian map.

**Finding a scalar function $g_\delta(x)$ such that $g_\delta(\|\beta\|^2) M^{-1}$ is a Hessian map.** While we have shown that $M^{-1}$ is not a Hessian map, it is very close to a Hessian map. Here we will show that there exists a scalar function $g_\delta(x)$ such that $g_\delta \left( \|\beta\|^2 \right) M^{-1}$ is a Hessian map. For any $g_\delta(x)$ can define $g_\delta \left( \|\beta\|^2 \right) M^{-1}$ in terms of two new functions $\tilde{f}_\delta(x)$ and $\tilde{h}_\delta(x)$ evaluated at $x = \|\beta\|^2$,

$$g_\delta \left( \|\beta\|^2 \right) M^{-1} = \underbrace{g_\delta \left( \|\beta\|^2 \right) f_\delta \left( \|\beta\|^2 \right)}_{\tilde{f}_\delta(\|\beta\|^2)} I_d - \underbrace{g_\delta \left( \|\beta\|^2 \right) h_\delta \left( \|\beta\|^2 \right)}_{\tilde{h}_\delta(\|\beta\|^2)} \beta \beta^\mathsf{T}. \tag{56}$$

Thus, as derived in the previous section, we get the analogous condition on $\tilde{f}_\delta(x)$ and $\tilde{h}_\delta(x)$ for $g_\delta \left( \|\beta\|^2 \right) M^{-1}$ to be a Hessian map,

$$2 \underbrace{\left( \nabla g_\delta(x) f_\delta(x) + g(x) \nabla f_\delta(x) \right)}_{\nabla \tilde{f}_\delta(x)} + \underbrace{g_\delta(x) h_\delta(x)}_{\tilde{h}_\delta(x)} = 0 \tag{57}$$

Rearranging terms we find that $g_\delta(x)$ must solve the ODE

$$\nabla g_\delta(x) = - \left( 2 f_\delta(x) \right)^{-1} \left( 2 \nabla f_\delta(x) + h_\delta(x) \right) g_\delta(x). \tag{58}$$

Using our previous expressions (Eq. (50) and Eq. (55)) we find

$$- (2f_\delta(x))^{-1} (2\nabla f_\delta(x) + h_\delta(x)) = \frac{1}{\sqrt{\delta^2 + 4x}(\sqrt{\delta^2 + 4x} + \delta)}, \tag{59}$$

which implies $g_\delta(x)$ solves the differential equation, $\nabla g_\delta(x) = \frac{g_\delta(x)}{\sqrt{\delta^2+4x}(\sqrt{\delta^2+4x}+\delta)}$. The solution is $g_\delta(x) = c\sqrt{\sqrt{\delta^2 + 4x} + \delta}$, where $c \in \mathbb{R}$ is a constant. Let $c = 1$. Plugging in our expressions for $g_\delta\left(\|\beta\|^2\right)$, $f_\delta\left(\|\beta\|^2\right)$, $h_\delta\left(\|\beta\|^2\right)$, we get that

$$g_\delta\left(\|\beta\|^2\right) M^{-1} = \left(\frac{2}{\sqrt{\sqrt{\delta^2 + 4\|\beta\|^2} + \delta}}\right) I_d - \left(\frac{\frac{\sqrt{\delta^2 + 4\|\beta\|^2} - \delta}{\sqrt{\sqrt{\delta^2 + 4\|\beta\|^2} + \delta}}}{\|\beta\|^2 \sqrt{\delta^2 + 4\|\beta\|^2}}\right) \beta\beta^\mathsf{T} \tag{60}$$

is a Hessian map for some unknown potential $\Phi_\delta(\beta)$.

**Solving for the potential $\Phi_\delta(\beta)$.** Take the ansatz that there exists some function scalar $q(x)$ such that $\Phi_\delta(\beta) = q_\delta(\|\beta\|) + c_\delta$ where $c_\delta$ is a constant such that $\Phi_\delta(\beta) > 0$ for all $\beta \neq 0$ and $\Phi_\delta(0) = 0$. The Hessian of this ansatz takes the form,

$$\nabla^2 \Phi_\delta(\beta) = \left(\frac{\nabla q(\|\beta\|)}{\|\beta\|}\right) I_d - \left(\frac{\nabla q(\|\beta\|)}{\|\beta\|^3} - \frac{\nabla^2 q(\|\beta\|)}{\|\beta\|^2}\right) \beta\beta^\mathsf{T}. \tag{61}$$

Equating terms from our expression for $g_\delta\left(\|\beta\|^2\right) M^{-1}$ (equation 60) we get the expression for $\nabla q(\|\beta\|)$

$$\nabla q(\|\beta\|) = \frac{2\|\beta\|}{\sqrt{\sqrt{\delta^2 + 4\|\beta\|^2} + \delta}}, \tag{62}$$

which plugged into the second term gives the expression for $\nabla^2 q(\|\beta\|)$,

$$\nabla^2 q(\|\beta\|) = \frac{2}{\sqrt{\sqrt{\delta^2 + 4\|\beta\|^2} + \delta}} - \left(\frac{\frac{\sqrt{\delta^2 + 4\|\beta\|^2} - \delta}{\sqrt{\sqrt{\delta^2 + 4\|\beta\|^2} + \delta}}}{\sqrt{\delta^2 + 4\|\beta\|^2}}\right) = \frac{\sqrt{\sqrt{\delta^2 + 4\|\beta\|^2} + \delta}}{\sqrt{\delta^2 + 4\|\beta\|^2}}. \tag{63}$$

We now look for a function $q(x)$ such that both these conditions (Eq. (62) and Eq. (63)) are true. Consider the following function and its derivatives,

$$q(x) = \frac{1}{3}\left(\sqrt{\delta^2 + 4x^2} - 2\delta\right)\sqrt{\sqrt{\delta^2 + 4x^2} + \delta} \tag{64}$$

$$\nabla q(x) = \frac{2x}{\sqrt{\sqrt{\delta^2 + 4x^2} + \delta}} \tag{65}$$

$$\nabla^2 q(x) = \frac{\sqrt{\sqrt{\delta^2 + 4x^2} + \delta}}{\sqrt{\delta^2 + 4x^2}} \tag{66}$$

Letting $x = \|\beta\|$ notice $\nabla q(\|\beta\|)$ and $\nabla^2 q(\|\beta\|)$ satisfies the previous conditions. Furthermore, $\nabla^2 q(x) > 0$ for all $\delta$ as long as $x \neq 0$ and thus $q(x)$ is a convex function which achieves its minimum at $x = 0$. Thus, the constant $c_\delta = -q(0)$ is

$$c_\delta = \begin{cases} 0 & \text{if } \delta \leq 0 \\ \frac{\sqrt{2}|\delta|^{\frac{3}{2}}}{3} & \text{if } \delta > 0 \end{cases} = \max\left(0, \mathrm{sgn}(\delta)\frac{\sqrt{2}|\delta|^{\frac{3}{2}}}{3}\right), \tag{67}$$

and the potential $\Phi_\delta(\beta)$ is

$$\Phi_\delta(\beta) = \frac{1}{3}\left(\sqrt{\delta^2 + 4\|\beta\|^2} - 2\delta\right)\sqrt{\sqrt{\delta^2 + 4\|\beta\|^2} + \delta} + \max\left(0, \mathrm{sgn}(\delta)\frac{\sqrt{2}|\delta|^{\frac{3}{2}}}{3}\right). \tag{68}$$

Finally, putting it all together, we can express the inductive bias as in Theorem 3.1.

### A.4.3 Connection to Theorem 2 in Azulay et al. [9]

We discuss how Theorem 3.1 connects to Theorem 2 in Azulay et al. [9], which we rewrite:

**Theorem A.2** (Theorem 2 from Azulay et al. [9]). *For a depth 2 fully connected network with a single hidden neuron ($h = 1$), any $\delta \geq 0$, and initialization $\beta_0$ such that $\beta_0 \neq 0$, if the gradient flow solution $\beta(\infty)$ satisfies $X\beta(\infty) = y$, then,*

$$\beta(\infty) = \arg\min_{\beta \in \mathbb{R}^d} q_\delta(\|\beta\|) + z^\mathsf{T}\beta \quad \text{s.t.} \quad X\beta = y \tag{69}$$

*where $q_\delta(x) = \dfrac{\left(x^2 - \frac{\delta}{2}\left(\frac{\delta}{2} + \sqrt{x^2 + \frac{\delta^2}{4}}\right)\right)\sqrt{\sqrt{x^2 + \frac{\delta^2}{4}} - \frac{\delta}{2}}}{x}$ and $z = -\frac{3}{2}\sqrt{\sqrt{\|\beta_0\|^2 + \frac{\delta^2}{4}} - \frac{\delta}{2}}\frac{\beta_0}{\|\beta_0\|}$.*

The most striking difference is in the expressions for the inductive bias. Azulay et al. [9] take an alternative route towards deriving the inductive bias by inverting $M$ in terms of the original parameters $a$ and $w$ and then simplifying $M^{-1}$ in terms of $\beta$, which results in quite a different expression for their inductive bias. However, they are actually functionally equivalent. It requires a bit of algebra, but one can show that

$$\Phi_\delta(\beta) = \frac{2\sqrt{2}}{3}q_\delta(\|\beta\|) + c_\delta. \tag{70}$$

Another important distinction between our two theorems lies in the assumptions we make. Azulay et al. [9] consider only initializations such that $\delta \geq 0$ and $\beta_0 \neq 0$. We make a less restrictive assumption by considering initializations $\beta_0$ such that $\|\beta(t)\| > 0$ for all $t \geq 0$, which allows for both positive and negative $\delta$. Except for a measure zero set of initializations, all initializations considered by Azulay et al. [9] also satisfy our assumptions. In both cases, our assumptions ensure that $M$ is invertible for the entire trajectory from initialization to interpolating solution. However, it is worth considering whether the theorems would hold even when there exists a point on the trajectory where $M$ is low-rank. As discussed in Appendix A.3, this can only happen for an initialization with $\delta < 0$ and where the sign of $a$ changes. Only at the point where $a(t) = 0$ does $M$ become low-rank. A similar challenge arose in this setting when deriving the exact solutions presented in Appendix A.2.4. We were able to circumvent the issue in part by introducing Lemma A.1 proving that this sign change could only happen at most once given any initialization. This lemma was based on the setting with whitened input, but a similar statement likely holds for the general setting. If this were the case, we could define $M$ at this unique point on the trajectory in terms of the limit of $M$ as it approached this point. This could potentially allow us to extend the time-warped mirror flow analysis to all initializations such that $\|\beta_0\| > 0$.

### A.4.4 Exact solution when interpolating manifold is one-dimensional

When the null space of $X^\mathsf{T}X$ is one-dimensional, the constrained optimization problems in Theorem 3.1 and Theorem A.2 have an exact analytic solution. In this case we can parameterize all interpolating solutions $\beta$ with a single scalar $\alpha \in \mathbb{R}$ such that $\beta = \beta_* + \alpha v$ where $X^\mathsf{T}Xv = 0$ and $\|v\| = 1$. Using this description of $\beta$, we can then differentiate the inductive bias with respect to $\alpha$, set to zero, and solve for $\alpha$. We will use the following expressions,

$$\nabla_x q(x) = \frac{3}{2}\text{sign}(x)\sqrt{\sqrt{x^2 + \frac{\delta^2}{4}} - \frac{\delta}{2}}, \quad \nabla_\alpha\|\beta\| = \frac{\alpha}{\|\beta\|}, \quad \nabla_\alpha z^\mathsf{T}\beta = z^\mathsf{T}v. \tag{71}$$

We will also use the expression, $\|\beta\|^2 = \|\beta_*\|^2 + \alpha^2$. Pulling these expressions together we get the following equation for $\alpha$,

$$\sqrt{\sqrt{\|\beta_*\|^2 + \alpha^2 + \frac{\delta^2}{4}} - \frac{\delta}{2}}\frac{\alpha}{\sqrt{\|\beta_*\|^2 + \alpha^2}} = -\frac{2z^\mathsf{T}v}{3}. \tag{72}$$

If we let $k = -\frac{2z^\mathsf{T}v}{3}$, the solution for $\alpha$ is

$$\alpha = k\sqrt{\frac{k^2 + \delta}{2} + \sqrt{\left(\frac{k^2 + \delta}{2}\right)^2 + \|\beta_*\|^2}}. \tag{73}$$

This solution always works for the initializations we considered in Theorem 3.1. Interestingly, it appears that $\beta = \beta_* - \alpha v$ also works for initializations not previously considered. This includes trajectories that pass through the origin, resulting in a change in the sign of $a$.

# B Wide and Deep Linear Networks

Here we discuss how our analysis techniques, developed in the previous section for a single-neuron linear network, can be extended to linear networks with multiple neurons, outputs, and layers.

## B.1 Wide linear networks

We consider the dynamics of a two-layer linear network with $h$ hidden neurons and $c$ outputs, $f(x;\theta) = A^\mathsf{T}Wx$, where $W \in \mathbb{R}^{h\times d}$ and $A \in \mathbb{R}^{h\times c}$. We assume that $h \geq \min(d, c)$, such that this parameterization can represent all linear maps from $\mathbb{R}^d \to \mathbb{R}^c$. As in the single-neuron setting, the rescaling symmetry in this model between the first and second layer implies the $h \times h$ matrix $\Delta = A_0 A_0^\mathsf{T} - W_0 W_0^\mathsf{T}$ determined at initialization remains conserved throughout gradient flow [62]. This can be easily shown from the temporal dynamics of $A$ and $W$,

$$\dot{A} = -\eta_a W X^\mathsf{T}(X\beta - Y), \tag{74}$$

$$\dot{W}^\mathsf{T} = -\eta_w X^\mathsf{T}(X\beta - Y)A^\mathsf{T}. \tag{75}$$

Extending derivations in [30], the NTK matrix can be expressed as

$$K = (\mathbf{I}_c \otimes X)\left(\eta_w A^\mathsf{T}A \oplus \eta_a W^\mathsf{T}W\right)(\mathbf{I}_c \otimes X^\mathsf{T}), \tag{76}$$

where $\otimes$ and $\oplus$ denote the Kronecker product and sum respectively. The Kronecker sum is defined for square matrices $C \in \mathbb{R}^{c\times c}$ and $D \in \mathbb{R}^{d\times d}$ as $C \oplus D = C \otimes \mathbf{I}_d + \mathbf{I}_c \otimes D$.

### B.1.1 Parameter space dynamics

Inspired by our analysis of the single-neuron setting, we introduce two coordinate transformations to study the parameter space dynamics of a wide two-layer linear network. In both analyses we assume whitened input $X^\mathsf{T}X = \mathbf{I}_d$ and let $\eta_a = \eta_w = 1$. However, we will find that the analysis of the dynamics in function space, for general unwhitened data, is more tractable.

**Parameter dynamics when $c = 1$.** Drawing insights from our analysis of the single-neuron scenario ($h = c = 1$), we might consider a combination of hyperbolic and spherical coordinate transformations to study the parameter space dynamics of a wide two-layer linear network. We consider the following two quantities for each hidden neuron $k \in [h]$:

$$\mu_k = a_k\|w_k\|, \qquad \phi_k = \frac{w_k^\mathsf{T}\beta_*}{\|w_k\|\|\beta_*\|}. \tag{77}$$

We will also consider a new matrix quantity $Q \in \mathbb{R}^{h\times h}$ with elements $Q_{kk'} = \frac{w_k^\mathsf{T}w_{k'}}{\|w_k\|\|w_{k'}\|}$. The resulting dynamics for $\mu$ and $\phi$ can be entirely written in terms $\mu, \phi, \Delta$:

$$\dot{\mu} = \sqrt{\mathrm{Diag}(\Delta)^2 + 4\mathrm{Diag}(\mu)^2}\,(\phi - Q\mu), \tag{78}$$

$$\dot{\phi} = M\mathrm{Diag}(\mu)\left((\|\beta_*\|^2 - \phi^\mathsf{T}\mu)I_h + \mathrm{Diag}(\phi)Q\mu - \phi^2\right), \tag{79}$$

where $M = 2\left(\sqrt{\mathrm{Diag}(\Delta)^2 + 4\mathrm{Diag}(\mu)^2} - \mathrm{Diag}(\Delta)\right)^{-1}$. Using the conserved structure of $\Delta$ we can express $Q$ as a function of $\mu$ and $M$,

$$Q = M\mu\mu^\mathsf{T}M - M^{1/2}\Delta M^{1/2}. \tag{80}$$

This approach yields a coupled nonlinear dynamical system with $2h$ variables. Imposing additional assumptions on the initialization, such as permutation invariance between hidden neurons, can simplify the system of differential equations. A similar approach was used by Saad and Solla [78] to derive a set of differential equations for a soft committee machine model, capturing its online learning dynamics in a teacher-student setup, which Goldt et al. [79] extended to its generalization error dynamics.

**Parameter dynamics when $c = h$.** In this analysis we assume an initialization such that the conserved quantities $\Delta = \delta\mathbf{I}_h$, an assumption we will discuss further in Appendix B.1.6, and that $A$ is invertible throughout training. Let $\beta_* = X^\mathsf{T}Y$, which for whitened input, is the unique minimum of the dynamics in function space. We consider the variable $\nu = A^{-1}W\beta_* \in \mathbb{R}^{c\times c}$. Using the identity

that $\dot{A^{-1}} = -A^{-1}\dot{A}A^{-1}$ and our assumption on $\Delta$, we find that the matrix $\nu$ evolves according to the matrix Riccati ODE,

$$\dot{\nu} = \beta_*^\mathsf{T}\beta_* - \delta\nu - \nu^2. \tag{81}$$

Additionally, consider the variable $C = A^\mathsf{T}A$, which evolves according to the matrix Bernoulli ODE

$$\dot{C} = C(\nu + \delta\mathbf{I}_h) + (\nu + \delta\mathbf{I}_h)^\mathsf{T}C - 2C^2. \tag{82}$$

Taken together we have found a change of variables, analogous to the one introduced in Appendix A.2.3 for the single-neuron setting, that evolves according to a matrix Riccati and Bernoulli equation,

$$\dot{\nu} = \beta_*^\mathsf{T}\beta_* - \delta\nu - \nu^2, \qquad\qquad \nu(0) = A_0^{-1}W_0\beta_*, \tag{83}$$

$$\dot{C} = C(\nu + \delta\mathbf{I}_h) + (\nu + \delta\mathbf{I}_h)^\mathsf{T}C - 2C^2, \qquad\qquad C(0) = A_0^\mathsf{T}A_0. \tag{84}$$

However, solving this system exactly as we did in the single-neuron setting is challenging. Unless we assume that $\nu$ and $\beta_*^\mathsf{T}\beta_*$ share the same eigenspace – allowing us to decouple the dynamics of $\nu$ into a set of scalar Riccati equations – the system cannot be easily solved. Instead, we will find that the dynamics of the product $W^\mathsf{T}A$ in function space is more tractable and requires fewer assumptions.

### B.1.2   Function space dynamics

We consider the dynamics of $\beta = W^\mathsf{T}A \in \mathbb{R}^{d\times c}$ in function space, which is governed by the ODE,

$$\dot{\beta} = W^\mathsf{T}\dot{A} + \dot{W}^\mathsf{T}A = -\left(\eta_w X^\mathsf{T}(X\beta - Y)A^\mathsf{T}A + \eta_a W^\mathsf{T}W X^\mathsf{T}(X\beta - Y)\right). \tag{85}$$

Vectorizing using the identity $\mathrm{vec}(ABC) = (C^\mathsf{T}\otimes A)\mathrm{vec}(B)$ equation 85 becomes

$$\mathrm{vec}\left(\dot{\beta}\right) = -\mathrm{vec}\left(\eta_w\mathbf{I}_dX^\mathsf{T}(X\beta - Y)A^\mathsf{T}A + \eta_aW^\mathsf{T}WX^\mathsf{T}(X\beta - Y)\mathbf{I}_c\right), \tag{86}$$

$$= -(\eta_wA^\mathsf{T}A\otimes\mathbf{I}_d + \eta_a\mathbf{I}_c\otimes W^\mathsf{T}W)\mathrm{vec}(X^\mathsf{T}X\beta - X^\mathsf{T}Y), \tag{87}$$

$$= -\underbrace{\left(\eta_wA^\mathsf{T}A\oplus\eta_aW^\mathsf{T}W\right)}_{M}\mathrm{vec}(X^\mathsf{T}X\beta - X^\mathsf{T}Y). \tag{88}$$

As in the single-neuron setting, we find that the dynamics of $\beta$ can be expressed as gradient flow preconditioned by a matrix $M$ that depends on quadratics of $A$ and $W$.

### B.1.3   Proving Theorem 4.1

We first prove Theorem 4.1. Consider a single hidden neuron $k \in [h]$ of the multi-output model defined by the parameters $w_k \in \mathbb{R}^d$ and $a_k \in \mathbb{R}^c$. Let $\beta_k = w_ka_k^\mathsf{T}$ be the $\mathbb{R}^{d\times c}$ matrix representing the contribution of this hidden neuron to the input-output map of the network $\beta = \sum_{k=1}^h\beta_k$. Consider the two gram matrices $\beta_k^\mathsf{T}\beta_k \in \mathbb{R}^{c\times c}$ and $\beta_k\beta_k^\mathsf{T} \in \mathbb{R}^{d\times d}$,

$$\beta_k^\mathsf{T}\beta_k = \|w_k\|^2a_ka_k^\mathsf{T}, \qquad \beta_k\beta_k^\mathsf{T} = \|a_k\|^2w_kw_k^\mathsf{T}. \tag{89}$$

Notice that we can express $\|\beta_k\|_F^2$ as

$$\|\beta_k\|_F^2 = \mathrm{Tr}(\beta_k^\mathsf{T}\beta_k) = \mathrm{Tr}(\beta_k\beta_k^\mathsf{T}) = \|a_k\|^2\|w_k\|^2 \tag{90}$$

At each hidden neuron we have the conserved quantity[8] $\eta_w\|a_k\|^2 - \eta_a\|w_k\|^2 = \delta_k$ where $\delta_k \in \mathbb{R}$. Using this quantity we can invert the expression for $\|\beta_k\|_F^2$ to get

$$\|a_k\|^2 = \frac{\sqrt{\delta_k^2 + 4\eta_a\eta_w\|\beta_k\|_F^2} + \delta_k}{2\eta_w}, \tag{91}$$

$$\|w_k\|^2 = \frac{\sqrt{\delta_k^2 + 4\eta_a\eta_w\|\beta_k\|_F^2} - \delta_k}{2\eta_a}. \tag{92}$$

---

[8]As long as $c > 1$, then the surface of this $d + c$ hyperboloid is always connected, however its topology will depend on the relationship between $d$ and $c$.

When $\|\beta_k\|_F^2 > 0$, we can use these expressions to solve for the outer products $a_k a_k^\mathsf{T}$ and $w_k w_k^\mathsf{T}$ in terms of $\beta_k$ and $\delta_k$,

$$a_k a_k^\mathsf{T} = \frac{\sqrt{\delta_k^2 + 4\eta_a \eta_w \|\beta_k\|_F^2} + \delta_k}{2\eta_w} \frac{\beta_k^\mathsf{T} \beta_k}{\|\beta_k\|_F^2}, \tag{93}$$

$$w_k w_k^\mathsf{T} = \frac{\sqrt{\delta_k^2 + 4\eta_a \eta_w \|\beta_k\|_F^2} - \delta_k}{2\eta_a} \frac{\beta_k \beta_k^\mathsf{T}}{\|\beta_k\|_F^2}. \tag{94}$$

By substituting these expressions into the decompositions $A^\mathsf{T} A = \sum_{k=1}^h a_k a_k^\mathsf{T}$ and $W^\mathsf{T} W = \sum_{k=1}^h w_k w_k^\mathsf{T}$, we derive the representation for $M$ presented in Theorem 4.1: $M = \sum_{k=1}^h M_k$ where

$$M_k = \left( \frac{\sqrt{\delta_k^2 + 4\eta_a \eta_w \|\beta_k\|_F^2} + \delta_k}{2} \right) \frac{\beta_k^\mathsf{T} \beta_k}{\|\beta_k\|_F^2} \oplus \left( \frac{\sqrt{\delta_k^2 + 4\eta_a \eta_w \|\beta_k\|_F^2} - \delta_k}{2} \right) \frac{\beta_k \beta_k^\mathsf{T}}{\|\beta_k\|_F^2}. \tag{95}$$

### B.1.4   Understanding $M$ when there is a single-neuron $h = 1$

When there is a single-hidden neuron $h = \min(d, c) = 1$, the expression for $M$ presented in Theorem 4.1 simplifies allowing us to precisely understand the influence of $\delta$ on the learning regime. When $h = c = 1$, then $\frac{\beta^\mathsf{T} \beta}{\|\beta\|_F^2} = 1$. Therefore, Eq. (7) simplifies to

$$M = \frac{\sqrt{\delta^2 + \eta_a \eta_w 4\|\beta\|^2} + \delta}{2} \mathbf{I}_d + \frac{\sqrt{\delta^2 + \eta_a \eta_w 4\|\beta\|^2} - \delta}{2} \frac{\beta \beta^\mathsf{T}}{\|\beta\|^2}, \tag{96}$$

and we recover Eq. (4) presented in Section 3. When $h = d = 1$, then $\frac{\beta \beta^\mathsf{T}}{\|\beta\|_F^2} = 1$ and thus Eq. (7) simplifies to,

$$M = \frac{\sqrt{\delta^2 + \eta_a \eta_w 4\|\beta\|^2} + \delta}{2} \frac{\beta^\mathsf{T} \beta}{\|\beta\|^2} + \frac{\sqrt{\delta^2 + \eta_a \eta_w 4\|\beta\|^2} - \delta}{2} \mathbf{I}_c. \tag{97}$$

In both settings, $M$ is the weighted sum of the identity matrix and a rank-one projection matrix. While these equations are strikingly similar there is an interesting distinction that arises in the limits of $\delta$. As $\delta \to \infty$, then the first expression for $M$ becomes proportional to $\mathbf{I}_d$, while the second expression for $M$ becomes proportional to the rank-1 projection $\frac{\beta^\mathsf{T} \beta}{\|\beta\|^2}$. Conversely, as $\delta \to -\infty$, then the first expression for $M$ becomes proportional to the rank-1 projection $\frac{\beta \beta^\mathsf{T}}{\|\beta\|^2}$, while the second expression for $M$ becomes proportional to $\mathbf{I}_c$. When $h = d = c = 1$, then $M = \sqrt{\delta^2 + \eta_a \eta_w 4\|\beta\|^2}$ and thus in both limits of $\delta \to \pm\infty$, $M$ becomes a constant independent of $\beta$. In all settings, when $\delta = 0$, $M$ depends on $\beta$. In other words, the influence of $\delta$ on whether the dynamics are lazy, rich, or delayed rich, crucially depends on the relative sizes of dimensions $d$, $h$, and $c$.

### B.1.5   Interpreting $M$ in different limits and architectures

We now seek to more generally understand the influence of the conserved quantities $\delta_i$ and the relative sizes of dimensions $d$, $h$ and $c$ on the learning regime. For a matrix $A \in \mathbb{R}^{d \times c}$, let $\mathrm{Row}(A) \subseteq \mathbb{R}^c$ and $\mathrm{Col}(A) \subseteq \mathbb{R}^d$ denote the row and column space of $A$ respectively.

**Theorem B.1.** *The dynamics are in the lazy regime, for all $t \geq 0$, if $\delta_k \to \infty$ for all $k \in [h]$ and there exists a least squares solution $\beta_* \in \mathbb{R}^{d \times c}$ such that*

$$\mathrm{Row}(\beta_*) \subseteq \mathrm{Span}\left( \bigcup_{k=1}^h \mathrm{Row}\left(\beta_k(0)\right) \right), \tag{98}$$

*or $\delta_k \to -\infty$ for all $k \in [h]$ and there exists a solution such that*

$$\mathrm{Col}(\beta_*) \subseteq \mathrm{Span}\left( \bigcup_{k=1}^h \mathrm{Col}\left(\beta_k(0)\right) \right). \tag{99}$$

*Proof.* As $\delta_k \to \infty$, $M_k \to |\delta_k| \frac{\beta_k^\intercal \beta_k}{\|\beta_k\|_F^2} \otimes \mathbf{I}_d$, implying $\dot\beta_k = -|\delta_k| \frac{\partial \mathcal{L}}{\partial \beta} \left( \frac{\beta_k^\intercal \beta_k}{\|\beta_k\|_F^2} \right)$. Notice that $\left( \frac{\beta_k^\intercal \beta_k}{\|\beta_k\|_F^2} \right)$ is the unique orthogonal projection matrix onto the one-dimensional row space of $\beta_k$. Thus, the dynamics of each $\beta_k$ follow a projected gradient descent in their row space. As a result, $M_k$ will not change and thus the NTK will be static. By assumption there exists a least squares solution $\beta_*$ such that the rows of $\beta_*$ are in the span of the rows of $\beta_k$. Thus, a solution will be reached as $t \to \infty$, while the $M_k$ remain static.

As $\delta_k \to -\infty$ for all $k \in [h]$, $M_k \to \mathbf{I}_c \otimes |\delta_k| \frac{\beta_k \beta_k^\intercal}{\|\beta_k\|_F^2}$, and an analogous argument can be made. $\qquad\square$

Note that the assumptions in Theorem B.1 can be more intuitively expressed in terms of the parameter space $(W, A)$. Except in highly degenerate cases, the assumption $\mathrm{Row}(\beta_*) \subseteq \mathrm{Span}\left( \bigcup_{k=1}^h \mathrm{Row}\left( \beta_k(0) \right) \right)$ is equivalent to the existence of a $\beta_*$ whose rows lie in the span of $\{a_k(0)\}_{k=1}^h$, or, equivalently, to the existence of a matrix $W$ such that $\beta_* = W^\intercal A(0)$. Similarly, the condition $\mathrm{Col}(\beta_*) \subseteq \mathrm{Span}\left( \bigcup_{k=1}^h \mathrm{Col}\left( \beta_k(0) \right) \right)$ is in most cases equivalent to the existence of a matrix $A$ such that $\beta_* = W(0)^\intercal A$.

A direct consequence of Theorem B.1 is that networks which narrow from input to output ($d > c$) must enter the lazy regime with probability 1 as all $\delta_k \to \infty$ whenever $h \geq c$ and assuming independent initializations for all $\beta_k$. In this case, the rows of $\{\beta_1, \ldots, \beta_h\}$ span all of $\mathbb{R}^c$ and thus the condition on the least squares solution is trivially true. By the same logic, networks which expand from input to output ($d < c$) do so as all $\delta_k \to -\infty$ whenever $h \geq d$ and assuming independent initializations for all $\beta_k$. Additionally, when $h \geq \max(d, c)$ and assuming independent initializations for all $\beta_k$, then all networks enter the lazy regime as either all $\delta_k \to \infty$ or all $\delta_k \to -\infty$.

Another interesting implication of Theorem B.1, is that if there does not exist a least squares solution $\beta_*$ with rows in the span of the rows of $\{\beta_1, \ldots, \beta_h\}$, then the network will enter a delayed rich regime as all $\delta_k \to \infty$, where the magnitude of the $\delta_k$ will determine the delay. In this setting, the network is initially lazy, attempting to fit the solution within the row space of the $\beta_k$, but eventually the direction of the rows must change in order to fit the problem, leading to a rich phase. A similar statement involving the columns of $\beta_*$ is true as all $\delta_k \to -\infty$.

### B.1.6  Simplifying $M$ through assumptions on $\Delta$

We now consider how introducing structures on $\Delta$ can lead to simpler expressions for $M$. A natural assumption to consider is the following:

**Assumption B.2** (Isotropic initialization). Let $A \in \mathbb{R}^{h \times c}$ and $W \in \mathbb{R}^{h \times d}$ be initialized such that $\Delta = \eta_w A(0)A(0)^\intercal - \eta_a W(0)W(0)^\intercal = \delta \mathbf{I}_h$.

In *square networks*, where the dimensions of the input, hidden, and output layers coincide ($d = h = c$), and the weights are initialized as $A \sim \mathcal{N}(0, \sigma_a^2/c)$ and $W \sim \mathcal{N}(0, \sigma_w^2/d)$, this assumption is naturally satisfied with $\delta = \sigma_a^2 - \sigma_w^2$ as the dimension $h \to \infty$. However, a limitation of this assumption is that for general $\delta$ it requires $h \leq \min(d, c)$. Specifically, when $\delta > 0$, the isotropic initialization requires that $A(0)A(0)^\intercal \succ 0$, which implies $h \leq c$. Similarly, when $\delta < 0$, the isotropic initialization requires that $W(0)W(0)^\intercal \succ 0$, which implies $h \leq d$. Now we prove two important implications of the isotropic initialization assumption.

**Lemma B.3.** *Let $\Delta = \delta \mathbf{I}_h$. If either $\delta \geq 0$ or $\delta < 0$ and $h \geq d$, we have that*

$$W^\intercal W = \frac{1}{\eta_a} \left( -\frac{\delta}{2} \mathbf{I}_d + \sqrt{\eta_a \eta_w \beta \beta^\intercal + \frac{\delta^2}{4} \mathbf{I}_d} \right). \tag{100}$$

*Proof.* The quantity $\eta_w A A^\intercal - \eta_a W W^\intercal = \delta \mathbf{I}_h$ is conserved in gradient flow. Multiplying on the left by $W^\intercal$ and on the right by $W$ we have that

$$\eta_a (W^\intercal W)^2 + \delta W^\intercal W = \eta_w \beta \beta^\intercal. \tag{101}$$

Completing the square by adding $\frac{\delta^2}{4\eta_a} \mathbf{I}_d$ to both sides and dividing by $\eta_a$ we get the equality,

$$\left( W^\intercal W + \frac{\delta}{2\eta_a} \mathbf{I}_d \right)^2 = \frac{\delta^2}{4\eta_a^2} \mathbf{I}_d + \frac{\eta_w}{\eta_a} \beta \beta^\intercal \tag{102}$$

For $\delta \geq 0$, $W^\intercal W + \frac{\delta}{2\eta_a}\mathbf{I}_d \succeq 0$. For $\delta < 0$, then we know from the conserved quantity that $WW^\intercal + \frac{\delta}{2\eta_a}\mathbf{I}_h = \frac{\eta_w}{\eta_a}AA^\intercal - \frac{\delta}{2\eta_a}\mathbf{I}_h \succ 0$, which implies when $h \geq d$ that $W^\intercal W + \frac{\delta}{2\eta_a}\mathbf{I}_d \succ 0$. As a result, we can take the principal square root of each side,

$$W^\intercal W + \frac{\delta}{2\eta_a}\mathbf{I}_d = \sqrt{\frac{\delta^2}{4\eta_a^2}\mathbf{I}_d + \frac{\eta_w}{\eta_a}\beta\beta^\intercal}, \tag{103}$$

which rearranged gives the final result. $\qquad\square$

**Lemma B.4.** *Let $\Delta = \delta\mathbf{I}_h$. If either $\delta \leq 0$ or $\delta > 0$ and $h \geq c$, we have that*

$$A^\intercal A = \frac{1}{\eta_w}\left(\frac{\delta}{2}\mathbf{I}_c + \sqrt{\eta_a\eta_w\beta^\intercal\beta + \frac{\delta^2}{4}\mathbf{I}_c}\right). \tag{104}$$

*Proof.* The proof is analogous to the proof of Lemma B.3.

$\qquad\square$

From Lemma B.3 and Lemma B.4 we can prove Theorem 4.2, as shown below.

*Proof.* We start from

$$\mathrm{vec}\left(\dot{\beta}\right) = -\underbrace{\left(\eta_w A^\intercal A \oplus \eta_a W^\intercal W\right)}_{M}\mathrm{vec}(X^\intercal X\beta - X^\intercal Y), \tag{105}$$

Plugging in expressions for $W^\intercal W$ from Lemma B.3 and $A^\intercal A$ from Lemma B.4 we can directly write,

$$M = \left(\frac{\delta}{2}\mathbf{I}_c + \sqrt{\eta_a\eta_w\beta^\intercal\beta + \frac{\delta^2}{4}\mathbf{I}_c}\right) \oplus \left(-\frac{\delta}{2}\mathbf{I}_d + \sqrt{\eta_a\eta_w\beta\beta^\intercal + \frac{\delta^2}{4}\mathbf{I}_d}\right) \tag{106}$$

$$= \left(\sqrt{\eta_a\eta_w\beta^\intercal\beta + \frac{\delta^2}{4}\mathbf{I}_c} \otimes \mathbf{I}_d\right) + \left(\mathbf{I}_c \otimes \sqrt{\eta_a\eta_w\beta\beta^\intercal + \frac{\delta^2}{4}\mathbf{I}_d}\right) \tag{107}$$

$\qquad\square$

From this expression for $M(\beta)$ we can easily consider how it simplifies in limiting settings of $\delta$:

$$M \to \begin{cases} \delta\mathbf{I}_{dc} & \delta \to -\infty \\ \sqrt{\eta_a\eta_w\beta^\intercal\beta} \otimes \mathbf{I}_d + \mathbf{I}_c \otimes \sqrt{\eta_a\eta_w\beta\beta^\intercal} & \delta = 0 \\ \delta\mathbf{I}_{dc} & \delta \to \infty. \end{cases} \tag{108}$$

As $\delta \to \pm\infty$, $M \to \delta\mathbf{I}_{dc}$, and the dynamics are lazy. In this limit, the dynamics of $\beta$ converge to the trajectory of linear regression trained by gradient flow and along this trajectory the NTK matrix remains constant. When $\delta = 0$, $M = \sqrt{\eta_a\eta_w\beta^\intercal\beta} \otimes \mathbf{I}_d + \mathbf{I}_c \otimes \sqrt{\eta_a\eta_w\beta\beta^\intercal}$, and the dynamics are rich. Here the NTK changes in both magnitude and direction through training. In the next section we will attempt to better understand these dynamics for intermediate values of $\delta$ through the lens of a mirror flow.

### B.1.7  Deriving a mirror flow for the singular values of $\beta$

For a matrix $\beta$, the dynamics of one of its singular values are given by $\dot{\sigma} = u^\intercal\dot{\beta}v$, where $u$ and $v$ are the corresponding left and right singular vectors. This equality can be derived from chain rule and the fact that $\|u\| = \|v\| = 1$:

$$\dot{\sigma} = \dot{u}^\intercal\beta v + u^\intercal\dot{\beta}v + u^\intercal\beta\dot{v} = \dot{u}^\intercal u\sigma + u^\intercal\dot{\beta}v + \sigma v^\intercal\dot{v} = u^\intercal\dot{\beta}v. \tag{109}$$

In the last equality we used that fact that for any vector $z$ with a fixed norm, $\dot{\|z\|^2} = 2\dot{z}^\intercal z = 0$. Letting $\mathrm{diag} : \mathbb{R}^{d \times c} \to \mathbb{R}^{\min(d,c)}$ be the operator that, given a rectangular matrix, returns a vector of the elements on the main diagonal, we can then write,

$$\dot{\lambda} = \mathrm{diag}(U^\intercal\dot{\beta}V) \tag{110}$$

where $\lambda \in \mathbb{R}^{\min(d,c)}$ is the vector of singular values of $\beta$. In the following lemma, we use the shared singular vector structure between $\beta$ and $A$ and $W$ to rewrite these dynamics as

$$\dot{\lambda} = -M\nabla_\lambda\mathcal{L} \tag{111}$$

where $M$ is a diagonal matrix and $\nabla_\lambda\mathcal{L}$ is the gradient of the loss with respect to the singular values of $\beta$. Without loss of generality we consider $\eta_a = \eta_w = 1$.

**Lemma B.5.** *Let $\Delta = \delta\mathbf{I}_h$. We then have that $\dot{\lambda} = -M\nabla_\lambda\mathcal{L}$, where $M \in \mathbb{R}^{\min(d,c)\times\min(d,c)}$ is a diagonal matrix with*

$$M_{ii} = \begin{cases} \sqrt{\delta^2 + 4\lambda_i^2} & i \leq \min(d,h,c) \\ 0 & \text{otherwise} \end{cases} \tag{112}$$

*Proof.* First note that

$$\dot{\lambda} = \mathrm{diag}(U^\mathsf{T}\dot{\beta}V) \tag{113}$$

$$= -\mathrm{diag}\left(U^\mathsf{T}\left[X^\mathsf{T}(X\beta - Y)A^\mathsf{T}A + W^\mathsf{T}WX^\mathsf{T}(X\beta - Y)\right]V\right) \tag{114}$$

$$= -\mathrm{diag}\left(U^\mathsf{T}X^\mathsf{T}(X\beta - Y)V\Sigma_A^2 + \Sigma_W^2 U^\mathsf{T}X^\mathsf{T}(X\beta - Y)V\right) \tag{115}$$

where we let $W^\mathsf{T}W = U\Sigma_W^2 U^\mathsf{T}$ and $A^\mathsf{T}A = V\Sigma_A^2 V^\mathsf{T}$, using the fact that, under $\Delta = \mathbf{I}_h$, the eigenvectors of $A^\mathsf{T}A$ are the right singular vectors of $\beta$ and the eigenvectors of $W^\mathsf{T}W$ are the left singular vectors of $\beta$. This expression rewrites as

$$\dot{\lambda} = -M\mathrm{diag}\left(U^\mathsf{T}X^\mathsf{T}(X\beta - Y)V\right) \tag{116}$$

where $M \in \mathbb{R}^{\min(d,c)\times\min(d,c)}$ is a diagonal matrix with $M_{ii} = (\Sigma_A^2)_{ii} + (\Sigma_W^2)_{ii}$. For $i \leq \min(d,h,c)$, one can show that $M_{ii} = \sqrt{\delta^2 + 4\lambda_i^2}$. This is because for $i \leq \min(d,h,c)$, $(\Sigma_A^2)_{ii} = (\Sigma_W^2)_{ii} + \delta$ from the conservation law and $(\Sigma_W^2)_{ii}(\Sigma_A^2)_{ii} = \lambda_i^2$ from the definition of $\lambda$. Together this implies $(\Sigma_W^2)_{ii}\left(\delta + (\Sigma_W^2)_{ii}\right) = \lambda_i^2$, which is a quadratic equation in $(\Sigma_W^2)_{ii}$. If $h < \min(d,c)$ then $M_{ii} = 0$ for $i > \min(d,c)$ accounting for rank deficiency of both $A$ and $W$ in this case. Additionally, in our setting of MSE loss, it is straightforward to show that

$$\frac{\partial\mathcal{L}}{\partial\lambda_i} = (U^\mathsf{T}X^\mathsf{T}(X\beta - Y)V)_{ii} \tag{117}$$

We then have that $\nabla_\lambda\mathcal{L} = \mathrm{diag}\left(U^\mathsf{T}X^\mathsf{T}(X\beta - Y)V\right)$, which, combined with our expression for $M$, completes the proof. $\square$

Leveraging Lemma B.5, we can show that the singular values of $\beta$ evolve under a mirror flow in the following theorem.

**Theorem B.6.** *Let $\Delta = \delta\mathbf{I}_h$ and assume $h \geq \min(d,c)$ and $\delta \neq 0$. We then have that the dynamics of $\lambda$, the singular values of $\beta$, are given by the mirror flow*

$$\dot{\lambda} = -\left(\nabla^2\Phi_\delta(\lambda)\right)^{-1}\nabla_\lambda\mathcal{L}, \tag{118}$$

*where $\Phi_\delta(\lambda) = \sum_{i=1}^{\min(d,c)} q_\delta(\lambda_i)$ and $q_\delta$ is the hyperbolic entropy potential*

$$q_\delta(x) = \frac{1}{4}\left(2x\sinh^{-1}\left(\frac{2x}{|\delta|}\right) - \sqrt{4x^2 + \delta^2} + |\delta|\right). \tag{119}$$

*Proof.* When $\Delta = \delta\mathbf{I}_h$, then by Lemma B.5 the dynamics of the singular values of $\beta$ can be expressed as $\dot{\lambda} = -M\nabla_\lambda\mathcal{L}$. Furthermore, when $h \geq \min(d,c)$ and $\delta \neq 0$, we have that $M = \sqrt{\delta^2 + 4\lambda^2}\mathbf{I}_{\min(d,c)}$, where $\lambda^2$ is element-wise, which is always invertible. Observe, this expression for $M$ is the inverse Hessian of the potential $\Phi_\delta(\lambda) = \sum_i q_\delta(\lambda_i)$ for $q_\delta$ specified in the theorem statement. Thus, the dynamics for the singular values are the mirror flow $\dot{\lambda} = -\left(\nabla^2\Phi_\delta(\lambda)\right)^{-1}\nabla_\lambda\mathcal{L}$. $\square$

Theorem B.6 implies that the dynamics for the singular values of $\beta$ can be described as a mirror flow with a $\delta$-dependent potential. This potential was first identified as the inductive bias for diagonal linear networks by Woodworth et al. [14]. Termed *hyperbolic entropy*, this potential smoothly interpolates between an $\ell^1$ and $\ell^2$ penalty on the singular values for the rich ($\delta \to 0$) and lazy ($\delta \to \pm\infty$) regimes respectively. Unfortunately, in our setting we cannot adapt our mirror flow interpretation into a statement on the inductive bias at interpolation because the singular vectors evolve through training. If we introduce additional assumptions — specifically, whitened input data ($X^\mathsf{T}X = \mathbf{I}_d$) and a task-aligned initialization such that the singular vectors of $\beta_0$ are aligned with those of $\beta_*$ — we can ensure that the singular vectors remain constant and thus derive an inductive bias on the singular values. However, in this setting the dynamics decouple completely, implying there is no difference between applying an $\ell^1$ or $\ell^2$ penalty on the singular values. Consequently, even though the dynamics will depend on $\delta$, the final interpolating solution will be independent of $\delta$, making a statement on the inductive bias insignificant.

## B.2  Deep linear networks

We now consider the influence of depth by studying a depth-$(L + 1)$ linear network, $f(x; \theta) = a^\mathsf{T} \prod_{l=1}^{L} W_l x$, where $W_1 \in \mathbb{R}^{h \times d}$, $W_l \in \mathbb{R}^{h \times h}$ for $1 < l \leq L$, and $a \in \mathbb{R}^h$. We assume that the dimensions $d = h$ and that all parameters share the same learning rate $\eta = 1$. For this model the predictor coefficients are computed by the product $\beta = \prod_{l=1}^{L} W_l^\mathsf{T} a \in \mathbb{R}^d$. Similar to our analysis of a two-layer setting, we assume an isotropic initializations of the parameters.

**Definition B.7.** There exists a $\delta \in \mathbb{R}$ such that $aa^\mathsf{T} - W_L W_L^\mathsf{T} = \delta \mathbf{I}_h$ and for all $l \in [L - 1]$ $W_{l+1}^\mathsf{T} W_{l+1} = W_l W_l^\mathsf{T}$.

This assumption can easily be achieved by setting $a = 0$ and $W_l = \alpha O_l$ for all $l \in [L]$, where $O_l \in \mathbb{R}^{d \times d}$ is an random orthogonal matrix and $\alpha \geq 0$. In this case $\delta = -\alpha^2$. Further, notice this parameterization is naturally achieved in the high-dimensional limit as $d \to \infty$ under a standard Gaussian initialization with a variance inversely proportional with width. As in the two-layer setting, this structure of the initialization will remain conserved throughout gradient flow. We now show how two natural quantities of $\beta$, its squared norm $\|\beta\|^2$ and its outer product $\beta\beta^\mathsf{T}$, can always be expressed as polynomials of $\|a\|^2$ and $W_1^\mathsf{T} W_1$ respectively.

**Lemma B.8.** *For a depth-$(L+1)$ linear network with square width ($d = h$) and isotropic initialization, then for all $t \geq 0$,*

$$\|\beta\|^2 = \|a\|^2 \left(\|a\|^2 - \delta\right)^L, \tag{120}$$

$$\beta\beta^\mathsf{T} = \left(W_1^\mathsf{T} W_1\right)^{L+1} + \delta \left(W_1^\mathsf{T} W_1\right)^L. \tag{121}$$

*Proof.* The norm of the regression coefficients is the product $\|\beta\|^2 = a^\mathsf{T} \left(\prod_{l=1}^{L} W_l\right) \left(\prod_{l=1}^{L} W_l\right)^\mathsf{T} a$. Using the conservation of the initial conditions between consecutive weight matrices, $W_{l+1}^\mathsf{T} W_{l+1} = W_l W_l^\mathsf{T}$, we can express this telescoped product as $\|\beta\|^2 = a^\mathsf{T} \left(W_L W_L^\mathsf{T}\right)^d a$. When plugging in the conservation between last two layers, this implies $\|\beta\|^2 = a^\mathsf{T} \left(aa^\mathsf{T} - \delta\mathbf{I}_h\right)^d a$, which expanded gives the desired result.

The outer product of the regression coefficients is $\beta\beta^\mathsf{T} = \left(\prod_{l=1}^{L} W_l\right)^\mathsf{T} aa^\mathsf{T} \left(\prod_{l=1}^{L} W_l\right)$. Using the conserved initial conditions of the last weights we can factor the outer product as the sum, $\beta\beta^\mathsf{T} = \left(\prod_{l=1}^{L} W_l\right)^\mathsf{T} W_L W_L^\mathsf{T} \left(\prod_{l=1}^{L} W_l\right) + \delta \left(\prod_{l=1}^{L} W_l\right)^\mathsf{T} \left(\prod_{l=1}^{L} W_l\right)$. Both these telescoping products factor using the conservation of the initial conditions between consecutive weight matrices giving the desired result. $\square$

We now demonstrate how the quadratic terms $|a|^2$ and $W_1^\mathsf{T} W_1$ significantly influence the dynamics of $\beta$, similar to our analysis in the two-layer setting.

**Lemma B.9.** *The dynamics of $\beta$ are given by a differential equation $\dot{\beta} = -MX^\mathsf{T}\rho$ where $M$ is a positive semi-definite matrix that solely depends on $\|a\|^2$, $W_1^\mathsf{T} W_1$, and $\delta$,*

$$M = (W_1^\mathsf{T} W_1)^L + \|a\|^2 \left(\sum_{l=0}^{L-1} (\|a\|^2 - \delta)^l (W_1^\mathsf{T} W_1)^{L-1-l}\right). \tag{122}$$

*Proof.* Using a similar telescoping strategy used in the previous proof we obtain the form of $M$. $\square$

Finally, we consider how the expression for $M$ simplifies in the limit as $\delta \to 0$ allowing us to be precise about the inductive bias in this setting.

**Theorem B.10.** *For a depth-$(L+1)$ linear network with square width $(d = h)$ and isotropic initialization $\beta_0$ such that $\|\beta(t)\| > 0$ for all $t \geq 0$, then in the limit as $\delta \to 0$, if the gradient flow solution $\beta(\infty)$ satisfies $X\beta(\infty) = y$, then,*

$$\beta(\infty) = \arg\min_{\beta \in \mathbb{R}^d} \left(\frac{L+1}{L+2}\right) \|\beta\|^{\frac{L+2}{L+1}} - \left(\frac{\beta(0)}{\|\beta(0)\|^{\frac{L}{L+1}}}\right)^{\mathsf{T}} \beta \quad \text{s.t.} \quad X\beta = y. \quad (123)$$

*Proof.* Whenever $\|\beta\| > 0$ and in the limit as $\delta \to 0$, then we can find a unique expression for $\|a\|^2$ and $W_1^{\mathsf{T}} W_1$ in terms of $\|\beta\|^2$ and $\beta\beta^{\mathsf{T}}$,

$$\|a\|^2 = \|\beta\|^{\frac{2}{L+1}}, \qquad W_1^{\mathsf{T}} W_1 = \|\beta\|^{-\frac{2L}{L+1}} \beta\beta^{\mathsf{T}}. \quad (124)$$

Plugged into the previous expression for $M$ results in a positive definite rank-one perturbation to the identity,

$$M = \|\beta\|^{\frac{2L}{L+1}} \mathbf{I}_d + L\|\beta\|^{-\frac{2}{L+1}} \beta\beta^{\mathsf{T}}. \quad (125)$$

Using the Sherman-Morrison formula we find that $M^{-1}$ is

$$M^{-1} = \|\beta\|^{-\frac{2L}{L+1}} \mathbf{I}_d + \left(\frac{L}{L+1}\right) \|\beta\|^{-\frac{4L+2}{L+1}} \beta\beta^{\mathsf{T}} \quad (126)$$

We can now apply a time-warped mirror flow analysis similar to the analysis presented in Appendix A.4. Consider the time-warping function $g_\delta(\|\beta\|) = \|\beta\|^{-\frac{L}{L+1}}$ and the potential $\Phi(\beta) = \left(\frac{L+1}{L+2}\right) \|\beta\|^{\frac{L+2}{L+1}}$, then its not hard to show $M^{-1} = g_\delta(\|\beta\|)\nabla^2\Phi(\beta)$. This gives the desired result. $\square$

This theorem is a generalization of Proposition 1 derived in [9] for two-layer linear networks in the rich limit to deep linear networks in the rich limit. We find that the inductive bias, $Q(\beta) = (\frac{L+1}{L+2})\|\beta\|^{\frac{L+2}{L+1}} - \|\beta_0\|^{-\frac{L}{L+1}} \beta_0^{\mathsf{T}} \beta$, strikes a depth-dependent balance between attaining the minimum norm solution and preserving the initialization direction.

# C  Piecewise Linear Networks

Here, we elaborate on the theoretical results presented in Section 5. Our goal is to extend the tools developed in our analysis of linear networks to piecewise linear networks and understand their limitations. We focus on the dynamics of the input-output map, rather than on the inductive bias of the interpolating solutions. As discussed in Azulay et al. [9], Vardi and Shamir [80], extending a mirror flow style analysis directly to non-trivial piecewise linear networks is very difficult or provably impossible. In this section, we first describe the properties of the input-output map of a piecewise linear function, then describe the dynamics of a two-layer network, and finally discuss the challenges in extending this analysis to deeper networks and potential directions for future work.

## C.1  Surface of a piecewise linear network

The input-output map of a piecewise linear network $f(x; \theta)$, with $l$ hidden layers and $h$ hidden neurons per layer, is comprised of potentially $O(h^{dl})$ connected linear regions, each with their own vector of predictor coefficients [65]. The exploration of this complex surface has been the focus of numerous prior works, the vast majority of them focused on counting and bounding the number of linear regions as a function of the width and depth [81, 82, 83, 84, 65, 85, 86, 87]. The central object in all of these studies is the *activation region*,

**Definition C.1.** For a piecewise linear network $f(x; \theta)$, comprising $N$ hidden neurons with pre-activation $z_i(x; \theta)$ for $i \in [N]$, let the *activation pattern* $\mathcal{A}$ represent an assignment of signs $a_i \in \{-1, 1\}$ to each hidden neuron. The *activation region* $\mathcal{R}(\mathcal{A}; \theta)$ is the subset of input space that generates $\mathcal{A}$,

$$\mathcal{R}(\mathcal{A}; \theta) = \{x \in \mathbb{R}^d \mid \forall i \ a_i z_i(x; \theta) > 0\}. \quad (127)$$

The input-output map is linear within each non-empty activation region and continuous at the boundary be-

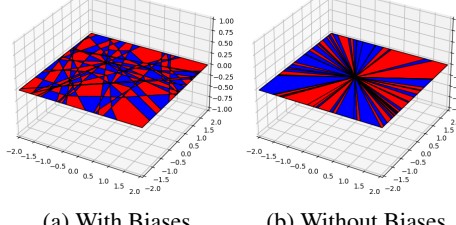

|   (a) With Biases   |   (b) Without Biases   |

Figure 9: **Surface of a ReLU network.** Here we depict the surface of a three-layer ReLU network $f(x; \theta) : \mathbb{R}^2 \to \mathbb{R}$ with twenty hidden units per layer at initialization, comparing configurations with biases (left) and without biases (right). The network with biases partitions input space into convex polytopes that tile input space. The network without biases partitions input space into convex conic sections emanating from the origin. Each region exhibits a distinct activation pattern, allowing the partition to be colored with two colors based on the parity of active neurons. The network operates linearly within each region and maintains continuity across boundaries.

tween regions. Linearity implies that every non-empty[9] activation region is associated with a *linear predictor* vector $\beta_{\mathcal{R}} \in \mathbb{R}^d$ such that for all $x \in R(\mathcal{A}; \theta)$, $\beta_{\mathcal{R}} = \nabla_x f(x; \theta)$. Continuity implies that the boundary between regions is formed by a hyperplane determined by where the pre-activation for a neuron is exactly zero, $\{x : z_i(x; \theta) = 0\}$. When the neighboring regions have different linear predictors[10], then this hyperplane is orthogonal to their difference, which is a vector in the span of the first-layer weights. Taken together, this implies that the union of all activation regions forms a convex partition of input space, as shown in Fig. 9. We now present a surprisingly simple, yet to the best of our knowledge not previously understood property of this partition:

**Proposition C.2** (2-colorable). *If $f(x; \theta)$ lacks redundant neurons, implying that every neuron influences an activation region, then the partition of input space can be colored with two distinct colors such that neighboring regions do not share the same color.*

The justification for this proposition is straightforward. There is one color for regions with an even number of active neurons and another for regions with an odd number of active neurons. Because $f(x; \theta)$ lacks redundant neurons, there does not exist a boundary between activation regions where two neurons activations change simultaneously. In this work, we solely utilize this proposition for visualization purposes, as shown in Fig. 9. Nonetheless, we believe it may be of independent interest as it strengthens the connection between the surface of piecewise linear networks and the *mathematics of paper folding*, a connection previously alluded to in the literature [82].

---

[9]While it is trivial to see that for a network $f(x; \theta)$ with $N$ hidden neurons there are $2^N$ distinct activation patterns, not all activation patterns are attainable. See Raghu et al. [65] for a discussion.

[10]It is possible for neighboring regions to have the same linear predictor. Some works define linear regions as maximally connected component of input space with the same linear predictor [87].

## C.2 Dynamics of a two-layer piecewise linear network

We consider the dynamics of a two-layer piecewise linear network without biases, $f(x;\theta) = a^\intercal \sigma(Wx)$, where $W \in \mathbb{R}^{h \times d}$ and $a \in \mathbb{R}^h$. The activation function is $\sigma(z) = \max(z, \gamma z)$ for $\gamma \in [0, 1)$, which includes ReLU $\gamma = 0$ and Leaky ReLU $\gamma \in (0, 1)$. We permit $h > d$, which in the limit as $h \to \infty$, ensures the network possesses the functional expressivity to represent any continuous nonlinear function from $\mathbb{R}^d$ to $\mathbb{R}$ passing through the origin. Following a similar strategy used in Section 4, we consider the contribution to the input-output map from a single hidden neuron $k \in [h]$ with parameters $w_k \in \mathbb{R}^d$ and $a_k \in \mathbb{R}$. As in the linear setting, each hidden neuron is associated with a conserved quantity, $\delta_k = \eta_w a_k^2 - \eta_a \|w_k\|^2$. Unlike in the linear setting, this neuron's contribution to the output $f(x_i; \theta)$ is regulated by whether the input $x_i$ is in the neuron's *active halfspace*, $\{x \in \mathbb{R}^d : w_k^\intercal x > 0\}$. Let $C \in \mathbb{R}^{h \times n}$ be the matrix with elements $c_{ki} = \sigma'(w_k^\intercal x_i)$, which determines the activation of the $k^{\text{th}}$ neuron for the $i^{\text{th}}$ training data point. The subgradient $\sigma'(z) = 1$ if $z > 0$, $\sigma'(z) \in [\gamma, 1]$ if $z = 0$, and $\sigma'(z) = \gamma$ if $z < 0$. These activation functions exhibit positive homogeneity, implying $\sigma(z) = \sigma'(z)z$. Thus, we can express $\sigma(w_k^\intercal x_i) = c_{ki} w_k^\intercal x_i$, allowing us to express the gradient flow dynamics for $w_k$ and $a_k$ as

$$\dot{a}_k = -\eta_a w_k^\intercal \left( \sum_{i=1}^n c_{ki} x_i \rho_i \right), \qquad \dot{w}_k = -\eta_w a_k \left( \sum_{i=1}^n c_{ki} x_i \rho_i \right), \tag{128}$$

where $\rho_i = f(x_i; \theta) - y_i$ is the residual associated with the $i^{\text{th}}$ training data point. If we let $\beta_k = a_k w_k$, which determines the contribution of each hidden neuron to the output $f(x_i; \theta)$, then its not hard to see that the gradient flow dynamics of $\beta_k$ are

$$\dot{\beta}_k = - \underbrace{\left( \eta_w a_k^2 \mathbf{I}_d + \eta_a w_k w_k^\intercal \right)}_{M_k} \underbrace{\left( \sum_{i=1}^n c_{ki} x_i \rho_i \right)}_{\xi_k}. \tag{129}$$

As in the linear setting, the matrix $M_k \in \mathbb{R}^{d \times d}$ appears as a preconditioning matrix on the dynamics Using the exact same derivation presented in Appendix A.3, whenever $a_k^2 \neq 0$, we can express $M_k$ entirely in terms of $\beta_k$ and $\delta_k$,

$$M_k = \frac{\sqrt{\delta_k^2 + 4\eta_a \eta_w \|\beta_k\|^2} + \delta_k}{2} \mathbf{I}_d + \frac{\sqrt{\delta_k^2 + 4\eta_a \eta_w \|\beta_k\|^2} - \delta_k}{2} \frac{\beta_k \beta_k^\intercal}{\|\beta_k\|^2}. \tag{130}$$

However, unlike in the linear setting, the vector $\xi_k \in \mathbb{R}^d$ driving the dynamics is not shared for all neurons because of its dependence on $c_{ki}$. Additionally, the NTK matrix in this setting depends on $M_k$ and $C$, with elements $K_{ij} = \sum_{k=1}^h c_{ki} x_i^\intercal \left( \eta_w a_k^2 \mathbf{I}_d + \eta_a w_k w_k^\intercal \right) x_j c_{kj}$. Thus, in order to assess the temporal dynamics of the NTK matrix, we must understand the dynamics of $M_k$ and $C$. We consider a *signed spherical coordinate* transformation separating the dynamics of $\beta_k$ into its directional $\hat{\beta}_k = \text{sgn}(a_k) \frac{\beta_k}{\|\beta_k\|}$ and radial $\mu_k = \text{sgn}(a_k) \|\beta_k\|$ components, such that $\beta_k = \mu_k \hat{\beta}_k$. Here, $\hat{\beta}_k$ determines the orientation and direction of the halfspace where the $k^{\text{th}}$ neuron is active, while $\mu_k$ determines the slope of the linear region in this halfspace. These coordinates evolve according to,

$$\dot{\mu}_k = -\sqrt{\delta_k^2 + 4\eta_a \eta_w \mu_k^2} \hat{\beta}_k^\intercal \xi_k, \qquad \dot{\hat{\beta}}_k = -\frac{\sqrt{\delta_k^2 + 4\eta_a \eta_w \mu_k^2} + \delta_k}{2\mu_k} \left( \mathbf{I}_d - \hat{\beta}_k \hat{\beta}_k^\intercal \right) \xi_k. \tag{131}$$

These equations can be derived directly from Eq. (128) through chain rule similar to Appendix A.2.1. In fact its worth noting that the this change of coordinates is similar to the change of coordinates used in the single-neuron analysis. Expressed in terms of the parameters, $\hat{\beta}_k = \frac{w_k}{\|w_k\|}$ and $\mu_k = a_k \|w_k\|$.

# D   Experimental Details

We used Google Cloud Platform (GCP) nodes to run all experiments. Figure 1 experiments were run on a node with 360 AMD Genoa CPU cores with runtime totaling approximately 90 minutes including averaging over seeds as described below. Neural network training and NTK calculation for Figure 5 was performed on single A100 GPU nodes. Runtime was approximately 20 hours for Figure 5(a), four hours for 5(b), 12 hours for 5(c) (with individual runs ranging from five to 30 minutes depending on the number of datapoints), and 12 hours for 5(d). Figures 2, 3, and 4 are not compute-heavy, and these experiments were run on a personal computer. Overall, we estimate approximately 200 hours of single A100 runtime as well as 100 hours of the 360-core node accounting for failed runs and exploratory experiments.

## D.1   Figure 1: Teacher-Student with Two-layer ReLU Networks

For Fig. 1 we consider a student-teacher setup similar to that in [8], with one-hidden layer ReLU networks of the form $f(x; \theta) = \sum_{i=1}^{m} a_i \sigma(w_i^\mathsf{T} x)$, where $f : \mathbb{R}^d \to \mathbb{R}$ and $\sigma$ is the ReLU activation function. The teacher model, $f^{\text{teacher}}$, has $m = k$ hidden neurons initialized as $w_i^{\text{teacher}} \overset{\text{i.i.d.}}{\sim}$ $\text{Unif}(S^{d-1})$ and $a_i \overset{\text{i.i.d.}}{\sim} \text{Unif}(\{\pm 1\})$ for $i \le k$. The student, $f^{\text{student}}$, in turn, has $h$ hidden neurons. We use a symmetrized initialization, as considered in [8], where for $i \le h/2$, we sample $w_i \overset{\text{i.i.d.}}{\sim} S^{d-1}$ and $a_i \overset{\text{i.i.d.}}{\sim} \text{Unif}(\{\pm 1\})$, and then for $i \ge \frac{h}{2} + 1$ we symmetrize by setting $w_i = w_{i-h/2}$ and $a_i = -a_{i-h/2}$. This ensures that $f^{\text{student}}$ predicts 0 on any input at initialization.

Note that the *base* student initialization described thus far is perfectly balanced at each neuron, that is $\delta_i = 0$ for $i \in [m]$; we also define this to be our setting where the scale $\tau$ is 1. In order to transform the base initialization into a particular setting of $\tau$ and $\delta$, we first solve for the relative layer scaling $\alpha$ in $\delta^2 = \tau^2(\alpha^2 - \alpha^{-2})$ and then scale each $w_i$ by $\tau/\alpha$ and each $a_i$ by $\tau\alpha$. We obtain a training dataset $\{x^{(i)}, y^{(i)}\}_{i=1}^{n}$ by sampling $x^{(i)} \overset{\text{i.i.d.}}{\sim} S^{d-1}$ and computing noiseless labels as $y^{(i)} = f^{\text{teacher}}(x^{(i)}; \theta^{\text{teacher}})$. The student is then trained with full-batch gradient descent on a mean square loss objective.

**Figure 1 (a).**

Here the setting is: $d = 2$, $h = 50$, $k = 3$, and $n = 20$. We sample a single teacher and then train four students with the same base initialization but different configurations of $\tau$ and $\delta$: $(\tau = 0.1, \delta = 0)$ and $(\tau = 2, \delta = 0)$ for the left subfigure, and $(\tau = 0.1, \delta = 1)$ and $(\tau = 0.1, \delta = -1)$ for the right subfigure. Training is for 1 million steps at a learning rate of 1e-4.

**Figure 1 (b).**

Here the setting is: $d = 100$, $m = 50$, $k = 3$, and $n = 1000$, as in Fig. 1c of [8]. Training is performed with learning rate of 5e-3$/\tau^2$. Test error is computed as mean square error over a held-out set of 10,000 datapoints. We sweep over $\tau$ over a logarithmic scale in the range $[0.1, 2]$ and $\delta$ over a linear scale in the range $[-1, 1]$. We average over 16 random seeds, where the seed controls the sampling of: the teacher weights $\theta^{\text{teacher}}$, the base initialization of $\theta^{\text{student}}$, and the training data $\{x^{(i)}\}_{i=1}^{n}$. In this way, each random seed is used for a sweep over all combinations of $\tau$ and $\delta$ in the sweep; we simply apply the scaling described above to get to each point on the $(\tau, \delta)$ grid. The kernel distance computed is as defined in [27], where here we compute it at time $t$ relative to the kernel at initialization, i.e. $S(t) = 1 - \langle K_0, K_t \rangle / (\|K_0\|_F \|K_t\|_F)$. In Fig. 10, we additionally plot Hamming and parameter distances relative to initialization, as well as training loss, while training for ten times longer than in Fig. 1 (b).

Notebooks generating all two-layer experiment figures are provided here.

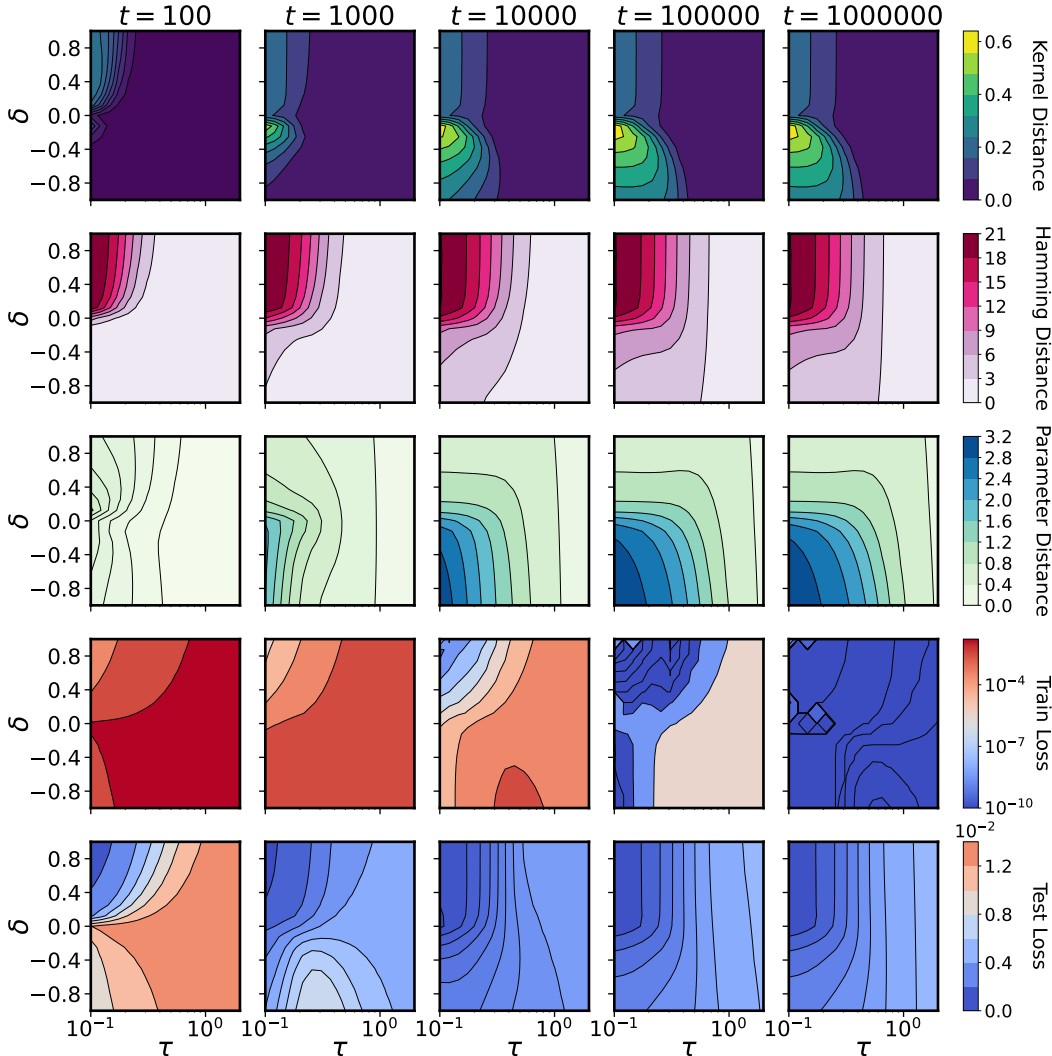

Figure 10: **Supporting figures for Fig. 1 (b).** We plot Hamming distance, parameter distance, and training loss, on top of the test loss and kernel distance considered in Fig. 1 (b), and train for ten times longer than in Fig. 1 (b). We observe that although training loss still drops between $10^5$ and $10^6$ steps, the test loss and other distances considered remain largely unchanged. Training loss is saturated at 1e-10.

## D.2  Figures 2, 3, 4: Single-Neuron Linear Network

Figures 2, 3, and 4 were generated by simulating gradient flow using `scipy.integrate.solve_ivp` function with the RK45 method for solving the ODEs, with a relative tolerance of $1 \times 10^{-6}$ and time span of $(0, 20)$. In the experiments with full-rank data, we used $X^\mathsf{T}X = \mathbf{I}_2$, $\beta_* = \begin{bmatrix} 0 \\ 1 \end{bmatrix}$, and $\beta_0 = \begin{bmatrix} -1 \\ 0 \end{bmatrix}$. For the experiment with low-rank data, we used $X^\mathsf{T}X = \begin{bmatrix} 0.25 & 0.5 \\ 0.5 & 1 \end{bmatrix}$, $\beta_* = \begin{bmatrix} 0.44 \\ 0.88 \end{bmatrix}$, and $\beta_0 = \begin{bmatrix} 0.4 \\ 0.05 \end{bmatrix}$. See the discussion in Appendix A.2 for details on how we determined our theoretical predictions. A notebook generating all the figures is provided here.

## D.3  Figure 5:

### Kernel Distance

We trained LeNet-5 [88] (with ReLU nonlinearity and Max Pooling) on MNIST [88]. We use He initialization [89] and divide the first layer weights by $\alpha$ and multiply the last layer weights by $\alpha$ at initialization, which keeps the network functionally the same at initialization. We trained the model

for 500 epochs with a learning rate of 1e-4 and a batch size of 512. The parameter distance is defined as the $L_2$ distance between all the parameters. To quantify the distance between the activations, we binarize the hidden activation with 1 representing an active neuron. We evaluate Hamming distance over all the binarized hidden activations normalized by the the total number of the activations. We use kernel distance [27], defined as $S(t_1, t_2) = 1 - \langle K_{t_1}, K_{t_2} \rangle / (\|K_{t_1}\|_F \|K_{t_2}\|_F)$, which is a scale invariant measure of similarity between the NTK at two points in time. We subsample 10% of MNIST to evaluate the Hamming distance and kernel distance. All curves in the figure are averaged over 8 runs.

**Gabor Filters**

We are training a small ResNet based on the CIFAR10 script provided in the DAWN benchmark (code available here). The only modifications to the provided code base are we increase the convolution kernel size from $3 \times 3$ to $15 \times 15$, to better observe the learned spatial patterns, and we set the weight decay parameter to $0$ to avoid confounding variables. Moreover, we are dividing the convolutional filters weights by a parameter $\alpha$ (after standard initialization) which controls the balancedness of the network. To quantify the smoothness of the filters, we compute the normalized Laplacian of each filter $w_{ij} \in \mathbb{R}^{15 \times 15}$, over input $i = (1, 2, 3)$ and output $j = (1, ..., 64)$ channels

$$\text{smoothness}(w_{ij}) := \left\| \frac{w_{ij}}{\|w_{ij}\|_2} * \Delta \right\|_2^2 \tag{132}$$

where the Laplacian kernel is defined as

$$\Delta := \begin{pmatrix} -0.25 & -0.5 & -0.25 \\ -0.5 & 2 & -0.5 \\ -0.25 & -0.5 & -0.25 \end{pmatrix}. \tag{133}$$

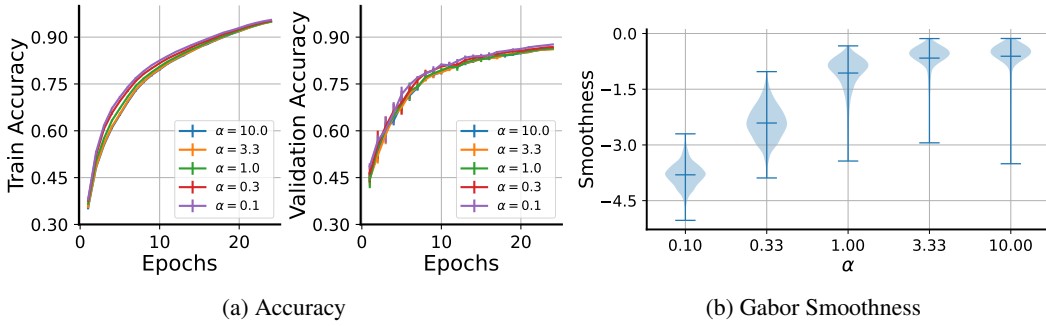

(a) Accuracy

(b) Gabor Smoothness

Figure 11: **Interpreting convolutional filters.** CNN experiments on CIFAR10. We can see in **A)** that all networks achieve comparable training and test accuracy, despite the modification in initialization. However, in **B)** we see that networks with a small initialization ($\alpha < 1$) learn much smoother filters, giving quantiative support to results in Fig. 6. The smoothness is defined as the normalized Laplacian of the filters (see text, eq. 132).

**Random Hierarchy Model**

We refer to [67], who originally proposed the random hierarchy model (RHM) as a tool for studying how deep networks learn compositional data, for a more in-depth treatment. Here we briefly recap the setup following the notation used in [67].

An RHM essentially lets us build a random classification task with a clear hierarchical structure. The top level of the RHM specifies $m$ equivalent high-level features for *each* class label in $\{1, \ldots, n_c\}$, where each feature has length $s$ and $n_c$ is the number of classes. For example, suppose the vocabulary at the top level is $\mathcal{V}_L = \{a, b, c\}$, $n_c = 2$, $m = 3$, and $s = 2$. Then in a particular instantiation of this RHM, we might have that Class 1 has $ab$, $aa$, and $ca$ as equivalent high-level features (this is precisely the example used in Fig.1 of [67]). Class 2 will then have three random high-level features, with the constraint that they are **not** features for Class 1, for example, $bb, bc, ac$.

Each successive level specifies $m$ equivalent lower-level features for each "token" in the vocabulary at the previous level. For example, if $\mathcal{V}_{L-1} = \{d, e, f\}$, we might have that $a$ can be equivalently

represented as $de$, $df$, or $ff$; $b$ and $c$ will each have $m$ equivalent representations of their own. We assume that the vocabulary size, $v$, is the same at all levels. Therefore, sampling an RHM with hyperparameters $n_c, m, s, v$ requires sampling $mn_c + (L-1)mv$ rules.

In order to sample a datapoint from an RHM, we first sample a class label (e.g. Class 1), then uniformly sample one of the highest level features, (e.g. $ab$), then for each "token" in this feature we sample lower level features (e.g. $a \rightarrow de$, $b \rightarrow ee$), and so on recursively. The generated sample will therefore have length $s^L$ and a class label. For training a neural network to perform this classification task, each input is converted into a one-hot representation, which will be of shape $(s^L, v)$, and is then flattened.

We use the code released by [67] to train an MLP of width 64 with three hidden layers to learn an RHM with $L = 3, n_c = 8, m = 4, s = 2, v = 8$. The main change we make is allowing for scaling the initialization of the first layer by $1/\alpha$ and the initialization the readout layer by $\alpha$. We then sweep over $\alpha \in \{0.03, 0.1, 0.3, 1, 3, 10\}$ and over the number of datapoints in the training set, which is specified as a fraction of the total number of datapoints the RHM can generate. We average test accuracy, which is by default computed on a held-out set of 20,000 samples, over six random seed configurations, where each configuration seeds the RHM, the neural network, and the data generation.

We train with the default settings used in [67], that is stochastic gradient descent with momentum of 0.9, run for 250 epochs with a learning rate initialized at 6.4 (0.1 times width) and decayed with a cosine schedule down to 80% of epochs. The batch size of 128; we do not use biases or weight decay.

**Grokking**

We are training a one layer transformer model on the modular arithmetic task in Power et al. [68]. Our experimental code is based on an existing Pytorch implementation (code available here). The only modifications to the provided code base is that we use a single transformer layer (instead of the default 2-layer model). Prior analysis in Nanda et al. [72] has shown that this model can learn a minimal (attention-based) circuit that solves the task.

We study the effects on grokking time (defined as $\geq 0.99$ accuracy on the validation data) of two manipulations. Firstly, we divide the embedding weights of the positional and token embeddings by the same balancedness parameter $\alpha$ as in the CNN gabor experiments. Secondly, like in Kumar et al. [69], we multiply the output of the model (i.e., the logits) by a factor $\tau$ and divide the learning rate by $\tau^2$.

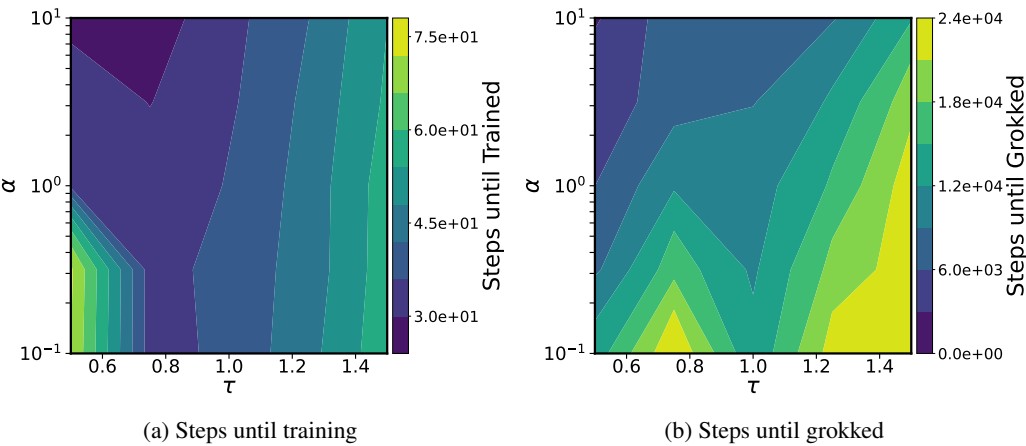

(a) Steps until training        (b) Steps until grokked

Figure 12: **Transformer Grokking in Modular Arithmetic Task. A)** Shows the number of training steps required until the training accuracy passes a predefined threshold of 99%; we sample scaling $\tau \in \{0.5, 0.75, 1.0, 1.25, 1.5\}$ [69] and balance $\alpha \in \{0.1, 0.3, 1.0, 3.0, 10\}$ on a regular grid over $n = 5$ random initializations with a maximal computational budget of $m = 30{,}000$ training steps. **B)** Same as **A)**, but reporting the number of training steps required until the test performance passes the predefined threshold of 99%. We clearly see the fastest grokking in an unbalanced rich setting.

