# OpenReview forum: "Get rich quick: exact solutions reveal how unbalanced initializations promote rapid feature learning"
_NeurIPS.cc/2024/Conference — NeurIPS 2024 spotlight_

### Official Review · Reviewer_K8a2 · 2024-06-21

**Soundness:** 3
**Presentation:** 3
**Contribution:** 3
**Rating:** 7
**Confidence:** 3

**Summary:**

This paper analyses the lazy versus rich learning dynamics of minimal but finite neural networks by deriving exact solutions under arbitrary layerwise initialization and learning rates. The theoretical insights are successively extended to networks of increasing complexity and corroborated by several experiments.

**Strengths:**

The paper is well written and provides an exact understanding of the learning dynamics of very simple neural networks, in increasing complexity over the course of the paper. The figures complement the derivations very comprehensively. In this way, the paper makes a significant contribution to the understanding of neural network training dynamics. I am not able to judge the claimed novelty, as I am not entirely familiar with the related work.

**Weaknesses:**

Overall the paper leaves little room for criticism. The results being limited to shallow, often linear neural networks trained with small learning rates is justified by the detailed understanding of the training dynamics, and mitigated by experiments. It could have been acknowledged more clearly how the maximal update parameterization by Yang and Hu (2021) already shows that upstream initializations are necessary for non-vanishing feature learning in large models.

**Questions:**

- Why do you call the layerwise initialization variances and learning rates *initialization geometry* when learning rate scalings clearly only influence the training dynamics? Similarly, the term geometry is unspecific and does not highlight the interplay between the layers. I encourage the authors to find a more suitable terminology for the choice of hyperparameters.
- Why is the kernel distance in Figure 1 eventually maximized by a downstream initialization $\delta<0$, and not by an upstream initialization $\delta>0$ where feature learning occurs rapidly both directionally and radially?

**Limitations:**

The main limitations are acknowledged in the Conclusion section. It is acceptable that an exact analysis of learning dynamics does not include deep nonlinear networks trained with large learning rates.

---

> ### Author Rebuttal · Authors · 2024-08-05
>
> Thank you for your thoughtful feedback and constructive questions. We are grateful for your positive feedback on our work and are committed to improving our manuscript by addressing each of the weaknesses you identified.
>
> **Connections to infinite width parametrizations.** We agree with the review and are adding a subsection to Section 5 carefully outlining how our results can be extended to infinite width settings and acknowledging connections to existing parametrizations. In the two-layer setting, muP corresponds to the mean-field parameterization, which, for input dimension constant in width, can be written as $f(x) = 1/k a^T\sigma(Wx)$ with $a_i \sim \mathcal{N}(0, 1)$ and $W_{ij} \sim \mathcal{N}(0, 1/d)$. We note that this actually leads to the per-neuron conserved quantity being 0, or balanced, in expectation, with a non-vanishing variance. Please see our response to reviewer JJEZ for a more detailed discussion on the connection between our analysis of finite-width networks and existing works on infinite-width networks.
>
> **Initialization geometry.** We chose the term initialization geometry because the per-layer learning rates and layer magnitudes collectively determine the geometry of the surface that the dynamics are constrained to. This geometry is leveraged consistently in our theory through the use of conserved quantities and is clearly visualized for the one hidden neuron model in Figure 2.
>
> **Kernel movement in downstream initialization.** In Figure 1(b), the small-scale downstream initialization ($\delta<0$) starts off lazy, attempting to fit the training data by only changing the small readout weights $a$ with minimal movement in the large first-layer weights $\{w_i\}$. Up to this stage, the downstream initialization exhibits smaller kernel movement than the upstream initialization, which undergoes a rapid change in the kernel. However, in order to interpolate the training data, the network with downstream initialization needs to align its hidden neurons by a non-trivial amount, thus undergoing a change from lazy to rich learning. This alignment of the hidden neurons, which are large in scale, results in a dramatic change in the kernel. In the case of the upstream initialization, the network is able to interpolate the training data by simply aligning its hidden neurons (which are small in scale) directionally and radially as needed, leading to a smaller movement in the kernel overall. See our response to reviewer JJEZ for a detailed discussion on the dynamics of the NTK matrix.
>
> We hope we addressed your points regarding our work. Thank you again for your constructive feedback!

---

> > ### Comment · Reviewer_K8a2 · 2024-08-09
> >
> > I thank the authors for their thoughtful response. I think both the added explanation on the dynamics of the NTK matrix and the connection to infinite-width mean-field literature are valuable additions. In particular, the fact that mean-field parameterization is balanced with non-vanishing variance is interesting, and the distinction between the function scale and relative scale mechanism. Can I understand the mean-field parameterization/muP as the unique width-dependent scaling rule that achieves $\delta$ to remain width-independent and hence not degenerate in behaviour with width $k\to \infty$ to for example strictly lazy behaviour in the NTK parameterization?
> >
> > Concerning the term ‘initialization geometry’, I still believe that per-layer learning rates are not part of the *initialization*, but only of the *optimization*, hence terms in the direction of ‘optimization geometry’ would make more sense to me.
> >
> > Overall, I believe this paper makes a valuable contribution and I will keep my positive evaluation.

---

> > > ### Author Response · Authors · 2024-08-13
> > >
> > > Thank you for your response, and we agree that the added discussion of NTK dynamics and the connection to infinite-width literature are valuable additions to our work.
> > >
> > > To clarify our discussion on width-dependent parametrizations, there are two “knobs” that can determine the degree of feature learning: (1) the overall function scale and (2) the relative scale between layers (this is $\delta$ in our analysis). The difference in feature learning between a mean field parameterization and an NTK parameterization is due to a change in the first knob (the overall function scale). Under both these parameterizations the distribution of $\delta$ is the same even as width $k \to \infty$ (expected $\delta$ is zero, but with a non-vanishing variance). It is possible to have an infinite-width parameterization that uses the second knob to enter into the lazy regime (in this setting the expected $\delta$ would be non-zero and depend on width). We found Reference [14] to be helpful in understanding the connection between our finite-width analysis of feature learning and existing works in the infinite-width setting.
> > >
> > > Thank you for clarifying your concern with our use of “initialization geometry”. Your point makes sense and the suggested “optimization geometry” is a good possibility, which we will consider.
> > >
> > > Thank you again for your questions, feedback, and constructive criticism!

---

### Official Review · Reviewer_JJEZ · 2024-07-09

**Soundness:** 3
**Presentation:** 4
**Contribution:** 3
**Rating:** 6
**Confidence:** 4

**Summary:**

The paper studies the dynamics of training in neural networks with the scope of identifying how the variance of weights' initialization together with layer-wise learning rates determines different learning regimes, encompassing lazy, rich, and the transition between them. The paper identifies a conserved quantity $delta$ that is preserved throughout training (i.e. $\dot{\delta} = 0$) and depends on both the weight's magnitude and the learning rates. Thus, the sign and magnitude of $\delta$ at initialization affect learning and the geometry of the landscape that is traversed through gradient flow. The paper finds three different regimes, starting from a solvable model of two-layer linear network.

1. $\delta > 0$ (named *upstream initialization*) induces lazy dynamics and corresponds to the case where $\eta_w a^2 >> \eta_a ||w||^2$, where $w$ and $a$ are the first and last layer weights, resp.

2. $\delta = 0$ (*Balanced*): corresponds to rich dynamics.

3. $\delta < 0$: initial lazy fitting regime and second rich phase.

The authors then extend these findings to wider and deeper networks and piece-wise linear activations.

**Strengths:**

1. The paper finds a simplified model that can be thoroughly analyzed and gives very precise statements on what causes rich and lazy regimes based on the single conserved quantity $\delta$, which makes the results very clean. This quantity is then naturally connected to the NTK, which makes intuitive sense given that the NTK is the ultimate factor that determines the learning regime (rich, lazy, or somewhere in between).

2. The formula for $\delta=\eta_w a^2 - \eta_a ||w||^2$ also is nicely interpretable, elucidating the interplay between learning rates and weights magnitude.

3. The authors test their theory in various settings and observe a close alignment between the experimental verification and the theoretical predictions, which makes me confident of the correctness of their theory.

**Weaknesses:**

1. I would have appreciated more on the NTK dynamics.  There is a clear connection between $\delta$ and (part of) the NTK $K$. I wonder whether this connection can be made more explicit by studying the NTK dynamics $\dot{K}$. This would make the connection between the conserved quantity and the feature learning more explicit.

2. Some equations are not included. The authors justify this in Line 1022: "We omit the solution due to its complexity, but provide a notebook used to generate our figures encoding the solution". I would just provide the formulas for the sake of completeness. Also, I do not see the equations describing the kernel distance S, which is related to the NTK movement, thus to my previous point.

3. In section 4 the paper includes various extensions, including wide networks. It would have been nice to recover known results by relating the conserved quantity $\Delta$ and specific parameterizations of the network as a function of the width. This would have connected existing results in the NTK and $\mu$P literature to the conserved quantities identified in this paper and the learning regime.

4. The connection between the scale of the initialization and the learning rate is not entirely new. This is quite obvious at grokking problems, where to exhibit grokking is often necessary to play with the initialization parameters of different layers. In this context, Kumar et al [61] already clarified that the delayed generalization was caused by the NTK dynamics transitioning from lazy to rich regimes.

Overall, I am in favor of the paper's acceptance because it devised a model where the three different learning regimes are clean and crystallized.

**Questions:**

See weaknesses

**Limitations:**

Addressed.

---

> ### Author Rebuttal · Authors · 2024-08-05
>
> Thank you for your thorough review and constructive suggestions. We appreciate your positive feedback and address the weaknesses you highlighted individually. We hope this will increase your confidence in the importance of our work.
>
> **NTK dynamics.**  This is a good point; we can in fact study the dynamics of the NTK matrix directly, which leads to an argument similar to the one provided in lines 239 - 264. Here we outline this analysis, which we will add to appendix A.
>
> The dynamics of the NTK matrix $K = X M X^\intercal$ is determined by $\dot{M}$. From Equation 3 in the main text, we can write $\dot{M} = \frac{2 \|\beta\|}{\kappa} (I_d + \hat{\beta}\hat{\beta}^\intercal) \partial_t \|\beta\| + \frac{\kappa - \delta}{2} \partial_t (\hat{\beta}\hat{\beta}^\intercal)$. From this expression we see that the change in $M$ is driven by two terms, one that depends on the change in the magnitude of $\beta$ and another that depends on the change in the direction of $\beta$. As done in the main text, we consider the limits as $\delta\to\pm\infty$ and when $\delta = 0$ to identify different regimes of learning. For $\delta\to\infty$, the coefficients in front of both terms vanish, and thus, irrespective of the trajectory taken from $\beta(0)$ to $\beta_*$, the change in the NTK is vanishing, indicative of a lazy regime. For $\delta \to -\infty$, the coefficient for the first term vanishes, while the coefficient on the second term diverges. Here, the change in the NTK is driven solely by the change in the direction of $\beta$. This is why for large negative delta we observe a delayed rich regime, where the eventual alignment of $\beta$ to $\beta_*$ leads to a dramatic change in the kernel. When $\delta = 0$, the coefficients for both terms are roughly of the same order, and thus changes in both the magnitude and direction of $\beta$ contribute to a change in the kernel, indicative of a rich regime.
>
> **Adding equations for completeness.** We agree that it is important to include expressions for completeness. We will include explicit solutions to the one hidden neuron model dynamics in the appendix. Specifically, we will write out the expression for $a(t)$, which is currently omitted because of its verbosity. We will also write out the expression for the kernel distance presented in figure 3c and include this in the added subsection of the appendix describing the kernel dynamics (as we discussed above).
>
> **Connections to width-dependent neural network parameterizations.** We agree, discussing how our analysis of finite-width networks connects to existing analyses of infinite-width networks is important. We outline a discussion we will add to the main:
>
> In a width-dependent parametrization, such as mean field or NTK, the random initialization of weights leads to a distribution over conserved quantities. For example, in the two-layer setting, the mean-field parametrization leads to the per-neuron conserved quantity being 0 in expectation, but with a non-vanishing variance. The NTK parametrization has the same distribution over conserved quantities, but at a larger function scale, leading to lazy dynamics. Thus, between NTK and mean-field, a change in function scale modulates the degree of feature learning. In our work, we show that relative scale between layers (i.e. $\delta$) is a separate knob that can also be tuned to influence the degree of feature learning.
>
> When considering these parametrizations in the infinite-width limit we can recover a phase diagram analogous to our results in the finite-width setting. Reference [14] previously considered $f(x) = \frac{1}{\alpha}\sum_{i\leq k}a_i\sigma(w_i^\intercal x)$ with weights initialized as $a_i \sim \mathcal{N}(0, \beta_a^2)$ and $w_i \sim \mathcal{N}(0, \beta_W^2I_d)$ as width $k\to\infty$. They obtain a phase diagram at infinite width capturing the dependence of learning regime on the overall function scale $\beta_a\beta_W/\alpha$ and the relative scale $\beta_a/\beta_W$. The resulting phase portrait is analogous to ours in Figure 1 (b), which considers the conserved quantity $\delta$ rather than the relative scale $\beta_a/\beta_W$. In particular, there is a lazy regime, which is always achieved at large scale (just as in the large-$\tau$ regions of Figure 1 (b)), but is also achieved at smaller scale if the first layer variance is sufficiently larger than the second (as in the downstream initializations at small $\tau$ in Figure 1 (b)). On the other side of the phase boundary is the infinite width analog of rapid rich learning, with all neurons condensing to a few directions. This is induced either at small function scale, or at larger function scale if the relative scale is sufficiently large, such that $W$ learns faster than $a$.
>
> **Connections to prior work in grokking.** Kumar et al. [61] explored how grokking arises as the result of the transition from lazy to rich dynamics, studying this transition as a function of overall function scale, initial NTK alignment with the test labels, and train set size. However, we add more nuance to this picture in that we show how grokking can be induced not just by modulating the overall function scale, but by simply changing the initialization geometry, that is by scaling the embedding weights for positional and token embeddings without affecting overall scale (as visualized in Figure 5 (d)). This leads to the complex phase diagram of time to grokking as a function of both scale and geometry, as presented in Figure 11. We believe this adds to the picture presented in Kumar et al. by showcasing the unique role of layer-wise initialization scales.
>
>
> We hope we addressed your points regarding our work. If we have, we would appreciate it if you would consider raising your score to reflect this. Thank you again for your constructive feedback.

---

> > ### Comment · Reviewer_JJEZ · 2024-08-13
> >
> > I sincerely thank the authors for the effort spent on this rebuttal. The connection between some of the results of this work and the NTK dynamics is now clear.
> >
> > **Connections to width-dependent neural network parameterizations.**
> >
> > I thank the authors for this clarification. This helps me better understand the connection of their work in the context of the scaling limit literature.
> >
> > The reason I am not confident enough to raise my score (which still favors acceptance) is that two of these regimes are already sort of well-understood (lazy vs rich) in the series of existing works that the authors mention. Thus, I believe the crucial contribution lies in the third regime (delayed rich). Having non-vanishing dynamics of the NTK is precisely achieved by correctly setting the initialization variance, learning rate, and output scale of the model. In a sense, $\delta$ is to some extent explicitly already controlled under $\mu$P and related works (e.g. https://arxiv.org/abs/2310.17813).

---

> > > ### Author Response · Authors · 2024-08-14
> > >
> > > Thank you for your reply. We’re glad that our response helped clarify the connection between our work and the scaling limit literature. We are also happy to hear that you favor acceptance of our work.
> > >
> > > We certainly agree that the delayed rich regime is intriguing and to our knowledge has not been studied before. While the lazy vs. rich dichotomy is well-studied in infinite-width settings, our analysis (alongside recent work by Xu and Ziyin [reference 17] and analysis of diagonal linear networks [reference 12]) stands out as one of the few analytically tractable models for the transition between lazy and rich learning in a finite-width network. We believe that introducing solvable systems where precise statements can be made, both in terms of dynamics and implicit bias, is a valuable contribution. Additionally, throughout our analysis we highlight the importance of conserved quantities in determining the learning regime, a perspective not considered in these prior works, nor in the work you mentioned (the difference in feature learning between a μP parameterization and an NTK parameterization is due to a change in function scale, not $\delta$, which remains the same in distribution for both parameterizations).
> > >
> > > Thank you for the time and effort you put into reading and reviewing our work.

---

### Official Review · Reviewer_KNvb · 2024-07-09

**Soundness:** 3
**Presentation:** 2
**Contribution:** 3
**Rating:** 7
**Confidence:** 3

**Summary:**

The paper studies the learning dynamics of a deep network by leveraging the conserved quantities due to symmetries

**Strengths:**

The theoretical finding that unbalancedness in the layers drives feature learning is a novel and interesting insight

The technical tool of using symmetries and conserved quantities to characterize the dynamics is also novel

Finding exact solutions can help us understand better the causal aspects of the phenomena in deep learning, and should be encouraged

Overall, I am positive towards this work

**Weaknesses:**

I think the paper could improve by explaining its results in more detail. For example, I find the following points rather confusing

1. Does theorem 4.1 hold after training? Or it holds at any time in training？

2. The paper argues that it finds an exact solution, but I am sure what this is referring to. Is theorem 4.1 the "exact solution" the paper finds? If so, what is it a solution to?

3. The title claims "rapid feature learning", but almost no place in paper discuss what it means to be rapid, nor what it means to be doing "feature learning." The only part in the paper that I find to be peripherally related to this claim is the discussion in lines 380-383. This result holds true without needing any result from sections 3-4. Then, what is the point of 3-4? If the title is the main claim of the paper, can the authors write it out in a more explicit manner, and with much greater mathematical detail?

4. In figure 5, a crucial quantity is $\alpha$, but it is not explained in the context. What is $\alpha$? It seems to me that this quantity is the scaling factor introduced in lines 171-172? Is this case? Even if so, the authors should have explained it much better in the immediate context

5. To be fair, I think the authors are putting too much content both in the appendix and in the main text. It feels to me much better to move one of section 3 or 4 to the appendix and expand what is in the main text to make it much more readable

6. A constructive suggestion is that it would be quite insightful to compare the results in section 4 to the results derived for SGD in deep linear models in https://arxiv.org/abs/2402.07193. SGD and GD have very different behaviors for these models and this discussion will be very helpful to the audience

**Questions:**

See the weakness section

---

> ### Author Rebuttal · Authors · 2024-08-05
>
> Thank you for taking the time to thoroughly review our paper and highlight areas needing further clarification. We appreciate your positive feedback on our work. We will address each of the weaknesses you mentioned individually, hoping this will enhance your confidence in the significance of our research.
>
> **Theorem 4.1.** This theorem holds throughout training. We will make this more clear in the theorem statement by adding a time dependence to $\beta_i(t)$ and write for all $t \ge 0$.
>
> **Exact solution.** The exact solutions we refer to are for the gradient flow dynamics of $a$ and $w$ in the minimal model we present in section 3. The gradient flow dynamics for this model boil down to a complex coupled system of nonlinear ODEs shown in equation 1. As stated on line 191, we solve this system of ODEs exactly, derived in appendix A in detail. See figures 2, 3, and 4; in all these figures the dashed black lines represent theoretical predictions (i.e exact solutions) while the colored lines are empirical (the results of training on a computer). We will make this more clear by referencing section 3 when we discuss exact solutions in the paragraph titled “our contributions” in section 1.
>
> **Rapid feature learning.** As discussed in the related work section (lines 45-62), feature learning in the context of this work is synonymous with rich learning. We mathematically define rich learning on lines 96-101, as a change in the NTK through training measured by the kernel distance metric proposed in Fort et al. 2020. The main claim of our paper is that an upstream initialization leads to rapid feature learning *relative* to a balanced or downstream initialization. By "rapid" in our title, we refer to this specific claim. We demonstrate this empirically in figure 1. Sections 3 and 4 provide the necessary analysis to prove this claim in section 5. This explains why our title states “exact solutions reveal how unbalanced initializations promote rapid feature learning.” To clarify this further, we will revise the writing on lines 63-84 under "Our contributions." This section outlines our contributions and sets up the paper’s layout. We aim to make this section more concise to clearly explain why we use the approach in sections 3 and 4 to prove our main claim and its connection to the title.
>
> **Alpha in figure 5.** Thank you for catching this. Yes you are correct, $\alpha$ in this context is equivalent to the scaling factor used in lines 171-172.  It is also described in the caption for figure 1 and in the experimental details in appendix D, however, we forgot to thoroughly describe it in the caption for figure 5. We will revise the caption to make this more clear.
>
> **Content distribution between the main and appendix.** We appreciate this comment and we agree that the paper is dense. We have tried to make the work as readable as possible, but we believe that section 3 and 4 are necessary steps to build the analysis used in section 5. With the additional page allotted for the final version of this paper we have moved up some aspects of the appendix into section 3 and 4 which should make it more readable, added clarifying transitions between the sections, and expanded the content in section 5 to clarify its connection to the previous sections and the main claims of our paper. We think these changes will significantly improve the readability.
>
> **Related work on SGD.** Thank you for highlighting this reference. We will include it in our discussion on the limitations of our work (lines 405-411). In this section, we explain how SGD disrupts the conservation laws that are central to our study.  As you pointed out, one of our key findings is how conserved quantities arising from symmetry affect the degree of feature learning. Other works have focused on the influence of stochasticity in SGD on these conserved quantities. We agree integrating these analyses to understand the influence of stochasticity on the degree of feature learning would be an insightful direction for future work. However, this is beyond the scope of our current paper, which is already densely packed with content.
>
> Thank you again for your constructive feedback. We hope we addressed your points regarding our work. If we have, we would appreciate it if you could raise your score to reflect this. Thank you!

---

> > ### Comment · Reviewer_KNvb · 2024-08-09
> > **Thanks for the reply**
> >
> > Thanks for the detailed explanation. The answer removed my concerns about the work. I will raise to 7. Although I am positive towards the work, I do not find the results significant enough for me to strongly support it.

---

> > > ### Author Response · Authors · 2024-08-13
> > >
> > > We are happy to hear that our answers removed your concerns and that you are positive towards our work. Thank you again for your thoughtful and constructive feedback.

---

### Official Review · Reviewer_hh3v · 2024-07-11

**Soundness:** 4
**Presentation:** 4
**Contribution:** 3
**Rating:** 8
**Confidence:** 4

**Summary:**

The authors derive exact solutions to a minimal model that transitions between lazy and rich learning, elucidating how unbalanced layer-specific initialization variances and learning rates determine the degree of feature learning in a finite-width network.
They provide evidence that this unbalanced rich regime drives feature learning in deep finite-width networks, promotes interpretability of early
layers in CNNs, reduces the sample complexity of learning hierarchical data, and decreases the time to grokking in modular arithmetic.

**Strengths:**

1. The paper is very well written:
- clear visuals and exposition, going from a minimal model and gradually adding complexity and realism.
- The quantities of interest (e.g. $\delta$) give an intuitive picture.
- The notations are very clear and easy to follow. One small comment is that it might be better to use a different index for $\theta$ and $x$.
- The contributions are distinguished from previous works and relevant connections to them are clearly stated.

2. Addressing the rich regime is arguably more interesting than the lazy regime and is much less studied, making the questions addressed here timely.

3. The experiments presented in Fig 5 show that the effects studied analytically in previous sections also appear in realistic settings, making this work relevant to a broad audience.

**Weaknesses:**

1. To yield the exact solutions, the authors evoke the assumption of whitened input, which is rather unrealistic. The low-rank case is also addressed but then one needs to additionally assume that the interpolating manifold is one-dimensional to find the solution in terms of $\delta$ exactly.

missing citations:
- In line 398, when mentioning grokking and the transition from lazy to rich learning - it is worth noting Ref. [1].

[1] https://arxiv.org/abs/2310.03789

**Questions:**

1. In Ref. [1] which you mention, the formula for the NTK with unbalanced LRs is (using your notation) $K = X  (\eta_a a^2 I_d + \eta_w ww^T) X^T$  (eq. therein), i.e. the LRs have switched places. I'm not sure where this discrepancy comes from or if it is an error.

2. Line 145 - can you explain why in the infinite width limit non-linearities act linearly? It is true that the model becomes linear in the weights, but do you mean that each layer acts linearly on its pre-activation?


[1] https://arxiv.org/abs/2401.07085

**Limitations:**

The authors adequately addressed the limitations.

---

> ### Author Rebuttal · Authors · 2024-08-05
>
> We appreciate your comprehensive review and the time you have taken to suggest improvements for our study. We are also happy to hear you think our work is a timely and important contribution to the field. We will respond individually to the weaknesses and questions you raised regarding our paper:
>
> **Notation for indices.** We agree with you that using a different index for parameters, data, and layers can help with readability. This is a more comprehensive change, but something we will implement in our updated draft.
>
> **Whitened input assumption.** We agree that the assumption of whitened input for our minimal model is quite strong, although a common assumption used in many prior works (see Saxe et al. 2014 for example). That said, we actually do relax this assumption throughout the analysis. As you noticed, a strength of our work is that we start with a simplified setting where we can derive exact solutions, then “gradually adding complexity and realism”, we show that the key-takeaways remain. To be specific, section 3 is composed of three parts:
>
> 1. *Deriving exact solutions in parameter space.* In order to solve analytically the coupled system of ODEs we need whitened input.
> 2. *Interpreting the dynamics as preconditioned gradient flow in function space.* This analysis does not require whitened input. We only require that $X^\intercal X$ is full-rank such that there exists a unique OLS solution. As shown in equation 2 we did not replace $X^\intercal X$ with $\mathbf{I}_d$.
> 3. *Identifying the implicit bias when $X^\intercal X$ is low-rank.* Actually the main theorem (Theorem A.2) in this analysis does not make any assumptions on the dimension of the null space. This theorem expresses the interpolating solution as a solution to a constrained optimization problem (which in general can only be solved numerically). We can then interpret the objective function being minimized in different limits of $\delta$. If we additionally assume the null space is one-dimensional then we can analytically express the solution to this constrained optimization problem.
>
> In summary, we do need the whitened input assumption for the first part of section 3, however the remaining subsections relax this assumption to full-rank and then low-rank. We have added comments in this section to make this more clear and we have brought theorem A.2 up from the appendix into the main to make it very clear that the assumption on a one-dimensional null space is only needed to solve analytically the constrained optimization problem in this theorem.
>
> **Missing citation.** Thank you for this additional reference we were unaware of. We have added this citation to line 398 as suggested.
>
> **Formula for the NTK with unbalanced LRs.** The formula (equation 10) in the paper you referenced seems incorrect to us. Here is an explanation for the form of the NTK in our paper (we also added this derivation to the top of appendix A):
>
> As discussed in our notation section, $K_{ij}  = \Theta(x_i, x_j; \theta) = \sum_{k = 1}^p \eta_{\theta_k} \partial_{\theta_k} f(x_i;\theta)\partial_{\theta_k} f(x_j;\theta)$. For the minimal model we study $f(x;\theta) = aw^\intercal x$ implies $\partial_a f = w^\intercal x$ and $\partial_w f = ax$. Thus, we see that the term associated with the gradient of $a$, and thus the learning rate $\eta_a$, depends on $w$, while the term associated with the gradient of $w$, and thus the learning rate $\eta_w$, depends on $a$. That is why the NTK is defined by the matrix $M = \eta_wa^2 \mathbf{I}_d + \eta_aww^\intercal$ in our expression. It seems to us that the equation from the paper you referenced has a notational mistake.
>
> **Clarifying line 145.** We agree this sentence is misleading. We were referencing a common simplification used in many infinite width analyses where the features before and after the nonlinearity are assumed to be of the same scale (see “A Spectral Condition for Feature Learning” by Yang et al. for example). These works will often build intuition for their analysis by replacing the nonlinearities with linear activations, without affecting their results. However, as line 145 is currently written, it sounds like the nonlinearities are always linear on the preactivation, which is not true. Thus, we have modified this sentence to read, “In this limit, analyzing dynamics becomes simpler in several respects: random variables concentrate and quantities will either vanish to zero, remain constant, or diverge to infinity.” Also, it is worth noting we have added a longer discussion on how our theory connects to infinite width limit analyses in our updated manuscript (see responses to Reviewers JJEZ and K8a2 for details).
>
> Please let us know if you have any other questions regarding our work. Overall we are happy to hear that you enjoyed our paper and we hope you will consider our paper an important contribution to the NeurIPS community. Thank you!

---

> > ### Comment · Reviewer_hh3v · 2024-08-08
> >
> > I have read the author's response and will keep my score

---

> > > ### Author Response · Authors · 2024-08-13
> > >
> > > Thank you again for your positive and constructive feedback.

---

### Author Rebuttal · Authors · 2024-08-05

We would like to thank all the reviewers for their careful and detailed comments. We greatly appreciate the time and effort you put into reviewing our paper, which we believe has significantly improved our work. We have addressed each reviewer’s questions individually and provided linked responses for similar questions. We hope you will consider our paper a valuable contribution to the NeurIPS community. We look forward to your continued feedback!

---

### Decision · Program_Chairs · 2024-09-25

**Decision:**

Accept (spotlight)

**Comment:**

This paper studies minimal models of lazy and rich learning with exact solutions and uses their derivations to gain insights into the learning dynamics of neural networks.
The latter is controlled by the quantity delta that combines layer-specific initialization variances and learning rates. It is argued that a layer-wise initialization out of balance can promote (delayed) feature learning (also in a finite-width network), which is related to Grokking. In contrast to previous work, the authors also consider layer-wise scale imbalance that does not affect the overall function scale.

All reviewers agree that the presented work has merit for the NeurIPS community and recommend acceptance.
They particularly highlight positively that the paper is well written (even though dense) and appreciate the fact that (small) tractable models are introduced that provide insights into the rich learning regime and also include finite-width results.